# ExCyTIn-Bench: Evaluating LLM agents on Cyber Threat Investigation

Yiran Wu [1]  Mauricio Velazco [2]  Andrew Zhao [3]  Manuel Raúl Meléndez Luján [2]  Srisuma Movva [2]
Yogesh K Roy [2]  Quang Nguyen [2]  Roberto Rodriguez [2]  Qingyun Wu [4]  Michael Albada [2]  Julia Kiseleva [2]
Anand Mudgerikar [2]

## Abstract

We present **ExCyTIn-Bench**, the first benchmark to **E**valuate an LLM agent **X** on the task of **Cy**ber **T**hreat **In**vestigation through security questions derived from investigation graphs. Real-world security analysts must sift through a large number of heterogeneous security logs, follow multi-hop chains of evidence to investigate threats. With the developments of LLMs, building LLM-based agents for automatic threat investigation is a promising direction. We construct a benchmark from a controlled Azure tenant including a SQL environment covering 57 log tables from Microsoft Sentinel and related services, and 7542 generated questions. We leverage security logs extracted with expert-crafted detection logic to build threat investigation graphs, and then generate questions with LLMs using paired nodes on the graph, taking the start node as background context and the end node as answer. Anchoring each question to these explicit nodes and edges not only provides automatic, explainable ground truth answers but also makes the pipeline reusable and readily extensible to new logs. Our comprehensive experiments on the test set with different models confirm the difficulty of the task: the best model so far can achieve a reward of 0.606, leaving much headroom for future research. The code is available at SecRL.

## 1. Introduction

The growing reliance on digital services for critical functions worldwide underscores the need to secure our digital future. Meanwhile, cyberattacks are growing in quantity, variety, and sophistication. For example, cloud environment intrusions increased by 75% from 2022 to 2023 (Crowd-Strike, 2024). Although traditional defenses like behavioral analysis, malware signature matching, and anomaly detection can mitigate threats (Hossain et al., 2017; Hassan et al., 2020), attackers continue to develop tactics to evade them (Mahboubi et al., 2024). Thus, human-led threat investigations have become critical, requiring analysts to manually go through system and network logs, apply reasoning, and leverage domain expertise to detect and respond to threats (Alevizos & Dekker, 2024).

Meanwhile, advancement of Large Language Models (LLMs) has enabled astonishing achievements in complex tasks (Yang et al., 2024; Hong et al., 2023; Wang et al., 2023a;b; 2024b; Chang et al., 2026), that LLM agents can understand observations and select actions in complex environments such as code interpretation and database interaction to perform sequential actions (Yang et al., 2023; Zhang et al., 2023b; Wu et al., 2024a; 2023; Li et al., 2023a). Also, LLMs trained with enormous corpora of text provide them a wealth of knowledge across a range of domains (Zhang et al., 2024b), including cybersecurity knowledge. Thus, Cyber-Security threat investigation is a promising area for the application of LLM-based autonomous agents, as previous works have shown that LLMs are capable of multi-step observation, reasoning, and actions, which are key components for successful investigation and detection of potential threat actors and indicator of compromises (IoCs) (Hassanin & Moustafa, 2024; Levi et al., 2025; Zhang et al., 2025).

To assist development, a standardized benchmark is needed to evaluate the cybersecurity investigation capability of LLM agents. The benchmark should resemble real-world threat investigation scenario with a critical mass of security event logs, heterogeneity across real-world multi-stage security incident types. However, existing literature focuses on evaluating LLMs from a knowledge memorization perspective (Yong et al., 2025; Jing et al., 2024), other than targeting the investigation and reasoning ability of LLM agents. For example, CTIBench (Alam et al., 2024) constructs a multi-choice Q&A to evaluate LLMs on Cyber Threat Intelligence knowledge, SECURE (Bhusal et al., 2024) evaluates LLMs on security extraction and understanding.

[1]Pennsylvania State University [2]Microsoft Security AI Research [3]Tsinghua University [4]AG2AI. Correspondence to: Yiran Wu <yiran.wu@psu.edu>, Anand Mudgerikar <amudgerikar@microsoft.com>.

*Proceedings of the 43rd International Conference on Machine Learning*, Seoul, South Korea. PMLR 306, 2026. Copyright 2026 by the author(s).

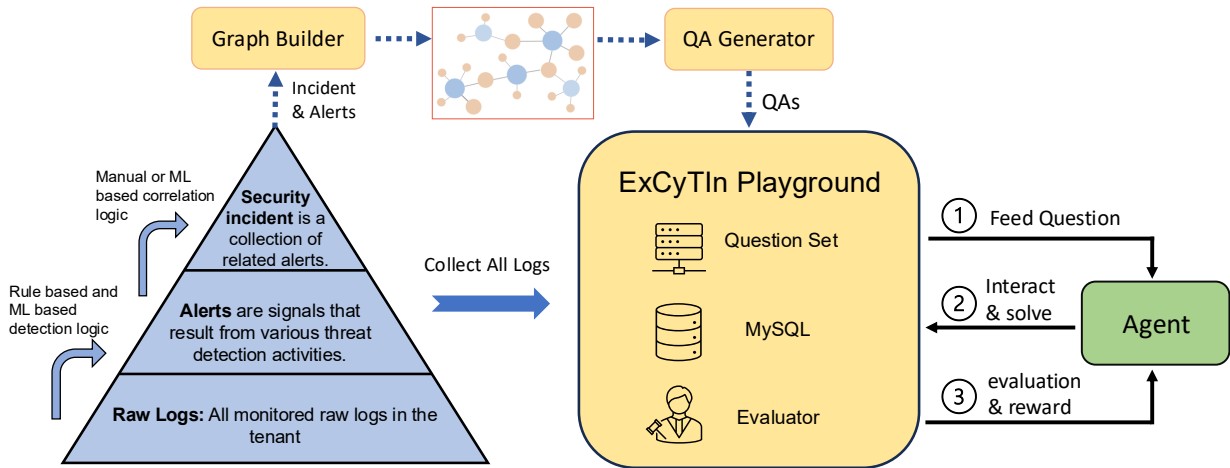

*Figure 1.* Overview of constructing **ExCyTin-Bench**. *Left Triangle:* A complete Azure tenant with security logs, alerts and incidents. *Middle Square*: The ExCyTin playground, with the questions, environment and evaluation ready for evaluating agents.

To fill this gap, we build **ExCyTIn-Bench** that evaluates LLM agents on cybersecurity threat investigation. Construction of the benchmark consisted of three steps (See Figure 1): *1. Data Collection.* From a fictional Microsoft Azure tenant, we collect 59 distinct log tables spanning eight *multi-stage attack chains*, each comprising multiple attack techniques and corresponding events. *2. Question Generation.* We propose a principled method to construct bipartite incident graphs from the alerts and entities involved in the attacks to be used to generate QA pairs. It results in a total of **7542** questions, where we split a test dataset of 589 questions for testing. *3. Environment Construction.* We construct a MySQL Docker image in which agents can submit queries and receive feedback similar to InterCode (Yang et al., 2023). The database queries are treated as actions, and the execution result as observations. Since our questions are generated from paths in the incident graphs, we can assign partial rewards if the agents find any intermediate information along the path.

We test with a wide range of current language models, including proprietary and open-source models, and models with different sizes and types (chat, reasoning, etc) in Section 4. We include a detailed analysis from the perspective of performance, behavior, and efficiency. We found that our benchmark is challenging even among the latest, highest-performing models, with `Claude-Opus-4.5` achieving the highest reward of 0.606. We also test different methods (e.g., ReAct (Yao et al., 2022), Expel (Zhao et al., 2024), Best-of-N, Self-Reflection (Shinn et al., 2023)) to help understand how different prompting and test-time scaling strategies affect performance on our benchmark. To summarize, our contributions are the following:

- We release ExCyTIn-Bench, which, to the best of our knowledge, is the first benchmark to evaluate LLM agents

on threat investigation tasks. The benchmark is built on real-world security logs generated from real-world attacks, and requires the agents to query logs to investigate.

- We propose a new QA generation method with LLM from bipartite incident graphs, where each question is non-repetitive and anchored to explicit nodes and edges.

- We conduct comprehensive experiments on the benchmark to provide insights for future directions.

## 2. Related Work

Our goal is to create a cybersecurity threat investigation benchmark for LLM agents, which is closely related to LLM-based agents and cybersecurity: this requires LLM agents to have cybersecurity domain expertise to be able to explore system logs, analyze suspicious behavior, and answer security-related questions. On the other hand, we build a SQL environment for LLM agents to interact with, to test models' ability on effective and efficient SQL generation.

**LLMs in Cybersecurity.** Most recent LLM-based cybersecurity efforts focus on knowledge memorization and information extraction (Liu et al., 2024; Li et al., 2023b; Liu, 2023; Sultana et al., 2023; Yong et al., 2025). CTIBench (Alam et al., 2024) evaluates LLMs' understanding of the threat landscape via the MITRE ATT&CK framework (MITRE, 2025), while Crimson (Jin et al., 2024) fine-tunes an LLM to map CVEs to MITRE ATT&CK techniques and generate actionable insights. SECURE (Bhusal et al., 2024) benchmarks models on security extraction, comprehension, and reasoning. (Siracusano et al., 2023) aggregates public CTI reports with structured intelligence, and (Gao et al., 2021) extracts threat behaviors from unstructured OS-CTI text. Perrina et al.'s tool (Perrina et al., 2023) produces CTI reports from entity graphs, and (Rigaki et al., 2023)

**Tables**

| AADManagedIdentitySignInLogs |
| --- |
| AADNonInteractiveUserSignInLogs |
| CloudAppEvents |
| DeviceInfo |
| EmailEvents |
| IdentityDirectoryEvents |
| MicrosoftGraphActivityLogs |
| Usage |

⋮ *(57 tables in total)*

**Column Length**

| Num Columns | <20 | 20-40 | 40-60 | >60 |
| --- | --- | --- | --- | --- |
| Table Count | 20 | 22 | 8 | 7 |

**Example table: SignInLogs**

| TenantId | TimeGenerated | Identity | AlternateSignInName | ... |
| --- | --- | --- | --- | --- |
| e34d562e-ef12... | 2024-06-20 07:00:04 | u754 | u754@ash.alpineskihouse.co | |
| e34d562e-ef12... | 2024-06-20 07:03:33 | John Anderson | jad@ash.alpineskihouse.co | |
| e34d562e-ef12... | 2024-06-20 07:06:39 | Marry K | marry@ash.alpineskihouse.co | |
| ... | | | | |

*Figure 2.* Overview of the database. We collect a total of 57 tables. The number of columns from these tables vary from 8 to 139.

uses reinforcement learning to simulate LLM-driven attacks on network topologies. Finally, CyBench (Zhang et al., 2024a) focuses on capture-the-flag (CTF) tasks. Although some prior work also employs graphs (Gao et al., 2021; Hossain et al., 2017), our graph-based approach differs in both concept and construction.

**Benchmark LLMs in Interactive Environments.** (Hu et al., 2024) benchmarks LLM-based agents on data analysis tasks through an execution environment. (Nan et al., 2023) introduces a database question answering system that LLMs need to interact with a SQL interpreter, reason, and organize the results. (Yang et al., 2023) creates interactive code environments (Bash, SQL, and Python) based on static datasets (Yu et al., 2018; Lin et al., 2018; Austin et al., 2021) for LLM to act on. (Huang et al., 2024) build a dataset of machine learning tasks that LLMs need to perform actions like reading/writing files, executing code. SWE-Bench (Jimenez et al., 2023) builds a dataset on real-world software engineering problems. (Xu et al., 2024) introduces a benchmark that supports cross-environment GUI tasks over websites, desktop computers, or mobile phones.

**LLMs in Text-to-SQL.** Text-to-SQL benchmarks (Yu et al., 2018; Li et al., 2023c) are proposed to test models on generating SQL queries given a question. Many works have been proposed to solve this task (Askari et al., 2024; Talaei et al., 2024). (Wang et al., 2024a) proposes a multi-agent framework, including a decomposer agent for Text-to-SQL generation and two auxiliary agents for tool execution and error refinement. (Gao et al., 2023) provides a systematic review of prompt engineering for Text-to-SQL generation. C3-SQL (Dong et al., 2023), StructGPT (Jiang et al., 2023), Din-SQL (Pourreza & Rafiei, 2024) propose frameworks targeting SQL generation that consist of several stages with different strategies such as self-consistency (Wang et al., 2022) or query decomposition. StateFlow (Wu et al., 2024b)

introduces a framework with state and transitions to control the data exploration and selection in SQL tasks.

## 3. ExCyTIn-Bench

### 3.1. Data collection

We collected data from Azure tenant "Alpine Ski House", which is a fictional company used for security products demonstration. It is set up just like a real-world tenant: with similar tenant activities and fully configured security monitors to collect all kinds of event logs (e.g., "EmailEvents" records email events including receivers and senders). We collect 57 tables in total (Figure 2) with the columns containing different data types.

From the tenant, we simulate eight distinct attacks chains (Table 1) *spanning multiple stages* from initial access to later-stage objectives; at each stage, several actions corresponding to different MITRE attack techniques are applied (See Figure 25-39 in appendix for report). Although the benchmark contains eight top-level scenarios, each scenario comprises many individual attacker actions, resulting in **hundreds of intermediate attack events** that need to be reconstructed from evidence distributed across multiple tables. Further, each attack is a *real-world attack* that has happened before. For example, incident 5 is a simulation of "Manatee Tempest Ransomware" (2021), and incident 134 covers the "BEC and Account Take-over" (2024) attack. These known attacks have been studied, and their behavior is tracked and recorded in table "SecurityIncident" and "SecurityAlert".

For each incident, we segment these attack chains based on time ranges resulting in time segments varying between 2 hours and 5 days, with no temporal overlap. We want to emphasize that each attack chain can be taken as an *independent tenant episode*: for each attack, the noise data (benign

| ID | Title | Time | #Alerts | #Qs | Labels |
|----|-------|------|---------|-----|--------|
| 5 | Operation Alpine Lockbit: Multi-Stage Manatee Tempest Ransomware Campaign | 47 | 2770 | 98 | Ransomware, Credential Theft, Lateral Movement |
| 34 | Macro-Enabled Document Dropper with Power-Shell Backdoor Deployment | 80 | 430 | 82 | Backdoor, Persistence |
| 38 | Multi-Stage Fileless Attack | 25 | 157 | 11 | Process Injection, Covert C2 |
| 39 | Operation Alpine Storm: Human-Operated intrusion chain | 475 | 1873 | 98 | Phish URL, Domain Compromise, Credential Harvest, Defense Evasion |
| 55 | Phishing-Enabled ADFS Key Exfiltration and Lateral Movement Campaign | 7739 | 1093 | 100 | Spear-Phish Email, Lateral Movement, Persistence |
| 134 | Multi-Stage Business Email Compromise and Account Takeover Attack | 17 | 352 | 57 | BEC Fraud, Compromised Credentials, Password Spray |
| 166 | SAP Financial Manipulation Attack | 88 | 430 | 87 | BEC Fraud, SAP Access, Data Ex-filtration |
| 322 | Domain Credential Harvest Attack | 11 | 352 | 56 | Proxy Evasion, Credential Phish, Domain Compromise |

*Table 1.* Table of collected incidents, including time span (in minutes), number of alerts, number of questions generated, and labels.

events) are random simulated and the attack events are also different. Although we are using one tenant, we actually yield **eight self-contained, distributionally different investigation environments**. We also compiled a continuous log stream covering the entire sequence of incidents spans 44 days. This is a much more complex but possible real-world scenario, where SOC analysts typically do not have prior knowledge of attack timelines, making detection and analysis considerably more challenging (see Appendix C.3 for more details on data collection).

### 3.2. Question Generation

We want to create questions that can measure LLM investigation skill: to answer a question, the LLM agent needs to probe into the log data, analyze, and link related events to find the source of the malicious activity. While manual creation of these questions is expensive and does not scale, LLMs have proven useful in QA generation (Bhusal et al., 2024; Alam et al., 2024). A straightforward way is to ask an LLM to read all the details of an incident and draft Q&As. However, we found that this method produced generic questions that ignore the key concepts used to link the alerts, events, and entities together into a cohesive investigation. Moreover, these questions can lack a deterministic answer or ask about security knowledge not present in the database.

We further investigated the behavior of human SOC analysts and the collected data to solve this issue. We gain two insights that motivate us to build a bipartite threat investigation graph, and then use the graph for QA generation: 1. Manual incident reviews confirm that SOC investigations are inherently relational: start with a seed alert or entity, analysts pivot related IoCs, pull connected events, then iterate to identify suspicious patterns. 2. In our data, we observe that each step of the attack is recorded in the "SecurityIncident" or "SecurityAlert" tables (explained in Section 3.1), which are suitable grounding sources for question generation.

**Bipartite Graph Construction.** We create $G = (U, V, E)$ for each incident, where the vertices set is partitioned into `Alert` vertices ($U$) and `Entity` vertices ($V$), and every edge from a vertex $u \in U$ connects one in $V$. The alert table contains an entity column that documents a list of entities associated with these alerts, and each entity can also appear in different alert entries. Thus, we can easily build such a graph from the incident and alert table.

The bipartite graph models the investigation process closely, and we can view the investigation of a SOC analyst as walking on the graph. The analyst starts with one given alert $u_s$ and a set of related entities $V_s$ (connected to $u_s$). Using this information and entities of interest (IoCs), the analyst can query the database to find related events/alerts $u_i$ and entities $V_i$. Note that the alert-entity graph is only a small sub-graph traced by the analyst during their investigation. This sub-graph is part of a much larger alert-entity graph of the environment, which includes all alerts and entities in the environment not relevant to the particular incident.

**QA Generation From Graph.** We then use LLMs to generate questions from the graph (See Algorithm 1). We pick any two `Alert` vertices $u_s$ and $u_e$ as the starting and ending points to construct one question. We extract the entities connected to one alert that is farthest from another alert, so that we can have a longer path (GETFARTHESTENTITIES in Algorithm 3). We select $k$ entity vertices $V_s$ as the starting vertices ($k = 2$) and one entity $v_e$ connected to $u_e$, since we want to give more context to the agent and some entities might not be useful. We instruct the LLM to use $u_s$ and $V_s$ as the background context, and write a question using $v_e$ and $u_e$, with $u_e$ as the final answer to this question (prompts in Figure 15 & 16). The question tests whether an agent can start with one alert and do investigations towards the goal.

For each question, we can obtain the shortest path between the selected alerts. We take this path in our graph as one

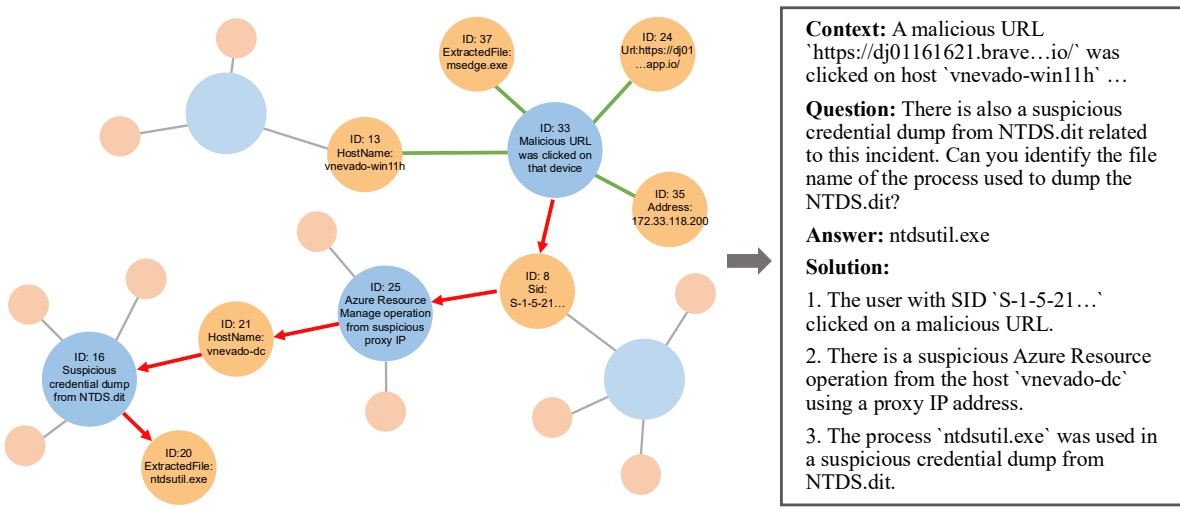

**Context:** A malicious URL `https://dj01161621.brave…io/` was clicked on host `vnevado-win11h` …

**Question:** There is also a suspicious credential dump from NTDS.dit related to this incident. Can you identify the file name of the process used to dump the NTDS.dit?

**Answer:** ntdsutil.exe

**Solution:**

1. The user with SID `S-1-5-21…` clicked on a malicious URL.

2. There is a suspicious Azure Resource operation from the host `vnevado-dc` using a proxy IP address.

3. The process `ntdsutil.exe` was used in a suspicious credential dump from NTDS.dit.

*Figure 3.* Example Question Generation. The start alert and entities will be used as context, and the question asks about the end entity (ID 20). A solution path is also based on the graph.

---

**Algorithm 1** Question–Answer–Solution Generation

---

**Require:** Bipartite Graph $G = (U, V, E)$, start entity count $k$, LLM $M$, question prompt $P_g$, solution prompt $P_s$, Set of question–answer–solution triples $Q$

1: $Q \leftarrow \emptyset$
2: **for all** distinct pairs $(u_s, u_t) \in U \times U$ **do**
3:      # Get start entities
4:      $V_s \leftarrow \text{SAMPLE}(\text{GETFARTHESTENTITIES}(u_s, u_t), k)$
5:      # Get answer
6:      $v_e \leftarrow \text{SAMPLE}(\text{GETFARTHESTENTITIES}(u_t, u_s), 1)$
7:      # Generate question and answer
8:      $(q, a) \leftarrow M(P_g, u_s, V_s, u_t, v_e)$
9:      # Generate solution path
10:      $s \leftarrow M(P_s, q, \text{SHORTESTPATH}(u_s, v_e))$
11:      $Q \leftarrow Q \cup \{(q, a, s)\}$
12: **end for**
13: **return** $Q$

---

optimal solution, but we note that there can be many different paths reaching the ending alert (some might not even be in the graph). We note that the entities in the path can be viewed as IoCs, since they are used to discover new events (alerts) and gather information. We pass the path to LLM to generate a step-by-step solution. The proposed QA generation strategy is a principled way to generate security questions from a bipartite alert-entity graph, and has the following benefits: 1. The questions are non-repetitive, and they test LLMs' abilities on querying the database to perform investigation. 2. We can obtain a clear answer and a solution path, which allows us to have a fine-grained and accurate evaluation. 3. The length of the shortest path can also be used as a measure of the difficulty of the question, so a longer path indicates a harder question. Based on this,

we can acquire a total of 7542 questions from the generated graphs, and we generate a total of 589 questions for testing, which is used in experiments, see Appendix D for full details of question logistics and train/test split.

### 3.3. Environment Setup

---

**Algorithm 2** Progress Reward Calculation

---

**Require:** $\mathcal{S}, a_{\text{sub}}, b_{\text{step}}$      ▷ $b_{\text{step}}$: enable step checking

1: **if** is_answer_correct($\mathcal{S}_{|\mathcal{S}|}, a_{\text{sub}}$) **then return** 1
2: **end if**
3: **if** $\neg b_{\text{step}}$ **then return** 0
4: **end if**
5: $r \leftarrow 0, \ \gamma \leftarrow 0.4, \ d \leftarrow 1$
6: **for** $i \leftarrow |\mathcal{S}| - 1$ **down to** 1 **do**
7:      $d \leftarrow d\gamma$
8:      **if** check_step($\mathcal{S}_i, a_{\text{sub}}$) **then**
9:          $r \leftarrow r + d$
10:      **end if**
11: **end for**
12: **return** $r$

---

To conduct a security investigation, the agent needs to interact with a given database. We set up a MYSQL docker environment following (Yang et al., 2023) to execute queries. In the environment, the agent can choose to output a query to be executed or submit the answer. The solving process ends when the agent submits the answer or a maximum number of steps is reached. We set the maximum number of entries and the maximum character length that can be returned to avoid context overflow from queries. Our evaluation consists of two steps (See Algorithm 2): 1. match the submitted answer $a_{sub}$ with the ground-truth. 2. If not correct, we check if the submitted answer contains any intermediate step solutions,

| | | | | Incident Number | | | | | Avg |
|---|---|---|---|---|---|---|---|---|---|
| | 5 | 34 | 38 | 39 | 55 | 134 | 166 | 322 | reward |
| GPT-4o | 0.338 | 0.293 | 0.364 | 0.273 | 0.249 | 0.491 | 0.166 | 0.315 | 0.293 |
| GPT-4o-mini | 0.163 | 0.195 | 0.273 | 0.185 | 0.174 | 0.228 | 0.163 | 0.276 | 0.192 |
| o1-mini[†] | 0.147 | 0.244 | 0.091 | 0.230 | 0.160 | 0.333 | 0.189 | 0.382 | 0.222 |
| Phi-4-14B | 0.086 | 0.037 | 0.182 | 0.082 | 0.066 | 0.130 | 0.085 | 0.125 | 0.085 |
| Llama4-17b-Mav | 0.259 | 0.302 | 0.545 | 0.324 | 0.216 | 0.421 | 0.189 | 0.371 | 0.290 |
| Llama4-17b-Scout | 0.216 | 0.285 | 0.182 | 0.228 | 0.220 | 0.453 | 0.193 | 0.367 | 0.262 |
| GPT-4.1 | 0.356 | 0.315 | 0.364 | 0.295 | 0.258 | 0.474 | 0.292 | 0.489 | 0.338 |
| GPT-4.1-mini | 0.324 | 0.210 | 0.182 | 0.248 | 0.248 | 0.333 | 0.216 | 0.387 | 0.271 |
| GPT-4.1-nano | 0.164 | 0.185 | 0.091 | 0.118 | 0.136 | 0.077 | 0.097 | 0.179 | 0.136 |
| o3-mini[†] | 0.350 | 0.293 | 0.273 | 0.257 | 0.227 | 0.404 | 0.253 | 0.360 | 0.296 |
| o4-mini[†] | 0.312 | 0.383 | 0.545 | 0.362 | 0.284 | 0.568 | 0.269 | 0.517 | 0.368 |
| Gemini 2.5 Flash | 0.312 | 0.329 | 0.364 | 0.248 | 0.224 | 0.491 | 0.260 | 0.375 | 0.305 |
| Qwen-3-32b | 0.191 | 0.229 | 0.091 | 0.207 | 0.116 | 0.2 | 0.133 | 0.25 | 0.182 |
| o3 | 0.433 | 0.359 | 0.727 | 0.443 | 0.459 | 0.632 | 0.379 | 0.543 | 0.456 |
| Grok-4 | 0.306 | 0.424 | 0.455 | 0.32 | 0.248 | 0.509 | 0.28 | 0.421 | 0.344 |
| Qwen-3-235b-thinking | 0.289 | 0.38 | 0.455 | 0.269 | 0.194 | 0.428 | 0.214 | 0.436 | 0.302 |
| Claude-Sonnet-4.5 | 0.485 | 0.522 | 0.673 | 0.481 | 0.418 | 0.632 | 0.322 | 0.643 | 0.487 |
| GPT-5.1 (Reasoning=High) | 0.53 | 0.639 | 0.727 | **0.506** | **0.521** | 0.779 | **0.568** | 0.625 | 0.582 |
| Claude-Opus-4.5 | **0.567** | **0.754** | **0.764** | 0.491 | 0.514 | **0.825** | 0.545 | **0.664** | **0.606** |

*Table 2.* Evaluation of the baseline agent with different models. The average per incident and the total average reward are shown (Total average reward = sum of reward / total question count). The model is sorted by release date (first being the earliest). [†] indicates the agent uses enhanced format prompt, since reasoning model like `o1-mini` struggles with the output format.

which shows how much progress the agent makes and useful information it acquires. We assign a decayed reward starting from the last step. This reward not only checks the final answer but also performs a fine-grained evaluation of the agent's intermediate steps. This also evaluates the agent's ability to identify and extract key information (IoCs) from its investigation process. For both `is_answer_correct` and `check_step`, either deterministic string matching or LLM-as-judge is allowed. However, we note that a well written answer template needs to be defined, in order to extract the answers for string matching. Thus, the environment can provide reliable rewards for intermediate steps, allowing future work on training agents with RL.

## 4. Experiments

### 4.1. Base Models Comparison

**Setup.** We first compare the performances of different LLMs in Table 2 using a base prompt (See Figure 9). We set temperature to 0 and max step to 25. We choose to use `GPT-4o` for answer evaluation since we notice that sometimes the model is not following the answer template. We also manually checked the evaluation results to confirm that LLM-as-judge can be applied here (See Appendix C.5).

**Result Analysis.** We observe the following: **(1)** `GPT-5.1` and `Claude-Opus-4.5` are the two best models on the dataset, surpassing the next-best `Claude-Sonnet-4.5` by 0.1. **(2)** `Phi-14B` barely solves any tasks, and smaller open-source models such as `Qwen-2.5-7B` and `Llama-3-8B` behave similarly, so we omit them. In contrast, `Llama4-Mav-17B` reaches 0.29, making it competitive with proprietary chat models like `GPT-4o` and `Gemini 2.5 Flash`. **(3)** Incident 39 has the lowest max score (0.506) while incident 134 has the highest score (0.825). **Takeaways:** Recent models are achieving higher rewards overall, open-source models are rapidly closing the gap with proprietary ones, and explicit-reasoning models are improving quickly. Since our benchmark is new and absent from any publicly available training data, test-set leakage is unlikely, showing genuine progress in LLM capabilities.

**Behavior Analysis.** Agents must first explore and infer the structure of the tables since the schema is not provided. To arrive at the correct answer, they have to combine information from several tables. Figure 4 shows a representative trajectory for the baseline agent: it gradually discovers the schema and refines its SQL queries whenever an error or empty result occurs. Like a human analyst, the agent bootstraps on intermediate findings to steer subsequent exploration. In the reference (gold) solution, the answer is

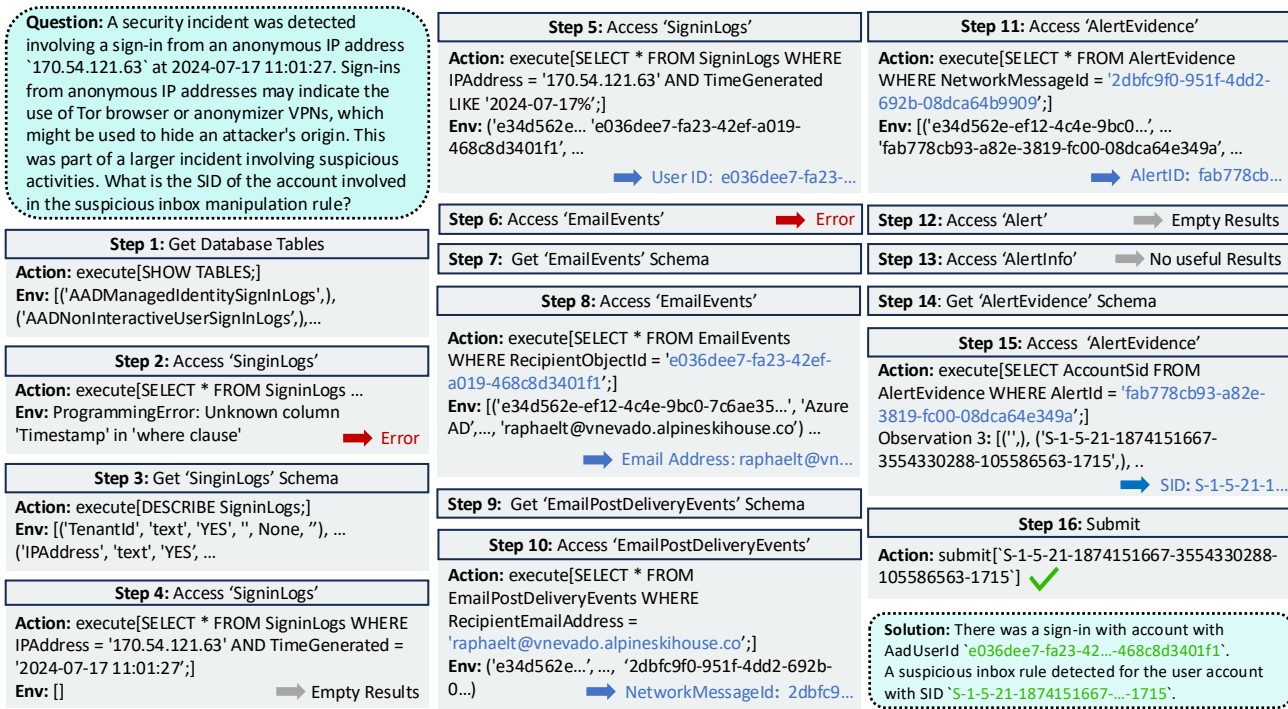

**Figure 4.** An example trajectory of Baseline Agent (with `GPT-4o`) solving a question. The agent goes through several steps to reaching the answer: After getting the user id from 'SignInLogs', it starts exploring two different email-related logs to get a network message id, and finally use it to find the SID. Since the agent also explores schema of tables as it progresses. Full example in Figure 13.

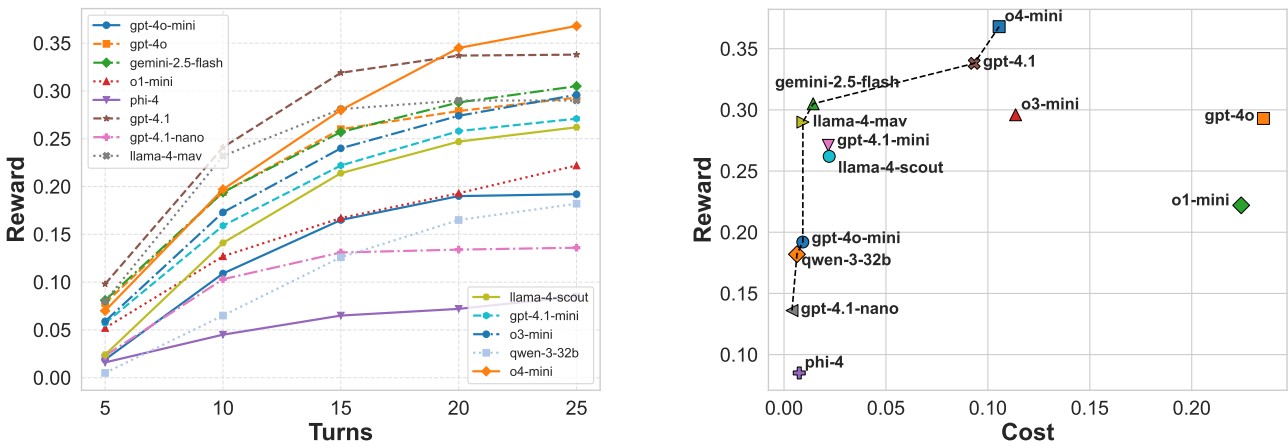

**Figure 5.** (a) Reward vs. Number of Turns. (b) Reward vs. Cost.

produced in two concise steps: (1) retrieve the user ID, then (2) use that ID to obtain the account SID. The agent pursues a longer, alternative route yet still converges on the correct answer, demonstrating that our benchmark supports multiple viable search strategies. The main difficulty remains in modelling the database accurately enough to construct valid queries. Across models, the Pearson correlation between query success rate and reward is 0.86, indicating a strong positive association.

**Efficiency Analysis. Turns:** In Figure 5a, we plot the change in reward by increasing max turns allowed to interact with the database (with a few selected models). We can see the reward spikes from 5 to 15 turns, then plateaus between 15 and 25. `o4-mini` scales well with increased turns, from around 0.07 to 0.37 at 25 turns. In comparison, chat models like `GPT-4o`, `Gemini-2.5-flash` start at around the same reward at 5 turns, but can only reach 0.3 with the turns allowed increased to 25. The

| k | Agent | GPT-4o | | | GPT-4o-mini | | | o3-mini | | |
|---|-------|--------|--|--|-------------|--|--|---------|--|--|
| | | reward | turn | cost | reward | turn | cost | reward | turn | cost |
| 1 | Base | 0.26 | 11 | 0.24 | 0.165 | 11 | 0.009 | 0.219 | 11 | **0.073** |
| | Strategy | 0.273 | 11 | **0.18** | 0.290 | 12 | 0.010 | 0.259 | 12 | 0.077 |
| | ReAct | 0.354 | 11 | 0.24 | 0.274 | 10 | 0.016 | 0.25 | 11 | 0.075 |
| | Expel | **0.390** | **9** | 0.38 | **0.311** | **9** | 0.023 | **0.265** | **9** | 0.191 |
| 3 | Strategy+BoN | 0.473 | 27 | **0.37** | 0.418 | 27 | **0.028** | 0.382 | 29.6 | 0.192 |
| | Strategy+Reflect | 0.505 | 25 | 0.47 | 0.440 | 26 | **0.028** | 0.394 | **26.1** | **0.172** |
| | ReAct + BoN | **0.563** | 21 | 0.49 | 0.423 | 24 | 0.036 | 0.378 | 28.6 | 0.197 |
| | ReAct+Reflect | **0.563** | **21** | 0.50 | **0.452** | **24** | 0.035 | **0.414** | 28.2 | 0.19 |

*Table 3.* Evaluation results across methods. Reward, turn, and cost are reported, grouped by number of trials.

comparison of `GPT-4o`, `GPT-4.1` with `GPT-4o-mini`, `GPT-4.1-nano` shows that smaller models have less gain with more increase. **Cost:** In Figure 5b, we plot the reward versus cost for the selected models and draw the Pareto front line. `Gemini-2.5-flash` and `Llama-4-Mav` are the most efficient models, with a competitive reward of around 0.3 while keeps its cost low. `GPT-4o` and `o1-mini` are the least efficient, which is expected since prices fall with each new release and they're the oldest models.

See Appendix E.2 for more details of the setup, additional analysis on path length and rewards, factors linked to high reward. A preliminary fine-tuning result on `GPT-4o` is shown in Appendix E.3. We also do a Best-of-N experiment to investigate the scaling bounds in Appendix E.4.

### 4.2. Comparison of different methods

**Setup.** To disentangle how different methods (e.g, prompting, test-time scaling) influence agent behavior, we evaluate six different configurations (Prompts in Figure 9 to 12). **(1) Base Prompt**. **(2) Strategy** adds additional notes on using the alert tables as a reference to investigate, which is usually how a SOC analysts find information. **(3) ReAct** We follow ReAct (Yao et al., 2022) to add 3 few-shot examples to base, which are selected from runs of the train set. **(4) Expel** (Zhao et al., 2024) first distills rules from a small training split, then invokes this external memory, as well as retrieving similar examples during inference. **(5) Best-of-N** retries at most three times and returns the highest-reward trajectory. This is an oracle setting in which the agent needs the reward to determine if a problem needs to be rerun. **(6) Reflection** (Shinn et al., 2023) extends Best-of-$N$ by letting the agent criticize its failed attempt, append the learned rule to the prompt, and retry. We run all methods with max turn set to 15 on `GPT-4o`, `GPT-4o-mini`, and `o3-mini` and report average reward, interaction turns, and cost in Table 3.

**Results.** For a *single* trial ($k = 1$), Expel attains the best reward among all models, while also finishing in the fewest

turns (9). Its performance comes at a higher price: Expel costs 1.6x more than Strategy on `GPT-4o` because the learned knowledge block and retrieved examples inflate the prompt. `o3-mini` can hardly benefit from methods like ReAct and Expel, with even a slight drop in accuracy when switching to ReAct, but `GPT-4o` and `GPT-4o-mini` have significant gains when switching from base to ReAct and Expel (+ around 0.1). For $k = 3$, we only test and compare Strategy and React prompting due to cost constraints. ReAct+Reflect achieves the best among the models, and we can see that Reflect can almost always help with the performance with different models and prompting strategies. We note that no improvement is made switching from ReAct+BoN to ReAct+Reflect, tested with `GPT-4o`, which may indicate that this setting has reached an upper bound.

### 4.3. Ablation on DB Scope and Time Window (Figure 4)

| DB Scope | Time Window | Reward |
|----------|-------------|--------|
| raw + alert | Per incident | 0.260 |
| raw | Per incident | 0.213 |
| alert | Per incident | 0.459 |
| raw + alert | Full history | 0.248 |
| raw | Full history | 0.184 |
| alert | Full history | 0.382 |

*Table 4.* Ablation on database setup.

By default, one database is set up per incident with both raw and alert logs. **DB Scope:** In the real world, zero-day and sophisticated attacks may evade security detections and not be shown in alert logs. We remove alert logs to simulate this and observe an obvious drop in performance. This indicates that these alert logs created with rules, heuristics, and ML-based detections are crucial for investigation. We also set up an alert-only database for comparison and found a substantial increase in reward. This is expected since our questions are built from security tables. This also shows that unrelated noise from the database can impact performance.

**Time Window:** We also test with a full version of the database (explained in Section 3.1). Moving from the per-incident slices to the full database lowers average reward to 0.248, which is expected since a longer horizon introduces extra noise. We note that degradation from using a longer time span is mild compared to switching the DB scope. Since the questions constructed by LLMs tends to include time information in the question, which relieves the effect of a noisier environment (Tested with GPT-4o).

## 5. Conclusion

In this paper, we create ExCyTIn-Bench, the first benchmark to evaluate LLM agents on cybersecurity threat investigations based on real-world setup. It includes an open-source Azure security database, a QA dataset, and a standardized environment. We also introduce an automated, structured approach that leverages LLMs to generate high-quality questions from bipartite alert–entity graphs, enabling fine-grained evaluation of an agent's intermediate steps. We evaluate various LLMs and agent systems on the benchmark. Open-source models are catching up with propriety models in our environment, and they can be used for further distillation and training to boost performance. Our environment offers fine-grained process rewards that allow precise credit assignment across steps, which is an uncommon but valuable feature for our environment. Overall, this makes it a promising testbed for training LLM agents with reinforcement learning (Zhao et al., 2025).

## Impact Statement

By open-sourcing both the benchmark and its underlying environment, ExCyTIn-Bench aims to accelerate research on autonomous LLM agents capable of navigating complex, multi-step security analyses. A shared, reproducible evaluation platform can foster more rapid iteration on prompting strategies, model architectures, and reinforcement-learning techniques tailored to cyber-defense tasks. In the long run, progress driven by this benchmark could lead to more effective automated triage and incident response, lowering the barrier to advanced threat hunting for under-resourced organizations. At the same time, publicly available benchmarks must be balanced against the risk that malicious actors might study agent behaviors to identify weaknesses or craft evasion techniques, underscoring the need for responsible disclosure and continued collaboration between defenders and the research community.

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

## A. LLM Usage

In this paper, we have minor usage of LLMs to help polish writing, which includes the following: 1. check for grammar. 2. Shorten and refine sentences. 3. Provide word choices.

## B. Background on Cybersecurity

Cybersecurity threat investigation systematically probes digital environments to detect, analyze, and mitigate malicious activity (IBM, 2023; CrowdStrike, 2023). Threat analysts serve as cyber-detectives: they parse vast logs, link evidence from diverse sources, and judge potential threats. Since modern infrastructures produce vast volumes of data daily, a central challenge is efficiently identifying and correlating threat signals, with other key skills required (e.g., *evidence gathering and synthesis)*. Our benchmark is designed to assess LLM agents on all of these capabilities.

**Threat Investigation Graphs.** Incident graphs portray multi-stage attacks by linking alerts, events, and indicators of compromise (IoCs) into a unified view. Nodes denote alerts (e.g, suspicious file downloads) or entities (e.g, user accounts) while edges capture their relationships (e.g., a phishing email that triggers a malicious download). Sequencing nodes along the kill chain (reconnaissance, intrusion, persistence, etc.) exposes adversary tactics, surfaces patterns, and clarifies next steps for responders. In our benchmark, we utilize these threat investigation (or incident graphs) for grounding LLMs during question generation and evaluation.

Below are key tasks and skills that cybersecurity threat analysts typically leverage in their day-to-day work:

- Understanding Logs and Data Sources: Familiarity with the wide range of log formats (e.g., system event logs, network traffic logs, web server logs) and how each source can reveal indicators of compromise.

- Triage and Data Analysis: Rapidly filtering large datasets to identify potential leads, such as unusual account behavior or abnormal traffic spikes.

- Querying and Coding Skills: Ability to write SQL or other specialized query languages to extract the exact data needed, enabling deeper inspection of suspicious events or user activity.

- Evidence Correlation and Synthesis: Combining data from multiple sources—such as SIEM (Security Information and Event Management) alerts, intrusion detection systems, endpoint security suites, and threat intelligence feeds—to construct a complete picture of the incident. This also involves recognizing patterns and drawing relationships between events, timestamps, and potential attackers' tactics, techniques, and procedures (TTPs).

- Hypothesis Testing: Formulating and testing possible explanations for alert signals—for instance, whether peculiar activity might stem from a misconfiguration or a targeted attack. Iterating through multiple hypotheses, gathering more evidence until one scenario best explains the observed behaviors.

- Noise Filtering: Distinguishing benign anomalies (e.g., legitimate system updates, authorized organization-wide password resets) from malicious behaviors. Employing data normalization and enrichment techniques to reduce extraneous signals and highlight true threats.

- Leveraging Cyber-security Domain Expertise: Applying deep knowledge of security frameworks and best practices (e.g., MITRE ATT&CK (The MITRE Corporation, 2025b), CVE (Common Vulnerabilities and Exposures) (The MITRE Corporation, 2025a), OSINT (Open Source Intelligence) (Nordine, 2024), etc.) to guide investigation processes and validate findings. Drawing on historical context about common threat actor tactics and industry-specific threats to anticipate potential entry points or attack vectors.

## C. Benchmark Details

### C.1. Features

*Real-world security database.* Although the tenant is for a fictional company, the log data we use is from real-world tenant environments. Our database consists of security log data, and the volume of the database is much bigger than previous works (Yang et al., 2023; Yu et al., 2018). We further note that it is extremely challenging to acquire actual client data for research, and it is nearly impossible to opensource such data. The dataset we are releasing is rare: a tenant built for a fictional company with many properties similar to real-world customer tenants.

*Real-world attacks.* The attacks are complete replications of real-world kill chains used against Azure clients. All of them actually occurred and are well documented in cybersecurity news and blogs. For example, Incident 5 is a simulation of "Manatee Tempest Ransomware" (2021), and Incident 134 covers the "BEC and Account Take-over" (2024) attack.

*Require multi-hop data exploration and analysis.* Our questions are constructed so that the agent has to interact with the database to reach the final answer, utilizing new information gained to continue the investigation.

*Domain Specific.* Our benchmark is built on security log data. So it requires the agent to have a strong knowledge in the security domain to understand the data and conduct meaningful reasoning.

*Fine-grained Rewards.* The decayed reward calculation adopted from RL allows us to evaluate agents' performances

| Benchmark / Setting | Avg. tables | Avg. rows (K) | Avg. DB size (GB) |
|---|---|---|---|
| Spider (avg. / DB) | 3.5 | 8.6 | 0.03 |
| Ours (avg. / incident, all 8) | 45 | 364.7 | 0.447 |
| Ours (avg. / incident, excluding incident_55) | 45 | 115.4 | 0.182 |
| Ours (full 44-day log) | 61 | 14,299.5 | 18.0 |

*Table 5.* Per-database statistics of Spider (Yu et al., 2018) and our benchmark. Spider database statistics are taken from (Zhang et al., 2023a).

with greater granularity, instead of only success and failures. This metric also enables better evaluation of the intermediate steps.

*Can Transfer to other clouds.* Our question generation method only assumes: Incident/alert logs from which to build an alert–entity graph and a correlated raw-log store queryable by the agent. Both are standard across SIEMs: *Azure*: directly matches our released environment; other Azure tenants can instantiate the same pipeline privately. *Google Cloud*: Log Analytics for centralized logs + alert investigation graphs. *AWS*: Amazon Detective's behavior graph + Security Lake for centralized raw logs.

*Database Size and Complexity* To more clearly position our benchmark relative to prior work, Table 5 reports dataset-scale statistics for Spider (Yu et al., 2018) and our benchmark. We use Spider as the primary point of comparison because it is a classic text-to-SQL benchmark; InterCode-SQL (Yang et al., 2023) is constructed from Spider examples rather than serving as an independent SQL benchmark, so we report the corresponding Spider database statistics, which are taken from (Zhang et al., 2023a). These per-database statistics reflect the amount of data available for answering a single question: each Spider question is associated with one database, whereas each question in our benchmark is associated with one incident database. Compared with Spider, our benchmark is substantially larger on a per-database basis, averaging 45 tables, 364.7K rows, and 0.447 GB per incident across all eight incidents. Because one incident, incident_55, is unusually large, we also report a robust comparison that excludes this outlier. Even under this setting, our benchmark averages 45 tables, 115.4K rows, and 0.182 GB per incident, which remains considerably larger than Spider. Beyond the per-incident setting, the full 44-day log available for investigation contains 61 tables, 14.3M rows, and 18 GB of data. These statistics help distinguish dataset scale from reasoning complexity and empirical difficulty in our evaluation.

### C.2. Database Setup

Our questions are constructed from alert and incident tables. In a common scenario, the security analysts will be given these tables to serve as starting points to conduct analysis. This is also the current setup for conducting experiments.

However, there may be new attacks that analytic rules cannot detect and summarize them into security alerts. This requires the security analysts to analyze and find IoCs from the rest of the logs. We also support the setting to remove the security logs from the database to simulate this scenario.

### C.3. Personally Identifiable Information (PII) Anonymization

Below we explain how we do PII Anonymization on our dataset through a joint effort of manual and LLM-based examination.

**1. Identification of PII Columns** Each table is scanned column-by-column. For every column we draw a random sample of five values and prompt a (LLM) to decide whether the column contains PII. Columns provisionally flagged in this first pass are examined once more with three focused prompts: 1. Confirm whether the column indeed holds PII. 2. Decide whether the column stores a dictionary/`JSON` structure. 3. If it does, enumerate which keys inside the structure contain PII.

The union of both LLM passes is then reviewed by domain experts, yielding a curated list of PII-bearing columns that serves as ground truth for the remainder of the pipeline.

**2. Creation of PII Value Mappings** For every confirmed PII column we gather its set of unique values. If the column encodes a dictionary, only the keys identified in the previous stage are considered.

- **Regex-based substitution.** We manually go through the tables to recognize common PII patterns, and each candidate value is matched against them (IPv4/IPv6 addresses, e-mail addresses, UUIDs, MAC addresses, latitude/longitude pairs, *etc.*). Matches are replaced by randomly generated surrogates that obey the same syntax.

- **LLM-based substitution.** Values that do not match any pattern are batched (ten values per batch) and passed to the LLM, which returns semantically plausible yet fictitious substitutes (e.g. "John" → "Javier").

All substitutions are cached in a dictionary so that a source value is always mapped to the same surrogate. The resulting

|  | Total questions | Submitted questions | TP | FP | TN | FN |
|---|---|---|---|---|---|---|
| Combined | 163 | 132 | 56 | 0 | 76 | 0 |

*Table 6.* Confusion matrix of result of manually going through 163 evaluation results graded by LLMs. Surprisingly, we observed no disagreements in this manually reviewed sample.

mappings are classified into coarse categories (`ip`, `email`, `other`) and briefly inspected to remove spurious or already anonymised tokens. Empirically, IP addresses account for roughly 95% of all distinct PII values encountered.

**3. Dataset-wide Replacement** In the final stage we stream every table in the dataset, globally replacing each source PII value with its surrogate. This guarantees *referential consistency*, which are queries that join on an anonymised IP address still succeed, and eliminates residual PII leakage while preserving analytical utility.

### C.4. Full Log Table Names

The 57 tables span identity, endpoint, email, cloud resource, and network-centric telemetry typical for Azure-based SOC operations. Concretely, this includes:

- **Identity & sign-in:** SigninLogs, AADManagedIdentitySignInLogs, AADNonInteractiveUserSignInLogs, AADServicePrincipalSignInLogs, AADRiskyUsers, AADUserRiskEvents, IdentityDirectoryEvents, IdentityLogonEvents, IdentityQueryEvents, AuditLogs.

- **Endpoint & host activity:** DeviceProcessEvents, DeviceLogonEvents, DeviceNetworkEvents, DeviceFileEvents, DeviceImageLoadEvents, DeviceRegistryEvents, DeviceInfo, DeviceNetworkInfo.

- **Email & collaboration:** EmailEvents, EmailPostDeliveryEvents, EmailAttachmentInfo, EmailUrlInfo, UrlClickEvents, OfficeActivity, CloudAppEvents.

- **Network / cloud infra:** AZFWApplicationRule*, AZFWNetworkRule*, AZFWNatRule*, AZFWDnsQuery, AZFWThreatIntel, MicrosoftAzureBastionAuditLogs, AzureMetrics, ThreatIntelligenceIndicator, Usage, Heartbeat, SentinelHealth.

- **Security analytics layer:** Alert, AlertInfo, AlertEvidence, SecurityAlert, SecurityIncident.

### C.5. LLM as Judge

The main reason we use **GPT-4o as the judge** is that the agents we evaluated often fail to follow the required answer format. We observed the following issues when we relied on string matching:

- **The agent does not follow the answer format.** For ex-

ample, when the correct answer is `18.27.43.343`, the agent writes: "The final answer is 18.27.43.343" instead of providing only the IP address.

- **The agent returns the answer in an alternative form.** For example, if the correct answer is the user name `user1`, the agent might output an email containing the same string—`user1@alpineskihouse.co`—which should still be regarded as correct.

- **The agent offers multiple answers, including the correct one.** For example, when only one IP address is correct, the LLM may supply two IPs, one of which is correct. After discussion with security experts, they still think it is pretty helpful to narrow down the final answer to 2 IP addresses using a LLM assistant, and can be judged as "correct".

We believe string matching is a valid evaluation method and can easily be enabled in our system. However, because current LLMs already struggle with our benchmark, we adopt the *LLM-as-judge* approach to measure agents' ability to solve tasks rather than secondary abilities such as strict formatting. We design the judging prompt with techniques to minimize hallucination (task decomposition, provision of all necessary information) and with the self-reflection technique. We also include explicit rules to make the judgments more closely aligned with human heuristics.

To verify the effectiveness of using an LLM as the judge, we randomly selected 163 answered questions and manually reviewed them; surprisingly, all were evaluated correctly (See Figure 6). We note that this is only a subset of evaluations so we cannot just draw a conclusion that LLM-as-judge is perfect. Instead, it suggests that LLM-based evaluation can be a reasonable choice, while we still acknowledge that some errors are inevitable but practically tolerable. Also, a main reason that LLM evaluation is so accurate is that the agent is submitting a single string that can be judged easily. We also observe the following scenarios in which the LLM's judgment aligns with human judgment, please see Figure 6 for 3 examples.

## D. Question Generation Details

### D.1. Graphs and Reports

In Figure 17-24, we put overview plots of graphs for all incidents. Note that the graphs are for illustrative purposes

---

**Sample 1: Several wrong IPs are listed**

Ground Truth: 170.54.121.63
Submitted: The IP addresses associated with the suspicious login activity for Nina Park are '191.109.229.245', '36.193.21.220', and '237.7.81.122'.
LLM judgment: Incorrect
Analysis: The submitted answer does not contain the golden answer's IP address (170.54.121.63), which is crucial to identifying the suspicious login activity for Nina Park. None of the submitted IP addresses match the golden answer.

---

**Sample 2: Comparing "dp5hn" with dp5hn@vnevado.alpineskihouse.co**

Ground Truth: dp5hn
Submitted: The account name involved in the Command and Control behavior is "dp5hn@vnevado.alpineskihouse.co".
LLM judgment: Correct
Analysis: The submitted answer contains the golden answer "dp5hn" with additional context in the form of a domain, which still identifies the account accurately.

---

**Sample 3: One of two submitted answers matches the ground truth**

Ground Truth: S-1-5-21-1440151660-4837083288-105596163-1120
Submitted: The SID values associated with the affected accounts are 'S-1-5-21-1440151660-4837083288-105596163-1120' and 'S-1-5-21-1846150160-3535302852-1555536564-2111'.
LLM judgment: Correct
Analysis: The submitted answer includes the golden answer 'S-1-5-21-1440151660-4837083288-105596163-1120' along with an additional SID value. It is still relevant and within the context of the question.

---

*Figure 6.* Examples of LLM judgments matching human heuristics.

---

**Algorithm 3** GETFARTHESTENTITIES (From Algorithm 1)

---

**Require:** Graph $G = (V, E)$ where each node has attribute type, start alert $a_s \in V$, end alert $a_e \in V$
1: $S \leftarrow \{v \in \text{NEIGHBORS}(G, a_s) \mid \text{type}(v) = \text{"entity"}\}$ ▷ entities adjacent to $a_s$
2: $D \leftarrow$ empty map ▷ keys: path length, values: entity lists
3: **for all** $e \in S$ **do**
4:     $d \leftarrow \text{SHORTESTPATHLENGTH}(G, e, a_e)$
5:     $D[d] \leftarrow D[d] \cup \{e\}$ ▷ bucket entity by distance
6: **end for**
7: $d_{\max} \leftarrow \max\{d \mid d \in D\}$ ▷ greatest distance observed
8: **return** $D[d_{\max}]$ ▷ all entities farthest from $a_e$

---

and some details might be hard to read due to the large graph size. The bigger blue nodes represent alerts, and the smaller red nodes represent entities. In Figure 25- 39, we put summarized reports of these graphs using LLM.

### D.2. Question Logistics

Since the deterministic analytic rules are run at an interval, there are many repetitive alerts generated. We first process each incident to remove repetitive alerts. If there are disjoint graphs in an incident, we will keep the larger one. Since we can generate one question from any two alerts, the number of questions is bounded by the 2nd exponential of the number of alerts, where we can generate at most 7542 questions. However, a number of alerts for each incident is extremely unbalanced, resulting in several questions from 4624 to 16 questions. We also want to split the data into a training and test set, with a main focus on the diversity and quality of the test set. To this end, we use the following split strategy: if an incident has less than 150 questions, we will take around 70% of the questions as test data. If the number of questions is bigger than 150, we will cap the number of questions to 100. Under these criteria and filtering after question generation, we collected a total of 589 questions as the test set (See Figure 7). We also created a strategy for sampling questions to split the training and test. Since we are building questions from the graph, and the train and test sets are all from one graph, we want the train samples to have less overlap in paths with the tested sets. When performing training, the model might remember knowledge over the path information and could "cheat" from these. Thus, we create an overlap score and use it to guide the random sampling. We will random split k times and select the split with the highest overlap score.

| Incident | Path Length | | | | | Total |
|---|---|---|---|---|---|---|
| | 1 | 3 | 5 | 7 | 9 | |
| 38 | 4 | 7 | 0 | 0 | 0 | 11 |
| 34 | 9 | 73 | 0 | 0 | 0 | 82 |
| 5 | 3 | 74 | 15 | 6 | 0 | 98 |
| 39 | 4 | 57 | 34 | 3 | 0 | 98 |
| 134 | 7 | 50 | 0 | 0 | 0 | 57 |
| 322 | 5 | 19 | 23 | 9 | 0 | 56 |
| 166 | 11 | 76 | 0 | 0 | 0 | 87 |
| 55 | 3 | 57 | 26 | 13 | 1 | 100 |

*Table 7.* Number of questions generated from different path length for each incident.

### D.3. Split Train Test Set

We want to split the train and test set so that they have fewer overlaps. For example, with a simple graph $\{A - B - C, D - B - E\}$, it is best to split the paths in the subgraph $A - B - C$ and $D - B - E$ into two different sets, instead of $A - B$, $D - E$ to train, $A - C, D - E$ to test. For this purpose, we randomly split the training or test set and compute a customized total overlap score between every two paths from the train and test sets. We run this for T trials and select the split with the lowest overlap score.

**Overlap Score Calculation** Given two paths $P_1 = (v_0, \ldots, v_m)$ and $P_2 = (u_0, \ldots, u_n)$ in the same graph, convert each path to its ordered set of directed edges:

$$E_1 = \{(v_i, v_{i+1}) \mid 0 \le i < m\},$$

$$E_2 = \{(u_j, u_{j+1}) \mid 0 \le j < n\}.$$

Let

- $E_{\text{shared}} = E_1 \cap E_2$ — edges appearing in both paths;

- $E_{\text{unshared}} = E_1 \triangle E_2$ — edges that appear in exactly one path (the symmetric difference).

We reward overlap with a positive weight $\alpha > 0$ and penalize divergence with a cost factor $\beta > 0$, scaling the penalty by the combined edge count so the two terms are on comparable footing. To guarantee that the score is zero whenever the paths share no edge, we define the overlap score piecewise:

$$\rho = \frac{|E_{\text{unshared}}|}{|E_1| + |E_2|}.$$

$$S(P_1, P_2; \alpha, \beta) = \begin{cases} 0, & |E_{\text{shared}}| = 0, \\ \alpha \, |E_{\text{shared}}| - \beta \, \rho, & \text{otherwise.} \end{cases}$$
$$(1)$$

Thus, $S$ ranges from $-\beta$ (complete mismatch) to $\alpha$ (identical edge sets), and is exactly 0 when the two paths have no common edges at all.

## E. Additional Experiments and Details

### E.1. Note on Agent Setup

We didn't include the full schema directly in the prompt because it is way too large, and also following InterCode (Yang et al., 2023). As we show in figure 2, we have a total of 57 tables and each table is big to have different column names. With an approximate estimation, there are around 1700 column entities. Incorporate them and corresponding description would be an extreme long prompt. Even it is possible to include them in the prompt due to the long-context window of current LLMs, it would inefficient and thus an undesired setup for us. Thus, in our main evaluation, we didn't adapt this setup. However, we welcome others to test with such a setup.

We want to further note that the schema is fully discoverable in our evaluation setup, it is not that we are hiding the schema. Agents can issue queries to list tables or inspect table schemas using the log platform's meta-commands. And schema discovery is a meaningful skill, that analysts could start an investigation by run exploratory queries and browse documentation.

### E.2. On Experiment with Different Models

**Setup** For the first experiment to test different LLMs, we set `max_step` = 25, `max_num_entry` = 15, `max_char_len` = 100000. We use `GPT-4o` as our evaluator for all experiments. We set the temperature to 0 for all LLMs. For the rest of experiments, we maintain the same setting, except that we reduce the `max_step` to 15.

We test with the following models: (1) `GPT-4o`: (May 2024, OpenAI) multimodal model that reasons across audio, vision, and text; `GPT-4o-mini` (July 2024, OpenAI): distilled variant. (2) `o1-mini`: (Sept 2024, OpenAI) cost-efficient small reasoning model optimized for math and coding. (3) `Phi-4-14B`: (Dec 2024, Microsoft) 14 B-parameter model trained largely on synthetic data. (4) `Llama4-Maverick`: (Apr 2025, Meta) open-weight Mixture-of-Experts multimodal model with 128 experts; `Llama4-Scout` (Apr 2025, Meta): lighter variant with 16 experts. (5) `GPT-4.1`: (Apr 2025, OpenAI) successor to GPT-4o with stronger coding/instruction following; `GPT-4.1-mini` and `GPT-4.1-nano`: smaller, cheaper versions. (6) `o3-mini`: (Apr 2025, OpenAI) upgraded small reasoning model. (7) `o4-mini`: (Apr 2025, OpenAI) latest reasoning model by OpenAI. (8) `Gemini 2.5 Flash`: (Apr 2025, Google DeepMind) budget multimodal model that "thinks" before responding. (9) `Qwen3-32b`:

| | **Incident Number** | | | | | | | | **Avg** |
|---|---|---|---|---|---|---|---|---|---|
| | 5 | 34 | 38 | 39 | 55 | 134 | 166 | 322 | reward |
| GPT-4o | 0.338 | 0.293 | **0.364** | 0.273 | 0.249 | **0.491** | 0.166 | 0.315 | 0.293 |
| o1-mini[†] | 0.147 | 0.244 | 0.091 | 0.230 | 0.160 | 0.333 | 0.189 | 0.382 | 0.222 |
| o3-mini[†] | 0.350 | 0.293 | 0.273 | 0.257 | 0.227 | 0.404 | **0.253** | 0.360 | 0.296 |
| MM o1-mini[*] | 0.304 | 0.256 | 0.273 | 0.238 | 0.296 | 0.316 | 0.211 | 0.379 | 0.279 |
| MM o1[*] | 0.398 | **0.317** | 0.091 | 0.265 | **0.297** | 0.474 | 0.228 | **0.391** | **0.323** |
| MM o3-mini[*] | **0.404** | 0.310 | **0.364** | **0.274** | 0.264 | 0.333 | 0.218 | 0.375 | 0.308 |
| Finetune GPT-4o | 0.345 | *0.241* | *0.091* | 0.262 | 0.299 | 0.418 | 0.246 | 0.355 | 0.298 |

*Table 8.* **Results with Master-Slave testing and finetuning.** Models related to the MM testing is also included for comparison. [*] denotes a special mixture of model use (MM).The average per incident and the total average reward are shown (Total average reward = sum of reward / total question count). We only should the related models in the table (Full result in Table 2). [*] indicates the agent is instructed with additional format notes. [†] indicates the agent is instructed with additional format notes. For finetune (Appendix E.3), incident 34 and 38 are the hold-out (test) incidents, and the performance drops significantly.

(Apr 2025, Alibaba) Open-source model with hybrid reasoning capabilities. (10) `Grok-4`: (Jul 2025, xAI) Frontier general-purpose model with native tool use and real-time search integration. (11) `Qwen3-235B-thinking`: (Jul 2025, Alibaba) Open-weights Qwen3 "thinking" (reasoning) MoE variant aimed at stronger deliberative performance. (12) `Claude-Sonnet-4.5`: (Sep 2025, Anthropic) General-purpose frontier model positioned for coding, agentic workflows, and computer-use tasks. (13) `GPT-5.1 (Reasoning=High)`: (Nov 2025, OpenAI) GPT-5.1 family; "Reasoning=High" denotes a higher-deliberation runtime setting rather than a separate model release. (14) `Claude-Opus-4.5`: (Nov 2025, Anthropic) Highest-capability Claude model in the 4.5 line, targeted at complex coding, agents, and long-horizon work.

If the agent doesn't submit answer before reaching the max step, we take it as failure (reward = 0). We use Azure services for OpenAI models and AI Foundry for Phi-4. We use Google service for Gemini. For other open-sourced models `Llama-4-Maverick`, `Llama-4-Scout`, `Qwen-3-32b`, we use cloud service DeepInfra.

**Master-Slave Testing With Base Agent** (See Table 8) Since the o1 model is very expensive and time-consuming, we set up a special master-slave model switching for it (displayed as MM o1), which will use GPT-4o for 4 steps and switch to o1 every 5th step. For comparison, we also test this setting with o1-mini and o3-mini. The "master–slave" interleaving strategy boosts performance over a single reasoning model, with larger gains when the master is stronger.

**Additional Behavior Analysis** In Figure 7, we plot reward versus different rates for selected models to help understand the behavior of baseline agents with different models.

| Path Len | # Questions | Avg. Reward |
|---|---|---|
| 1 | 46 | 0.347 |
| 3 | 413 | 0.249 |
| 5 | 98 | 0.237 |
| 7 | 31 | 0.328 |

*Table 9.* Counts and average rewards by path length.

From Figure 7b and 7c, we find there is a strong relation between reward and how well the agent can give queries. From Figure 7a, we find that models that are more likely to get higher rewards with higher empty results. A higher empty rate can indicate that the agent has better fundamental capabilities to understand the table schema to give correct queries. However, it may still struggle to get meaningful results even if it can understand the query well. In Figure 7d, we plot submit rate versus reward. Achieving a lower accuracy with a high submit rate indicates that the agent cannot access their progress correctly, and is overconfident in submitting their results. Smaller models like `gpt-4o-mini` and `gpt-4.1-nano` tend to submit their answer more often, but stronger models like `o4-mini` achieve high results without a relatively low submit rate.

**Results of questions generated from different length (Figure 9).** Our questions are generated from a different path. We count the questions generated from different path lengths, and averaged the model rewards in Table 2. From the question generated from path length 1-5, the reward is decreasing as expected. However, the reward suddenly rises when length is 7 (Since there is only 1 question generated from path length 9, we didn't show it here). We hypothesize that while we can have an explicit representation of the hops based on our constructed graph, this might not be a full view. As the hop grows larger, there may be another easier and

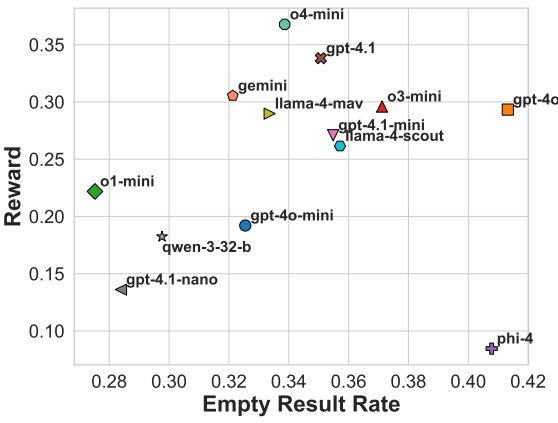

*(a)* Reward vs. Empty Result Rate (Number of queries that return nothing over total number of queries).

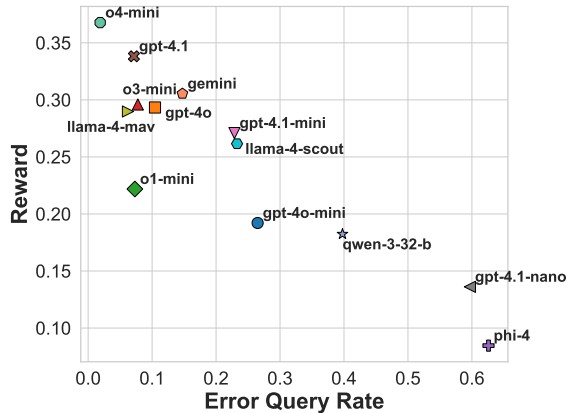

*(b)* Reward vs. Error Query Rate

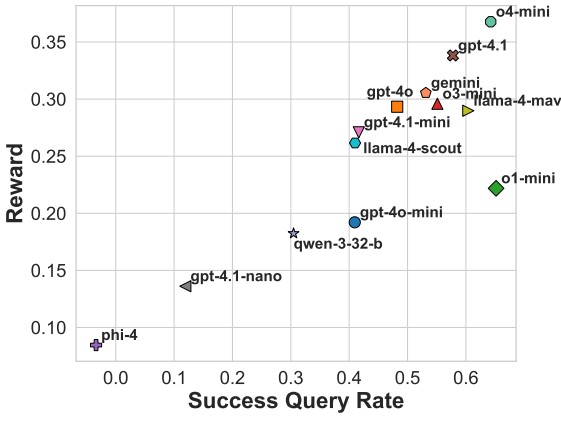

*(c)* Reward vs. Success Query Rate (successful queries that get non-empty returns / total queries) for each model.

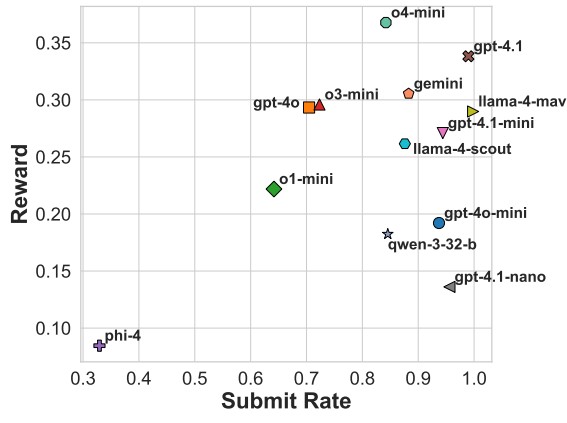

*(d)* Reward vs. Submit Rate (Number of questions submitted / Total number of questions).

*Figure 7.* Reward versus different query performance metrics and submit rate for each model.

unknown path that is not shown in our graph. Also note that there are fewer data points for path 7, so this result might be biased.

### E.3. Fine-Tuning

We conducted preliminary fine-tuning experiments on GPT-4o variants to assess whether it could improve accuracy (See Table 8). From our logs, we extracted successful trajectories produced by GPT-4o, GPT-4o-mini, o1-mini, and o3-mini, withholding those from incidents 28 and 34 as a held-out test set. The remaining 253 trajectories were used to fine-tune each model via Azure Training Service, training only on the assistant's responses. Although overall accuracy on the training incidents remained essentially unchanged, performance on the hold-out incidents suffered markedly: accuracy dropped from 0.293 to 0.241 on incident 34 and from 0.364 to 0.091 on incident 38. This suggests that fine-tuning amplified the model's bias toward the training incidents, degrading its ability to generalize. Given our

small sample size, additional studies are needed to characterize the impact of fine-tuning more precisely. However, these initial results imply that naïve fine-tuning may be ill-suited to this task and motivate exploring alternatives, such as reinforcement learning with value regularization (RLVR), to bolster performance on unseen incidents.

### E.4. Additional Experiment on Scaling

To further explore the limits of our test-time scaling method, we apply Best-of-N sampling to the baseline GPT-4o agent over 10 independent trials (see Figure 8). Across these trials, the mean reward rose from approximately 0.29 to 0.62. Although the reward has not fully converged by trial 10, the rate of improvement diminishes, indicating a flattening slope. Meanwhile, the average number of rounds continues to grow steadily. This follows that all our test-time in-context learning methods perform worse than baseline models in pass@k at high k values. A notion very similarly observed for RLVR approaches in (Yue et al., 2025).

| Agent | 5 | 34 | 38 | 39 | 55 | 134 | 166 | 322 | Avg. |
|---|---|---|---|---|---|---|---|---|---|
| Base | 0.293 | 0.268 | 0.364 | 0.242 | 0.245 | 0.421 | 0.131 | 0.268 | 0.260 |
| Strategy | 0.263 | 0.371 | 0.273 | 0.239 | 0.201 | 0.277 | 0.280 | 0.325 | 0.273 |
| ReAct | 0.365 | 0.383 | 0.091 | 0.308 | 0.324 | 0.526 | 0.264 | 0.446 | 0.354 |
| Expel | 0.461 | 0.395 | 0.309 | 0.363 | 0.324 | 0.456 | 0.372 | 0.400 | 0.390 |
| Strategy+BoN | 0.457 | 0.515 | 0.364 | 0.469 | 0.414 | 0.526 | 0.460 | 0.543 | 0.473 |
| Strategy+Reflect | 0.480 | 0.524 | 0.545 | **0.491** | 0.474 | 0.579 | 0.483 | 0.554 | 0.505 |
| ReAct+BoN | **0.614** | 0.554 | 0.455 | 0.473 | **0.554** | **0.737** | 0.517 | 0.574 | **0.563** |
| ReAct+Reflect | 0.581 | **0.573** | **0.818** | 0.449 | 0.530 | **0.737** | **0.538** | **0.585** | **0.563** |

*Table 10.* Per-incident performance of different agents on GPT-4o. Best scores for each incident and the average are bolded.

| Agent | 5 | 34 | 38 | 39 | 55 | 134 | 166 | 322 | Avg. |
|---|---|---|---|---|---|---|---|---|---|
| Base | 0.128 | 0.183 | 0.273 | 0.154 | 0.154 | 0.211 | 0.113 | 0.257 | 0.165 |
| Strategy | 0.324 | 0.334 | 0.455 | 0.298 | 0.224 | 0.270 | 0.257 | 0.311 | 0.290 |
| ReAct | 0.278 | 0.268 | 0.545 | 0.165 | 0.244 | 0.333 | 0.287 | 0.382 | 0.274 |
| Expel | 0.371 | 0.341 | 0.182 | 0.293 | 0.260 | 0.298 | 0.269 | 0.382 | 0.311 |
| Strategy+BoN | 0.411 | 0.485 | **0.636** | 0.355 | 0.334 | 0.439 | 0.402 | 0.556 | 0.418 |
| Strategy+Reflect | **0.483** | 0.480 | 0.364 | **0.436** | 0.314 | 0.456 | **0.425** | 0.561 | 0.440 |
| ReAct+BoN | 0.402 | 0.444 | **0.636** | 0.347 | 0.396 | 0.456 | 0.402 | **0.568** | 0.423 |
| ReAct+Reflect | 0.408 | **0.505** | 0.545 | 0.367 | **0.480** | **0.526** | 0.414 | 0.511 | **0.452** |

*Table 11.* Per-incident performance of different agents on GPT-4o-mini. Best scores for each incident and the average are bolded.

| Agent | 5 | 34 | 38 | 39 | 55 | 134 | 166 | 322 | Avg. |
|---|---|---|---|---|---|---|---|---|---|
| Base | 0.248 | 0.229 | 0.000 | 0.238 | 0.194 | 0.263 | 0.115 | 0.329 | 0.219 |
| Strategy | 0.261 | 0.285 | 0.273 | 0.249 | 0.256 | 0.298 | 0.207 | 0.275 | 0.259 |
| ReAct | 0.257 | 0.298 | 0.273 | 0.240 | 0.188 | 0.368 | 0.147 | 0.329 | 0.250 |
| Expel | 0.283 | 0.256 | 0.273 | 0.226 | 0.250 | 0.333 | 0.205 | 0.364 | 0.265 |
| Strategy+BoN | 0.395 | 0.420 | **0.545** | 0.318 | 0.354 | 0.439 | 0.287 | 0.525 | 0.382 |
| Strategy+Reflect | **0.434** | **0.476** | 0.455 | 0.366 | 0.350 | 0.421 | 0.269 | 0.489 | 0.394 |
| ReAct+BoN | 0.411 | 0.427 | 0.273 | 0.357 | 0.288 | **0.474** | 0.292 | 0.500 | 0.378 |
| ReAct+Reflect | 0.395 | 0.468 | 0.182 | **0.414** | **0.364** | **0.474** | **0.345** | **0.554** | **0.414** |

*Table 12.* Per-incident performance of different agents on o3-mini. Best scores for each incident and the average are bolded.

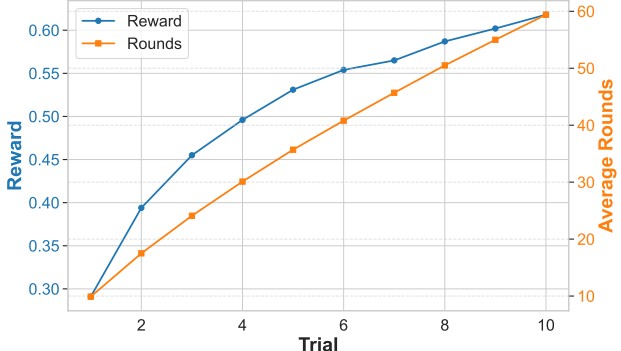

*Figure 8.* Average rounds and average reward with increasing trials. Tested with base agent + GPT-4o.

### E.5. Per-incident agent performance.

We report agent-method performance at the incident level to evaluate whether the benchmark can distinguish different agent strategies beyond aggregate model-level averages. Specifically, we collect a complete per-incident breakdown for all eight incidents and all evaluated agent strategies on GPT-4o, GPT-4o-mini, and o3-mini. Base, Strategy, ReAct, and Expel are evaluated in the single-trial setting, while BoN and Reflection variants are evaluated in the $k = 3$ setting. Within each group, methods use the same per-trial maximum-turn budget and environment configuration.

The per-incident results show several trends. First, no single agent dominates across all incidents. The best-performing

You are a security analyst.
You need to answer a given security question by querying the database.
The logs are stored in a MySQL database, you can use SQL queries to retrieve entries as needed. Note there are more than 20 tables in the database, so you may need to explore the schema or check example entries to understand the database structure.

Your response should always be a thought-action pair:
Thought: your reasoning
Action: your SQL query

In Thought, you can analyze and reason about the current situation,
Action can be one of the following:
(1) execute[your query], which executes the SQL query
(2) submit[your answer], which is the final answer to the question

*Figure 9.* Base Prompt for Baseline Agent.

**BASE_PROMPT +**
You should only give one thought-action per response. The action from your response will be executed and the result will be shown to you.
Follow the format "Thought: ....
nAction: ...." exactly.
Do not include any other information in your response. Wait for the response from one action before giving the next thought-action pair. DO NOT make assumptions about the data that are not observed in the logs.

*Figure 10.* Additional prompt added when testing with `o3-mini`, `o4-mini`, `o1-mini`.

method changes across incidents even for the same base model; for example, on GPT-4o, incident 39 is best solved by Strategy+Reflect with a score of 0.491, while incident 134 is best solved by ReAct+BoN and ReAct+Reflect, both with a score of 0.737. Second, the benchmark separates agent strategies on the same incident. Some incidents exhibit large performance gaps between methods; for example, on GPT-4o incident 38, ReAct obtains 0.091, while ReAct+Reflect obtains 0.818. Third, BoN and Reflection improve performance selectively rather than uniformly. Their gains are substantial on some incidents but smaller on others; for example, on o3-mini incident 38, Base obtains 0.000, Strategy obtains 0.273, and Strategy+BoN obtains 0.545, whereas on incident 134 the corresponding scores are 0.263, 0.298, and 0.439. Finally, the strongest method is model-dependent at the incident level. Even for the same incident, the preferred strategy can differ across models; for example, on incident 38, GPT-4o performs best with ReAct+Reflect at 0.818, while o3-mini performs best with Strategy+BoN at 0.545. These results indicate that the benchmark captures meaningful, incident-specific differences between agent strategies, rather than merely reflecting model-level performance trends.

### E.6. Prompts and Example

In Figure 13, we show the full processed details of an example run. At each step, we instruct LLM to give a thinking and an action in an ReACT style. The question generation prompt and the solution generation prompt are in Figure 15 and 16. We show the prompt for baseline model in Figure 9 and the enhanced version for reasoning models (e.g., `o1-mini`) in Figure 10. The strategy prompt is shown in Figure 12. We show five rules extracted with Expel in 11 and one example used in ReAct in Figure 14.

- Ensure that extracted IP addresses or other critical data directly aligns with the cybersecurity incident's timeframe and context. This avoids errors in connecting unrelated entities or events and ensures actionable insights.
- Align queries explicitly with essential relational identifiers (e.g., AlertId, ProcessId, DeviceName) relevant to the cybersecurity incident to ensure precise evidence extraction. Avoid unnecessary queries to broader tables unless relational data is exhausted, ensuring query efficiency and high investigation value.
- Refine queries to focus specifically on the relevant timeframe, user accounts, and IP details tied to suspicious activity to avoid noise from unrelated data. Leverage alert-specific metadata like Alert IDs or IP addresses directly linked to the cybersecurity incident to maintain query precision and deliver actionable insights.
- Focus queries specifically on the timeframe and context of the suspicious activity, ensuring alignment with the provided incident timeline to maintain investigation precision and actionable insights. This prevents errors and avoids unnecessary focus on unrelated data.
- Balancing initial exploration with leveraging primary attributes such as DeviceId, AccountUpn, or NetworkMessageId is essential. Avoid premature scope expansion before thoroughly investigating relevant relational links and structured data to target accessible evidence effectively while minimizing errors and misalignments.
...

*Figure 11.* **Five example rules extracted with Expel.** An Expel consists of the base prompt, all the extracted rules, and 1 demonstration trajectory.

You are a security analyst working on investigating a security incident.
You need to answer a given question about the security incident by querying the database of security logs provided to you.
The logs are stored in a MySQL database, you can use SQL queries to retrieve entries as needed.
Note there are more than 20 tables in the database, so you may need to explore the schema or check example entries to understand the database structure.

A security incident is composed of a group of related alerts connected by data elements or entities that are shared between the alerts such as User accounts, Hosts, Mailboxes, IP addresses, Files, Cloud applications, Processes, URLs etc. Alerts are signals that result from various threat detection activities. These signals indicate the occurrence of malicious or suspicious events in your environment. Besides the security alert signals, you can also find additional information in the raw logs like device events, user activities, network traffic, etc.

The best way to approach the question is to start from the alert(s) that are part of the security incident and then explore the related logs to understand the context of the incident. If the security incident table is available to you, first thing to do should be to find all the information about the given incident from the security incident table.

Other important tables to look at are the alert tables, which contains alerts and other additional information that was generated by the security monitoring tools. Once you find the alerts that are part of the security incident, you can explore the entities that are shared between the alerts to understand the context of the incident. These entities can be further used to find additional information from the logs and even alerts that you might have missed.

Your response should always be a thought-action pair:
Thought: ¡your reasoning¿
Action: ¡your SQL query¿

In Thought, you can analyze and reason about the current situation,
Action can be one of the following:
(1) execute[¡your query¿], which executes the SQL query
(2) submit[¡your answer¿], which is the final answer to the question

*Figure 12.* Strategy Prompt.

**Question:** A security incident was detected involving a sign-in from an anonymous IP address `170.54.121.63` at 2024-07-17 11:01:27. Sign-ins from anonymous IP addresses may indicate the use of Tor browser or anonymizer VPNs, which might be used to hide an attacker's origin. This was part of a larger incident involving suspicious activities. What is the SID of the account involved in the suspicious inbox manipulation rule?

**Thought 1:** First, I will look for tables that might contain information about sign-ins and inbox rules.
**Action 1:** execute[SHOW TABLES;]
**Observation 1:** [('AADManagedIdentitySignInLogs',), ('AADNonInteractiveUserSignInLogs',), ('AADRiskyUsers',), ('AADServicePrincipalSignInLogs',), ('AADUserRiskEvents',), ..

**Thought 2:** I will look for tables related to email or inbox manipulation rules, such as `EmailEvents` or `EmailPostDeliveryEvents`, to find the suspicious activity.
**Action 2:** execute[SELECT * FROM SigninLogs WHERE IPAddress = '170.54.121.63' AND Timestamp = '2024-07-17 11:01:27';]
**Observation 2:** ProgrammingError: Unknown column 'Timestamp' in 'where clause'               ➡️ Error

**Thought 3:** It seems that the column name for the timestamp is not `Timestamp`. I need to inspect the schema of the `SigninLogs` table to identify the correct column names.
**Action 3:** execute[DESCRIBE SigninLogs;]
**Observation 3:** [('TenantId', 'text', 'YES', '', None, ''), ... ('IPAddress', 'text', 'YES', ...              ➡️ Get Schema

**Thought 4:** The correct column name for the timestamp is `TimeGenerated`. I will use this column to filter the sign-in logs.
**Action 4:** execute[SELECT * FROM SigninLogs WHERE IPAddress = '170.54.121.63' AND TimeGenerated = '2024-07-17 11:01:27';]
**Observation 4:** []

**Thought 5:** I will broaden the search by looking for any sign-ins from the IP address `170.54.121.63` around the date `2024-07-17`.
**Action 5:** execute[SELECT * FROM SigninLogs WHERE IPAddress = '170.54.121.63' AND TimeGenerated LIKE '2024-07-17%';]
**Observation 5:** ('e34d562e-ef12-4c4e-9bc0-7c6ae357c01, ... 'e036dee7-fa23-42ef-a019-468c8d3401f1', ...              ➡️ User ID: e34d562e-...c015

Step 6: Try to access "EmailEvents" but got Error

Step 7: Get EmailEvents Schema

**Thought 8:** The `EmailEvents` table has a column named `RecipientObjectId` which likely corresponds to the user ID. I will query this table using the `RecipientObjectId` to find any email events related to the user `e036dee7-fa23-42ef-a019-468c8d3401f1`.
**Action 8:** execute[SELECT * FROM EmailEvents WHERE RecipientObjectId = 'e036dee7-fa23-42ef-a019-468c8d3401f1';]
**Observation 8:** [('e34d562e-ef12-4c4e-9bc0-7c6ae357c015', 'Azure AD', '2024-07-17 10:59:48.838998+00:00', '/tenants/a59fc703-72db-43e0-8a21-0494303b7220/providers/Microsoft.aadiam', 'Sign-in activity', '1.0', 'SignInLogs', '0', '', '', '0', '3b3e34f2-9ee3-4e34-856b-2fd71f795389', ...

Step 9: Get EmailPostDeliveryEvents Schema

**Thought 10:** I will query this table to find any suspicious actions related to the user `e036dee7-fa23-42ef-a019-468c8d3401f1`.
**Action 10:** execute[SELECT * FROM EmailPostDeliveryEvents WHERE RecipientEmailAddress = 'raphaelt@vnevado.alpineskihouse.co';]
**Observation 10 :** [('e34d562e-ef12-4c4e-9bc0-7c6ae357c015', ...'raphaelt@vnevado.alpineskihouse.co', 2dbfc9f0-951f-4dd2-692b-08dca64b9909-13604419176185138755'
                                    ➡️ **NetworkMessageId:** 2dbfc9...

**Thought 11:** The `NetworkMessageId` for this event is `2dbfc9f0-951f-4dd2-692b-08dca64b9909`. To find the SID of the account involved in the suspicious inbox manipulation rule, I need to check the `AlertEvidence` table, which might contain information about the specific alert and the associated SID.
**Action 11:** execute[SELECT * FROM AlertEvidence WHERE NetworkMessageId = '2dbfc9f0-951f-4dd2-692b-08dca64b9909';]
**Observation 11:** [('e34d562e-ef12-4c4e-9bc0-7c6ae357c015', ... 'fab778cb93-a82e-3819-fc00-08dca64e349a',
                                    ➡️ **AlertID:** fab778cb...

**Thought 12:** To find the SID of the account involved in the suspicious inbox manipulation rule, I need to check the Alert table. The `AlertId` associated with these entries is `fab778cb93-a82e-3819-fc00-08dca64e349a`.
**Action 12:** execute[SELECT * FROM Alert WHERE AlertId = 'fab778cb93-a82e-3819-fc00-08dca64e349a';]
**Observation 12:** []

**Thought 13:** I will check the `AlertInfo` table to see if it contains information about the alert.
**Action 13:** execute[SELECT * FROM AlertInfo WHERE AlertId = 'fab778cb93-a82e-3819-fc00-08dca64e349a';]
**Observation 13:** [('e34d562e-ef12-4c4e-9bc0-7c6ae357c015', '2024-07-17 10:50:25.276879+00:00', '2024-07-17 10:50:25.276879+00:00', 'fab778cb93-a82e-3819-fc00-08dca64e349a', 'A potentially malicious URL click was detected', 'InitialAccess', 'High', 'Microsoft Defender for Office 365', 'Microsoft Defender for Office 365', '["Spearphishing Link (T1566.002)"]', '', 'AlertInfo')]

Step 14: Get AlertEvidence Schema

**Thought 15:** I will query the `AlertEvidence` table to retrieve the `AccountSid` for the `AlertId` `fab778cb93-a82e-3819-fc00-08dca64e349a`.
**Action 15:** execute[SELECT AccountSid FROM AlertEvidence WHERE AlertId = 'fab778cb93-a82e-3819-fc00-08dca64e349a';]
**Observation 15:** [('',), ('S-1-5-21-1874151667-3554330288-105586563-1715',), ..
                                    ➡️ **SID: S-1-5-21-1...**

**Thought 16:** The `AccountSid` column contains the SID `S-1-5-21-1874151667-3554330288-105586563-1715` for the account involved in the suspicious inbox manipulation rule. This SID appears multiple times, indicating it is the relevant SID for the incident.
**Action 16:** submit[`S-1-5-21-1874151667-3554330288-105586563-1715`]

*Figure 13.* The full example of agent (with GPT-4o) solving a question.

Question:
An automated investigation was initiated manually by u141(u141@ash.alpineskihouse.co) on the host 'vnevado-win10e'. This investigation aimed to identify and review threat artifacts for potential remediation. As part of this security incident, a suspicious process executed a PowerShell command. Can you identify the account name that was associated with this activity?

Thought: I will start by listing all the tables in the database to understand the structure.
Action: execute[SHOW TABLES]
Observation:
```
[('AADManagedIdentitySignInLogs',), ('AADNonInteractiveUserSignInLogs',), ('AADProvisioningLogs',), ('AADRiskyUsers',),
('AADServicePrincipalSignInLogs',), ('AADUserRiskEvents',), ('AlertEvidence',), ('AlertInfo',), ('AuditLogs',),
('AzureMetrics',), ('CloudAppEvents',), ('DeviceEvents',), ('DeviceFileCertificateInfo',), ('DeviceFileEvents',),
('DeviceImageLoadEvents',), ('DeviceInfo',), ('DeviceLogonEvents',), ('DeviceNetworkEvents',), ('DeviceNetworkInfo',),
('DeviceProcessEvents',), ('DeviceRegistryEvents',), ('EmailAttachmentInfo',), ('EmailEvents',), ('EmailUrlInfo',),
('Heartbeat',), ('IdentityDirectoryEvents',), ('IdentityLogonEvents',), ('IdentityQueryEvents',), ('MicrosoftAzureBastionAuditLogs',),
('MicrosoftGraphActivityLogs',), ('SecurityAlert',), ('SecurityIncident',), ('SigninLogs',), ('ThreatIntelligenceIndicator',),
('UrlClickEvents',), ('Usage',)]
```

Thought: The 'SecurityIncident' table exists, which likely contains information about security incidents. I will query this table to find the incident initiated by 'u141' on the host 'vnevado-win10e'.
Action: execute[SELECT * FROM SecurityIncident WHERE initiated_by = 'u141' AND host = 'vnevado-win10e']
Observation:
ProgrammingError: Unknown column 'initiated_by' in 'where clause'

Thought: It seems the 'SecurityIncident' table does not have a column named 'initiated_by'. I need to inspect the structure of the 'SecurityIncident' table to understand its columns and find the relevant information.
Action: execute[DESCRIBE SecurityIncident]
Observation:
```
[('TenantId', 'text', 'YES', '', None, ''), ('TimeGenerated', 'text', 'YES', '', None, ''), ...]
```

Thought: The 'Owner' column may store the user who initiated the incident. I will check for incidents where 'Owner = 'u141'' and the description or title mentions 'vnevado-win10e'.
Action: execute[SELECT * FROM SecurityIncident WHERE Owner = 'u141' AND (Description LIKE 'Observation:
[]

Thought: The incident details might be in 'AlertInfo', which stores alerts related to security incidents. I will check its structure.
Action: execute[DESCRIBE AlertInfo]
Observation:
```
[('TenantId', 'text', 'YES', '', None, ''), ..., ]
```

Thought: To find the specific alert related to the PowerShell command execution, I will search for alerts with titles or categories related to PowerShell on the host 'vnevado-win10e'.
Action: execute[SELECT * FROM AlertInfo WHERE Title LIKE 'Observation:
```
[(..., 'dafa552c38-a2c9-4c21-a7a8-94e0ae671894_1', 'Suspicious PowerShell download or encoded command execution', ...),
(...,'da11fca4fc-2be9-4723-b9aa-7e66393c30c6_1', ...), ...]
```

Thought: I have found several alerts related to suspicious PowerShell activities. The next step is to identify the specific alert that occurred on the host 'vnevado-win10e'. I will need to look into the 'AlertEvidence' table to find evidence related to these alerts and check if any of them are associated with the host 'vnevado-win10e'.
Action: execute[SELECT * FROM AlertEvidence WHERE AlertId IN ('dafa552c38-a2c9-4c21-a7a8-94e0ae671894_1', 'da11fca4fc-2be9-4723-b9aa-7e66393c30c6_1', 'dadcc0cbd1-785d-4bbc-8217-22d99635321b_1', 'da6d201663-dfda-4823-8952-1a50128ba9bc_1', 'da3395ee5a-7be3-49e7-8b10-adcfd307354a_1') AND HostName = 'vnevado-win10e']
Observation:
ProgrammingError: Unknown column 'HostName' in 'where clause'

Thought: It seems the 'AlertEvidence' table does not have a column named 'HostName'. I need to inspect the structure of the 'AlertEvidence' table to understand its columns and find the relevant information.
Action: execute[DESCRIBE AlertEvidence]
Observation:
```
[('TenantId', 'text', 'YES', '', None, ''), ...]
```

Thought: I will check 'AlertEvidence' for any evidence linked to 'vnevado-win10e'.
Action: execute[SELECT * FROM AlertEvidence WHERE AlertId IN ('dafa552c38-a2c9-4c21-a7a8-94e0ae671894_1', ...) AND DeviceName = 'vnevado-win10e']
Observation:
[]

Thought: Since there are no results from the 'AlertEvidence' table for the given criteria, it is possible that the evidence related to the host 'vnevado-win10e' is not directly linked in the way I queried. I will query the 'AlertEvidence' table again, but this time I will look for any evidence related to the alerts that might contain account information, such as 'AccountName'.
Action: execute[SELECT AlertId, AccountName FROM AlertEvidence WHERE AlertId IN ('dafa552c38-a2c9-4c21-a7a8-94e0ae671894_1', 'da11fca4fc-2be9-4723-b9aa-7e66393c30c6_1', 'dadcc0cbd1-785d-4bbc-8217-22d99635321b_1', 'da6d201663-dfda-4823-8952-1a50128ba9bc_1', 'da3395ee5a-7be3-49e7-8b10-adcfd307354a_1')]
Observation:
```
[('dafa552c38-a2c9-4c21-a7a8-94e0ae671894_1', ''), ('dafa552c38-a2c9-4c21-a7a8-94e0ae671894_1', 'samuelf'),
('dadcc0cbd1-785d-4bbc-8217-22d99635321b_1', ''), ('da6d201663-dfda-4823-8952-1a50128ba9bc_1', 'samuelf'), ...]
```

Thought: The account 'samuelf' is associated with the suspicious PowerShell execution.
Action: submit[The account name associated with the suspicious PowerShell command execution is 'samuelf'.]

*Figure 14.* **ReAct Example.** For react prompt, we use the base prompt + 3 examples. Here we show one of the examples used.

Your goal is to ask a security question from the given data from a security analyst's perspective. You are given the start alert and end alert, and corresponding entities. The two alerts are connected by a alert-entity path. The start and end alert might be the same. You will use the start alert as the context, and ask a question about the entities in the end alert.

The JSON must have the following fields:
- "question": the question about the end alert. The question should be carefully crafted so that:
1. The question should be natural and relevant to the context, and it should be clear and have a deterministic answer.
2. But it should not leak the answer. If the start and end alert are the same, you should be more careful since the given entities may have overlapping information.
3. The question should be specific of the answer you are looking for, and the answer should match the question.
- "answer": the answer to the question. You may be given one or more entities from the end alert, select the most meaningful entity and make sure it is not leaked in the context or question.
- "context": the context from the start alert. you should combine the alert and the entities given in a consistent sentence. You can simplify the context a bit if it is too long. Make sure the answer is not leaked in the context. If the start alert or the related entities contains the answer, you should remove it from the context.

Examples:
##############
Start Alert:
Time: 8/14/2024, 10:34:41.578 PM
Name: Ntdsutil collecting Active Directory information
Description: Attackers might be using Ntdsutil to gather information for persistence or to move laterally in a network or organization. Ntdsutil is a command line tool that provides management facilities for Active Directory Domain Services (AD DS) and Active Directory Lightweight Directory Services (AD LDS). It was launched to maintain the database of AD DS.
Entities from this alert:
Type: process, Field: ExtractedFileName, Value: 'powershell.exe'
Type: host, Field: HostName, Value: 'vnevado-dc'

End Alert:
Time: 8/14/2024, 10:34:41.578 PM
Name: Ntdsutil collecting Active Directory information
Description: Attackers might be using Ntdsutil to gather information for persistence or to move laterally in a network or organization. Ntdsutil is a command line tool that provides management facilities for Active Directory Domain Services (AD DS) and Active Directory Lightweight Directory Services (AD LDS). It was launched to maintain the database of AD DS.
Entities from this alert: Type: process, Field: ProcessId‗CreatedTimeUtc‗CommandLine, Value: '2556‗2024-08-01t12:37:29.6522416z‗‗"powershell.exe" -encodedcommand iabuahqazabz...'
##############
Your response:

"context": "A file 'powershell.exe' was launched on host 'vnevado-dc', which might be an indicator of an attacker using Ntdsutil to gather information for persistence or to move laterally in a network or organization. Note: Ntdsutil is a command line tool that provides management facilities for Active Directory Domain Services (AD DS) and Active Directory Lightweight Directory Services (AD LDS). It was launched to maintain the database of AD DS.",
"question": "When was the last time the file 'powershell.exe' was launched on host 'vnevado-dc', and what was the process ID?",
"answer": "Time: 2024-08-01t12:37:29.6522416, Process Id: 2556"

##############
##############
Start Alert:
Time: 8/14/2024, 10:34:41.429 PM
Name: Suspicious credential dump from NTDS.dit
Description: Attackers dump NTDS.dit in order to obtain user's credentials which are stored in the domain controller.
Entities from this alert:
Type: process, Field: ProcessId‗CreatedTimeUtc‗CommandLine, Value: '6748‗2024-08-01t12:37:30.2769191z‗‗"ntdsutil.exe" "ac i ntds" ifm "create full c:
temp" q q'
Type: process, Field: ExtractedFileName, Value: 'ntdsutil.exe'

End Alert:
Time: 8/14/2024, 10:37:13.064 PM
Name: Suspicious Azure Resource Management activities by a risky user
Description: Suspicious cloud Azure Resource Management (ARM) activities were performed by a user account that signed in to a risky session. This alert was triggered based on a Microsoft Defender for Cloud alert related to ARM and Microsoft Entra ID Protection risk scores. Entities from this alert:
Type: account, Field: Email, Value: 'Megan Bower@vnevado.alpineskihouse.co'
##############
Your response:

"context": "A file 'ntdsutil.exe' was launched with this command line: 'ntdsutil.exe ac i ntds ifm create full c:
temp q q'. The Process ID was 6748. This process might be an indicator of an attacker dumping NTDS.dit in order to obtain user's credentials which are stored in the domain controller.",
"question": "Related to this alert, there is also a suspicious Azure Resource Management (ARM) activities, which is likely from the same user. Can you get the email of the user who performed the suspicious ARM activities?",
"answer": "Megan Bower@vnevado.alpineskihouse.co",

##############
(...one more example)
##############

*Figure 15.* Question Generation Prompt.

Given an alert-entity path, please generate a solution path, where the question asks about the end entity.
In each step of the solution path, please make sure you include the entity field and value.

Your response should be in JSON format, containing field "solution" which is a list of strings.

Examples:
##########
Solution path:
Time: 8/14/2024, 10:34:41.578 PM
Name: Ntdsutil collecting Active Directory information
Description: Attackers might be using Ntdsutil to gather information for persistence or to move laterally in a network or organization. Ntdsutil is a command line tool that provides management facilities for Active Directory Domain Services (AD DS) and Active Directory Lightweight Directory Services (AD LDS). It was launched to maintain the database of AD DS.
Entities from this alert:
Type: process, Field: ProcessId...CreatedTimeUtc...CommandLine, Value: '6748...2024-08-01t12:37:30.2769191z..."ntdsutil.exe" "ac i ntds" ifm "create full c:
temp" q q'
##########
Your response:
{
"solution": [
"The attacker launched ntdsutil with the command line 'ntdsutil.exe ac i ntds ifm create full c:
temp q q' at '2024-08-01t12:37:30.2769191z', with Process ID '6748'."
]
}
##########
##########
Solution path:
Time: 8/14/2024, 10:34:41.578 PM
Name: Ntdsutil collecting Active Directory information
Description: Attackers might be using Ntdsutil to gather information for persistence or to move laterally in a network or organization. Ntdsutil is a command line tool that provides management facilities for Active Directory Domain Services (AD DS) and Active Directory Lightweight Directory Services (AD LDS). It was launched to maintain the database of AD DS.
Entities from this alert:
Type: host, Field: HostName, Value: 'vnevado-dc'

Time: 8/14/2024, 10:37:13.045 PM
Name: Azure Resource Manager operation from suspicious proxy IP address
Description: Microsoft Defender for Resource Manager detected a resource management operation from an IP address that is associated with proxy services, such as TOR. While this behavior can be legitimate, it's often seen in malicious activities, when threat actors try to hide their source IP.
Entities from this alert:
Type: ip, Field: Address, Value: '185.220.101.1'

Time: 8/14/2024, 10:37:13.064 PM
Name: Suspicious Azure Resource Management activities by a risky user
Description: Suspicious cloud Azure Resource Management (ARM) activities were performed by a user account that signed in to a risky session. This alert was triggered based on a Microsoft Defender for Cloud alert related to ARM and Microsoft Entra ID Protection risk scores.
Entities from this alert:
Type: account, Field: AadUserId, Value: '6c16dea3-5326-461e-a48e-38b527df3a70'
##########
Your response:
{
"solution": [
"There is a collection of active directory information with ntdsutil.exe on host 'vnevado-dc'.",
"There is a suspicious Azure Resource Manager operation from a proxy IP address '185.220.101.1'.",
"There is a suspicious Azure Resource Management activities by a risky user with AadUserId '6c16dea3-5326-461e-a48e-38b527df3a70'."
]
}
##########
Solution path:
Time: 8/14/2024, 10:37:13.011 PM
Name: Email messages containing malicious URL removed after delivery
Description: Emails with malicious URL that were delivered and later removed -V1.0.0.3
Entities from this alert:
Type: account, Field: Name, Value: 'Megan Bower'

Time: 8/14/2024, 10:37:12.993 PM
Name: A potentially malicious URL click was detected
Description: We have detected that one of your users has recently clicked on a link that was found to be malicious. -V1.0.0.5
Entities from this alert:
Type: account, Field: Sid, Value: 'S-1-5-21-1840151660-3534030288-105586563-1127'
##########
Your response:

"solution": [
"The email account 'Megan Bower' received an email with a malicious URL.",
"The user with SID 'S-1-5-21-1840151660-3534030288-105586563-1127' clicked on the malicious URL."
]

##########

*Figure 16.* Solution Generation Prompt.

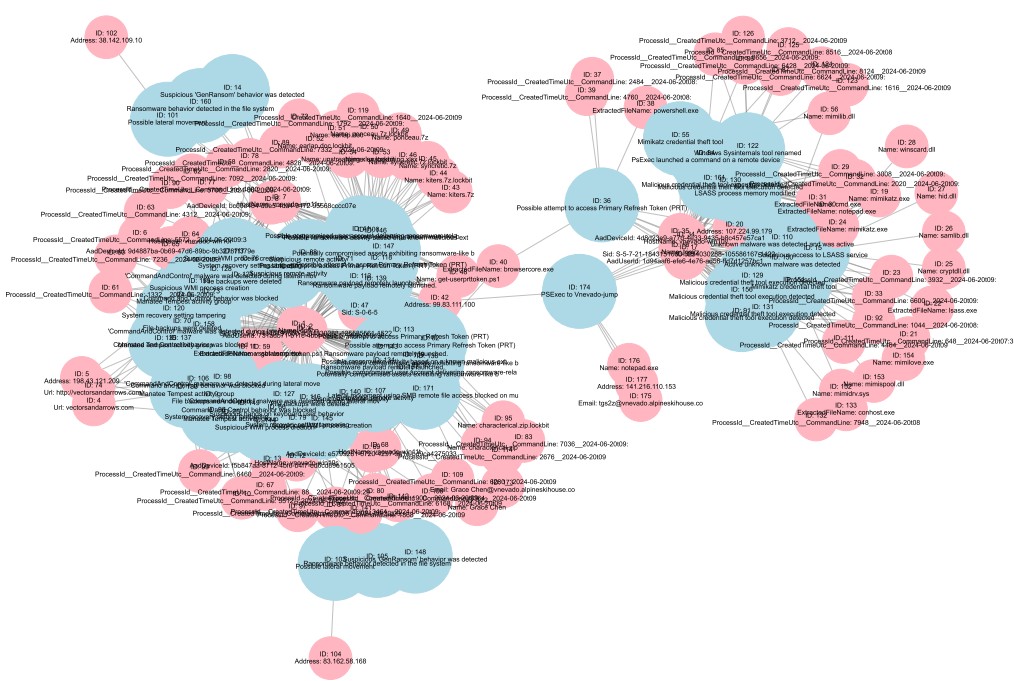

*Figure 17.* Graph of Incident 5. The bigger blue nodes represent alerts, and the smaller red nodes represent entities. (Only for illustrative purposes. Details can be hard to see.)

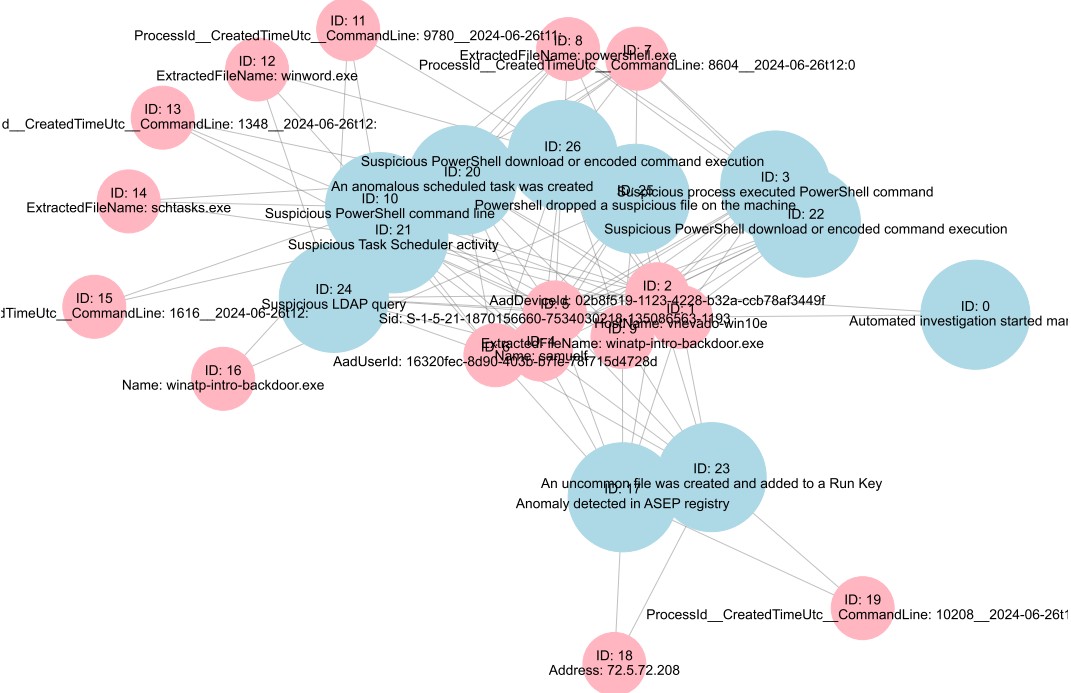

*Figure 18.* Graph of Incident 34.

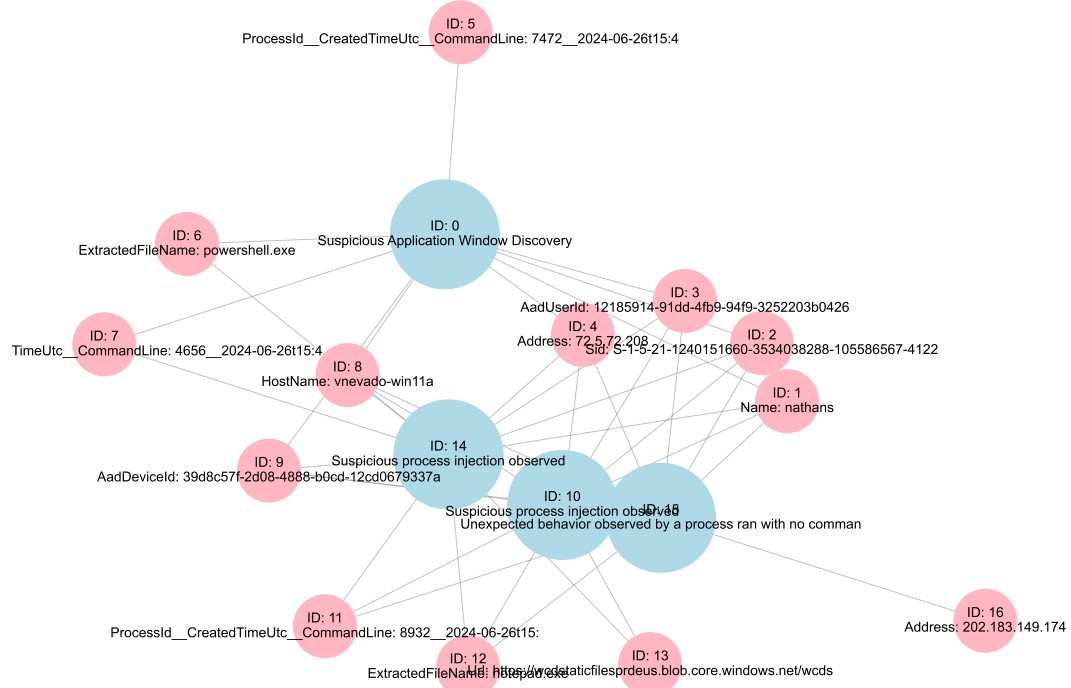

*Figure 19.* Graph of Incident 38.

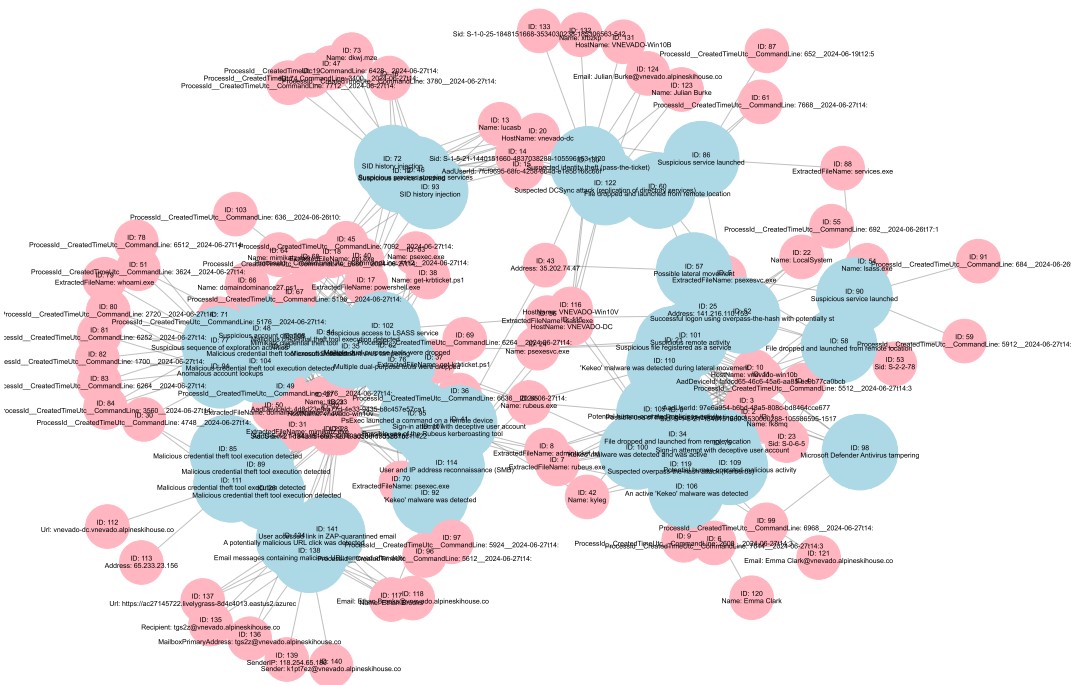

*Figure 20.* Graph of Incident 39.

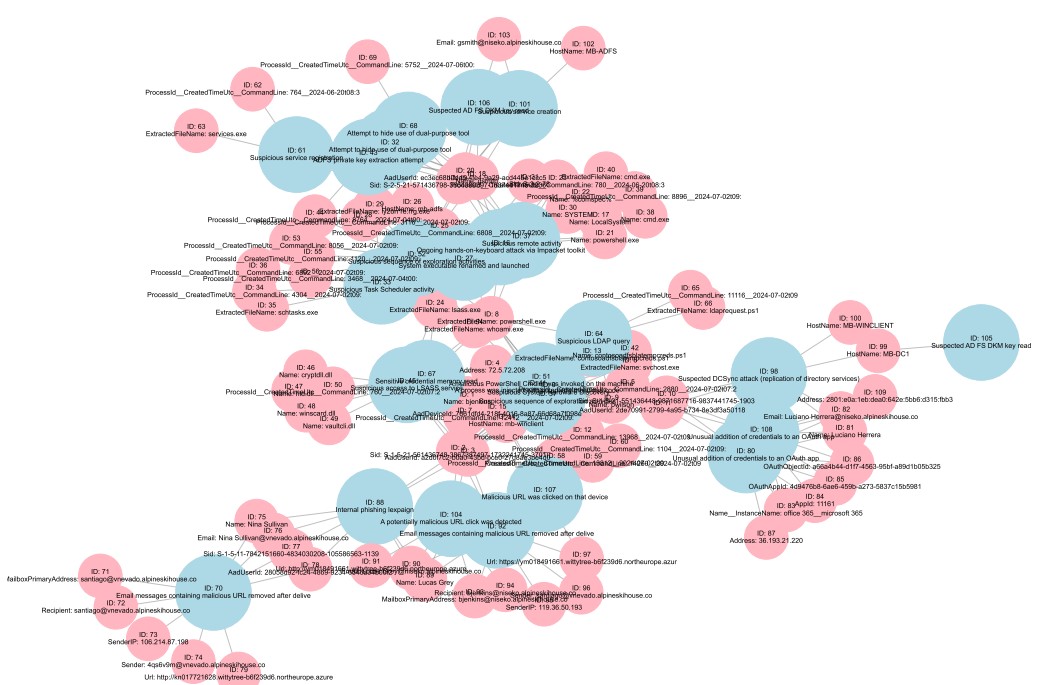

*Figure 21.* Graph of Incident 55.

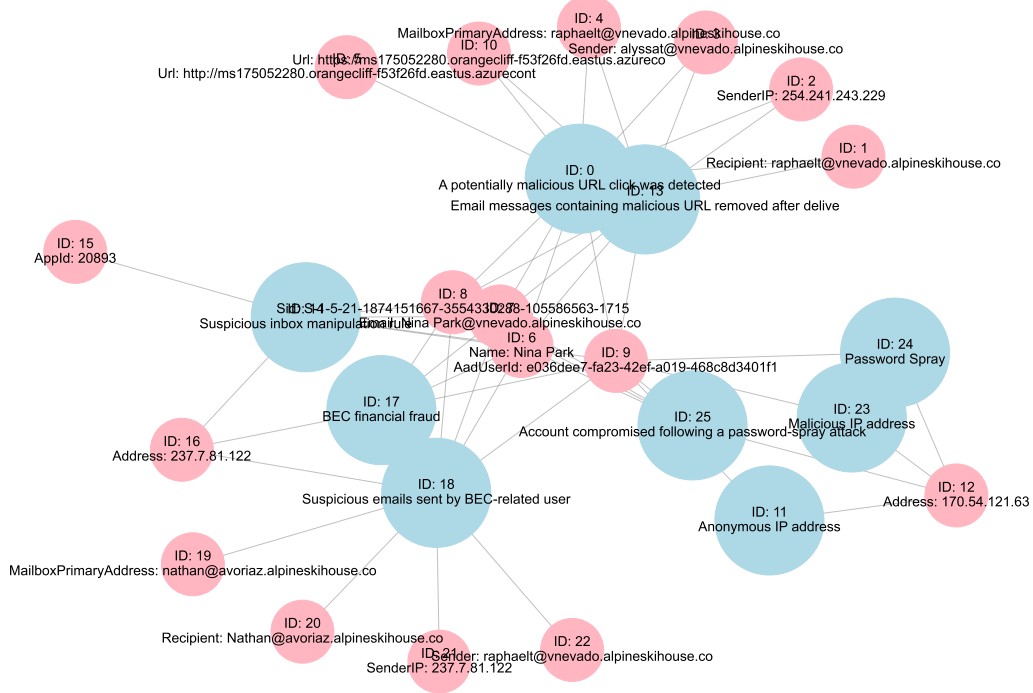

*Figure 22.* Graph of Incident 134.

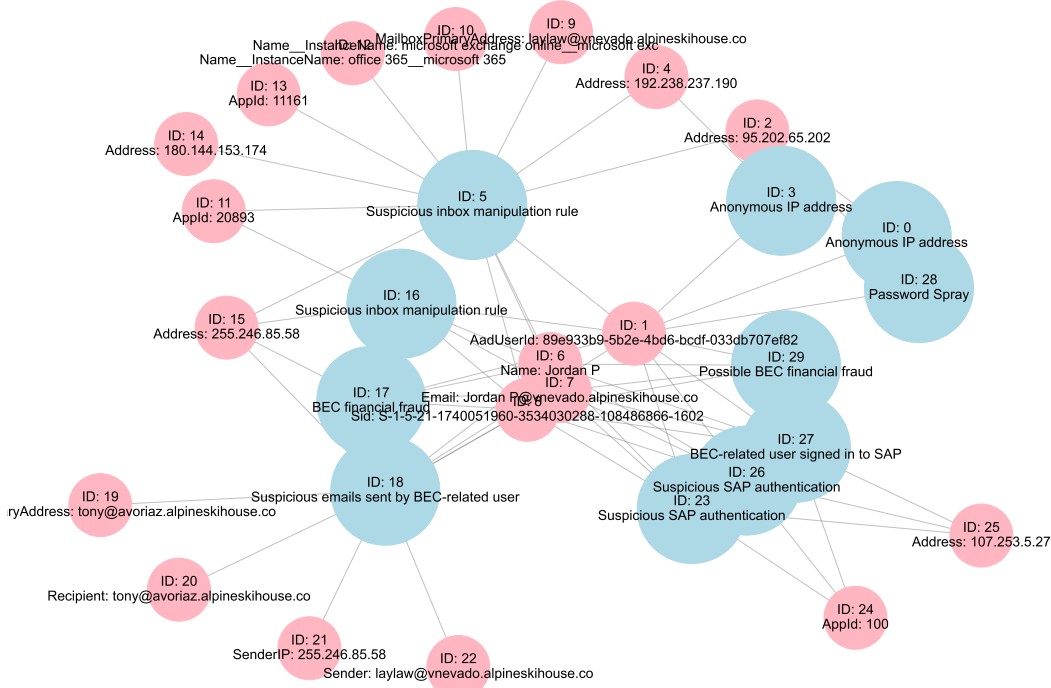

*Figure 23.* Graph of Incident 166.

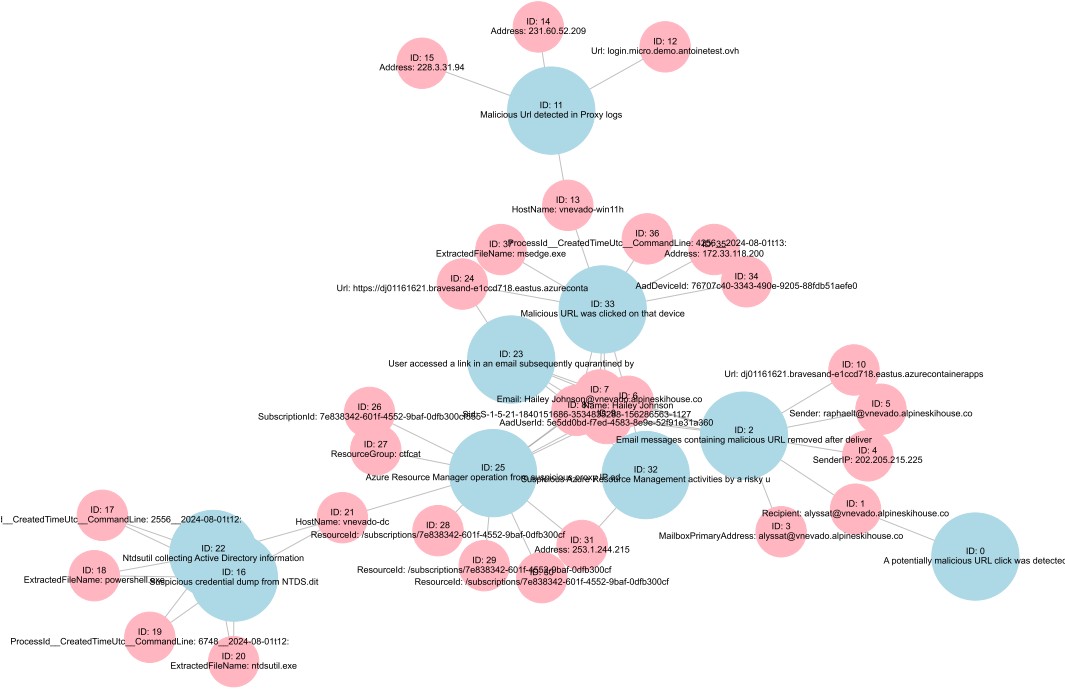

*Figure 24.* Graph of Incident 322.

Title: Operation Alpine Lockbit – Multi-Stage Manatee Tempest Ransomware Campaign

EXECUTIVE SUMMARY
On June 20, 2024, the Russia-based Manatee Tempest threat actor initiated a coordinated, multi-host campaign against the Alpine Ski House environment. The attack began with a drive-by download (FakeUpdate/SocGholish) via vectorsandarrows.com, delivering a BLISTER loader and Cobalt Strike beacons. The actor executed credential theft (Mimikatz, LSASS memory dumps, Primary Refresh Token access), leveraged WMI and PsExec for lateral movement across five Windows hosts, disabled backup and recovery features (VSSAdmin, bcdedit), and finally deployed LockBit-style ransomware payloads (.lockbit extension) to encrypt critical user data. Multiple Microsoft Defender alerts confirmed each stage, and automated containment actions blocked SMB lateral movement. No evidence of data exfiltration was observed, but asset recovery will require rebuilt systems and password resets.

INCIDENT TIMELINE
1. 2024-06-20 07:36 UTC – CredentialAccess: "Suspicious access to LSASS service" on vnevado-win10v via mimikatz.exe (Account: tgs2z).
2. 2024-06-20 08:51 – CredentialAccess: "Possible attempt to access Primary Refresh Token (PRT)" on vnevado-win10v by get-userprttoken.ps1 (tgs2z).
3. 2024-06-20 08:58 – Malware: "Mimikatz credential theft tool" detected on vnevado-win10v.
4. 2024-06-20 09:00 – CredentialAccess: "Malicious credential theft tool execution detected" on vnevado-win10v.
5. 2024-06-20 09:03 – Execution: "Suspicious WMI process creation" on vnevado-win10v (remote notepad spawn).
6. 2024-06-20 09:05 – Impact/CredentialAccess: LateralMovement: "PsExec launched a command on a remote device" from vnevado-win10v to vnevado-jump.
7. 2024-06-20 09:10 – DefenseEvasion/Impact: VSSAdmin & bcdedit tampering and "File backups were deleted" on win10s, win10r, win10v, win11u.
8. 2024-06-20 09:10 – CredentialAccess: Multiple "Possible attempt to access PRT" events and "Command and Control behavior was blocked" on win11u.
9. 2024-06-20 09:29 – InitialAccess: "Suspicious hands on keyboard user behavior" and "Manatee Tempest activity group" on win11t by curl vectorsandarrows.com.
10. 2024-06-20 09:29–09:31 – Drive-by download on win11t; backup tampering; multiple "Possible attempt to access PRT" and "Command and Control behavior was blocked."
11. 2024-06-20 09:31 – LateralMovement/Impact: "Possible compromised user account delivering ransomware-related files" (dp5hn) drops kiters.7z, syncretic.7z, ponceau.7z, unstreaming.xlsx and associated .lockbit files on win11u and win11t.
12. 2024-06-20 09:32 – Ransomware deployment: "Ransomware payload remotely launched" on win11u; "System recovery setting tampering" on win11u; "Ransomware behavior detected in the file system" on win11t.
13. 2024-06-20 09:34 – "Ransomware behavior detected in the file system" on win11t.
14. 2024-06-20 09:35 – Microsoft 365 Defender: SMB LateralMovement blocked on win11t.
15. 2024-06-20 09:37–09:38 – "Potentially compromised assets exhibiting ransomware-like behavior" across win11u and win11t.

TECHNICAL ANALYSIS
1. InitialAccess (T1189): A SocGholish-style fake-update landing page vectorsandarrows.com delivered via curl.exe on hosts vnevado-win11u and vnevado-win11t.
2. Execution (T1569.002/T1047): PowerShell and WMI spawned processes across remote hosts, breaking process trees. Multiple "Suspicious remote activity" and "Suspicious WMI process creation" alerts.
3. Credential Access (T1003/T1550.002/T1528): – LSASS memory dumps (mimikatz.exe) on vnevado-win10v. – Use of get-userprttoken.ps1 to steal PRT tokens on win10v, win10r, win11u, win11t. – Multiple credential-theft tool detections ("Mimikatz credential theft tool," "Malicious credential theft tool execution").
4. Persistence/Defense Evasion (T1036/T1547.005): – Sysinternals tools renamed (mimikatz.exe, conhost.exe) to evade detection on vnevado-win10v. – Automated disabling of Windows recovery features (vssblatemp.exe, bcdedit.exe) across hosts, deleting shadow copies (T1490).
5. Lateral Movement (T1021; T1021.002; T1021.006): – PsExec from vnevado-win10v to vnevado-jump. – SMB file operations blocked by Microsoft 365 Defender on win11t.
6. Collection (T1039): "Possible ransomware activity based on a known malicious extension" on win11u and win11t, observing mass file changes and .lockbit extension rhombus.
7. Impact (T1486): Ransomware payloads (kiters.7z.lockbit, syncretic.7z.lockbit, ponceau.7z.lockbit, characterical.zip.lockbit, earlap.doc.lockbit, unstreaming.xlsx.lockbit) dropped and executed, with subsequent ransomware behavior alerts.

AFFECTED ENTITIES
Hosts:
• vnevado-win11u.vnevado.alpineskihouse.co
• vnevado-win11t.vnevado.alpineskihouse.co
• vnevado-win10s.vnevado.alpineskihouse.co
• vnevado-win10v.vnevado.alpineskihouse.co
• vnevado-win10r.vnevado.alpineskihouse.co
• vnevado-jump.vnevado.alpineskihouse.co

Accounts:
• dp5hn (Grace Chen) – compromised initial account and ransomware delivery.
• tgs2z – malicious credential theft and lateral movement.
• k1pt7ez, 4qs6v9m, kelseyq, taylorz – lateral movement and ransomware targets.

Files:
• curl.exe (legitimate Windows tool abused)
• vssblatemp.exe (shadow-copy deletion)
• bcdedit.exe (boot config tampering)
• wbem
WmiPrvSE.exe (WMI spawn)
• mimikatz.exe, mimidrv.sys, mimispool.dll, mimilove.exe (credential theft)
• get-userprttoken.ps1 (PRT theft)
• kiters.7z(.lockbit), syncretic.7z(.lockbit), ponceau.7z(.lockbit), characterical.zip(.lockbit), earlap.doc(.lockbit), unstreaming.xlsx(.lockbit) – ransomware artifacts.

Network Indicators:
• Domain: vectorsandarrows.com
• IPs: 198.43.121.209, 99.83.111.100, 107.224.99.179, 38.142.109.10, 141.216.110.153

ATTACK METHODOLOGY
Phase 1 – Initial Access: Malvertising drive-by through vectorsandarrows.com (curl download).
Phase 2 – Execution: SocGholish loader → Cobalt Strike beacon; WMI and PowerShell process spawns for stealth.
Phase 3 – Credential Access: Mimikatz LSASS dumps; PRT token theft.
Phase 4 – Lateral Movement: PsExec and SMB; WMI remote activity on multiple endpoints.
Phase 5 – Persistence/Evasion: Sysinternals tool renaming; shadow-copy & recovery disabling (VSSAdmin & bcdedit).
Phase 6 – Collection: Enumeration of user documents and compression.
Phase 7 – Impact: Ransomware payload dropped and executed (.lockbit extension), file-system changes detected.

*Figure 25.* Incident 5 Report.

INDICATORS OF COMPROMISE
Malicious Domains / URLs:
• vectorsandarrows.com
Malicious IPs:
• 198.43.121.209
• 99.83.111.100
• 107.224.99.179
• 38.142.109.10
• 141.216.110.153

Malicious Files & Hashes (SHA256):
• BLISTER loader (curl.exe misuse): 2bbad800bc5058cad5631dbffd39fb8a293616479250c47b38dc8e8eb61dc3da
• vssblatemp.exe: 8c1fabcc2196e4d096b7d155837c5f699ad7f55edbf84571e4f8e03500b7a8b0
• mimikatz.exe: 61c0810a23580cf492a6ba4f7654566108331e7a4134c968c2d6a05261b2d8a1
• mimidrv.sys: 4ff7578df7293e50c9bdd48657a6ba0c60e1f6d06a2dd334f605af34fe6f75a5
• mimispool.dll: 05842de51ede327c0f55df963f6de4e32ab88f43a73b9e0e1d827bc70199eff0
• kiters.7z.lockbit, syncretic.7z.lockbit, ponceau.7z.lockbit, characterical.zip.lockbit, earlap.doc.lockbit, unstreaming.xlsx.lockbit

SEVERITY ASSESSMENT
Overall Impact: High
• Multiple confirmed credential thefts and lateral movements.
• Automated disabling of backup and recovery features.
• Deployment and execution of ransomware payloads across key user data.
• Loss of data integrity and potential, though unconfirmed, data encryption.
• Significant operational disruption requiring system rebuilds and password resets.

KEY LABELS & KEYWORDS Manatee Tempest, LockBit, SocGholish, BLISTER loader, Cobalt Strike, Mimikatz, LSASS dump, PRT theft, VSSAdmin, bcdedit, WMI, PsExec, Shadow copy deletion, Ransomware payload, .lockbit extension, Lateral Movement, Credential Access, Impact.

*Figure 26.* Incident 5 Report (Continued.)

Title
——

Macro-Enabled Document Dropper with PowerShell Backdoor Deployment and Dual Persistence Mechanisms

EXECUTIVE SUMMARY
——————

On June 26, 2024, the user "samuelf" on Windows 10 host vnevado-win10e opened a weaponized Word document (RS4_WinATP-Intro-Invoice.docm). A malicious macro triggered PowerShell execution in memory (T1059.001), which decoded and dropped a backdoor executable (WinATP-Intro-Backdoor.exe) to the user's Desktop. The attacker established persistence via:
• A RunOnce registry key entry (T1547.001) pointing to the dropped backdoor.
• A one-time Scheduled Task named "Yrei" (T1053.005) set to run the backdoor at a scheduled time.

Shortly thereafter, the backdoor initiated LDAP reconnaissance against the domain controller (T1018, T1087.x), obtaining directory information. Microsoft Defender ATP generated a sequence of alerts spanning "Execution," "Persistence," and "Discovery" tactics. Automated and manual investigations deemed some artifacts benign after remediation, but confirmed the attacker's multi-stage activity.

INCIDENT TIMELINE
——————

2024-06-26 11:57:19 UTC
• winword.exe (PID 9780) launched via user opening RS4_WinATP-Intro-Invoice.docm.
• PowerShell invoked by WINWORD.EXE with execution policy bypass to run embedded Base64 decoder script.

2024-06-26 12:00:39 UTC
• Suspicious PowerShell command line detected (Alert #3, T1059.001).
• PowerShell dropped WinATP-Intro-Backdoor.exe to Desktop and executed it (Alert #25).

2024-06-26 12:00:40 UTC
• schtasks.exe created a one-off scheduled task "Yrei" to run the backdoor (Alert #20).
• schtasks.exe ran the "Yrei" task immediately, launching the backdoor (Alert #21).

2024-06-26 13:17:18 UTC
• reg.exe added a RunOnce registry value under HKCU
RunOnce
Yrei to launch the backdoor on next logon (Alert #17 & #23).

2024-06-26 11:57:25 UTC
• Backdoor (or payload script) executed an LDAP query against the domain controller to enumerate users and groups (Alert #24).

2024-07-04 22:35 UTC
• Automated and user-initiated investigations in MDATP triaged and resolved alerts.

TECHNICAL ANALYSIS
——————

1. Delivery & Initial Execution (T1204.002 → T1059.001)
• Node 11: WINWORD.EXE opened the malicious .docm.
• Nodes 7 & 8: PowerShell launched with "-Exec Bypass -Command" to assemble Base64 chunks and write WinATP-Intro-Backdoor.exe.
• Alerts:
– #3 "Suspicious process executed PowerShell command"
– #10 "Suspicious PowerShell command line"
– #26 "Suspicious PowerShell download or encoded command execution"

2. Payload Drop & Execution (T1059.001)
• Node 16: WinATP-Intro-Backdoor.exe created on Desktop.
• Alert #25: "PowerShell dropped a suspicious file on the machine."

3. Persistence Mechanisms
a. Scheduled Task (T1053.005)
• Node 13 (PID 1348): schtasks.exe /create /SC ONCE /TN Yrei /TR ". . . Backdoor.exe"
• Alert #20: "An anomalous scheduled task was created."
• Alert #21: "Suspicious Task Scheduler activity" when the task ran.

b. Registry RunOnce (T1547.001)
• Node 19 (PID 10208): reg.exe ADD HKCU
RunOnce /v Yrei /d • Alert #17: "Anomaly detected in ASEP registry"
• Alert #23: "An uncommon file was created and added to a Run Key"

4. Discovery & Reconnaissance (T1018, T1069, T1087, T1558.003)
• Node 24: LDAP query via backdoor or script to enumerate directory info.
• Alert #24: "Suspicious LDAP query."

5. Investigation & Triage
• Node 0: Automated investigation started and marked benign for some artifacts.
• Alerts resolved in MDATP console; manual follow-up recommended (patch, AV, forensic).

*Figure 27.* Incident 34 Report

AFFECTED ENTITIES

Hosts:
• vnevado-win10e.vnevado.alpineskihouse.co (MdatpDeviceId cbb9f..., Windows 22H2)

Users/Accounts:
• samuelf (UPN: samuelf@vnevado.alpineskihouse.co, SID S-1-5-21...-1193)

Processes:
• WINWORD.EXE (PID 9780)
• powershell.exe (PIDs 8604 & 200)
• schtasks.exe (PIDs 1348 & 1616)
• reg.exe (PID 10208)
• cmd.exe (PID 2264)

Files:
• RS4_WinATP-Intro-Invoice.docm (delivery document)
• WinATP-Intro-Backdoor.exe (backdoor) – SHA1: 5e1c8874b29de480a0513516fb542cad2b049cc3; SHA256: 929cf5c2a2ce25d82699fc1bfe578bbe8abedce0e477a40980016ee32c2c7cbe
• YreianBackdoor.ps1 (indicated in LDAP-stage parent PS command)

Registry Keys/Values:
• HKCU
RunOnce
Yrei → "
Network Indicators:
• LDAP target domain controller (no external C2 observed)
• External IP seen (72.5.72.208) associated with Run key anomaly

ATTACK METHODOLOGY

Tactics & Techniques (MITRE):
Execution (T1059.001) – PowerShell
Persistence (T1053.005) – Scheduled Task
Persistence (T1547.001) – Registry Run Keys
Discovery (T1018, T1069, T1087.x) – LDAP, account enumeration
Privilege Escalation (T1112) – Potential credential theft via registry
Defense Evasion – Encoded commands in PowerShell

INDICATORS OF COMPROMISE

File Hashes:
• WinATP-Intro-Backdoor.exe – SHA256 929cf5...7cbe
• PowerShell.exe on device – SHA256 9785001b0dcf755eddb8af294a373c0b87b2498660f724e76c4d53f9c217c7a3

Registry:
• HKCU
RunOnce
Yrei

Scheduled Task:
• TaskName: Yrei – triggers C:
Users
samuelf
Desktop
WinATP-Intro-Backdoor.exe

Network:
• 72.5.72.208 (external IP in registry anomaly)

Process Artifacts:
• Command lines showing Base64 assembly and file write operations

SEVERITY ASSESSMENT

Impact: High – execution of a persistent backdoor, credential and directory enumeration.
Confidentiality & Integrity: Attacker maintained foothold and could exfiltrate or modify data.
Availability: No direct impact observed, but persistence allows future disruptive actions.

Important Labels & Keywords

T1059.001, T1053.005, T1547.001, T1018, T1087.002, T1087.003, T1558.003, WinATP-Intro-Backdoor.exe, RS4_WinATP-Intro-Invoice.docm, Yrei, RunOnce, schtasks, LDAP reconnaissance, Base64 PowerShell injector.

*Figure 28.* Incident 34 Report (Continued.)

Title
Multi-Stage Fileless Attack: PowerShell Execution, Process Injection, and Covert C2 over Azure Blob and External IP

EXECUTIVE SUMMARY
On June 26, 2024, the AlpineSkiHouse host vnevado-win11a was compromised by a low-and-slow, fileless attack. The attacker used PowerShell to fetch and execute code from an Azure blob URL, performed reconnaissance (Application Window Discovery), injected malicious code into Notepad.exe to evade defenses, and established covert command-and-control (C2) communications with an external IP. No new files were written to disk; all steps leveraged living-off-the-land binaries and in-memory execution.

INCIDENT TIMELINE
2024-06-26 15:49:15 UTC
• WindowsTerminal (wt.exe) spawns PowerShell (PID 7472) with an encoded command referencing an Azure blob URL (WinATP-Intro-Fileless.txt).
2024-06-26 15:49:16 UTC
• Alert "Suspicious Application Window Discovery" (Low) triggered during reconnaissance (T1010) on host vnevado-win11a.
2024-06-26 15:49:41 UTC
• Second PowerShell instance (PID 4656) launched with execution-policy bypass and the same encoded C2 retrieval command.
2024-06-26 15:49:42 UTC
• Alert "Suspicious process injection observed" (Medium) marks the moment PowerShell (PID 4656) injects code into a target process (T1055.001).
2024-06-26 15:49:51 UTC
• Alert "Suspicious process injection observed" (Medium) again flags injection of code into Notepad.exe (PID 8932). Notepad launches with no arguments, exhibiting anomalous behavior.
2024-06-26 15:49:52 UTC
• Alert "Unexpected behavior by a process ran with no command line arguments" (Medium) records Notepad connecting out to IP 202.183.149.174 (T1218.011).

TECHNICAL ANALYSIS
1. Reconnaissance (T1010,T1518)
• PowerShell enumeration commands gathered system and application window data.
• Microsoft Defender ATP generated a low-severity "Suspicious Application Window Discovery" alert at 15:49:16.
2. Payload Retrieval & Execution (T1059.001)
• Encoded PowerShell fetched content from https://wcdstaticfilesprdeus.blob.core.windows.net/.../WinATP-Intro-Fileless.txt.
• No files dropped; execution happened in memory under "bypass" policy.
3. Process Injection (T1055, sub-techniques .001–.005)
• The in-memory payload injected into Notepad.exe (PID 8932), hiding malicious code inside a trusted process.
• Two separate Defender ATP alerts ("Suspicious process injection observed", Medium) fired covering injection start and end times.
4. Masquerading & Unexpected Behavior (T1036,T1218.011)
• Notepad.exe, normally benign, exhibited network behavior without CLI args.
• It reached out to external IP 202.183.149.174, triggering the "Unexpected behavior" alert.

AFFECTED ENTITIES
Hosts
• vnevado-win11a.vnevado.alpineskihouse.co (DeviceId: 39d8c57f-2d08-4888-b0cd-12cd0679337a, RiskScore: Medium)
Accounts
• nathans (Luis Martinez; Sid: S-1-5-21-1240151660-3534038288-105586567-4122; UPN: nathans@vnevado.alpineskihouse.co)
Processes
• WindowsTerminal.exe (PID 9484 → 7472)
• PowerShell.exe (PID 7472, 4656)
• Notepad.exe (PID 8932)
External IPs & URLs
• 72.5.72.208 (public IP seen on host's last external IP)
• 202.183.149.174 (C2 callback)
• https://wcdstaticfilesprdeus.blob.core.windows.net/.../WinATP-Intro-Fileless.txt

ATTACK METHODOLOGY
• Initial reconnaissance via built-in tools (MITRE T1010,T1518)
• Fileless payload retrieval and execution through PowerShell (T1059.001)
• In-memory code injection into Notepad for defense evasion and persistence (T1055.*)
• Masquerading / unexpected behavior detection in Notepad (T1036)
• Network-based C2 communication over HTTPS and direct IP (T1218.011)

INDICATORS OF COMPROMISE (IOCs)
File Hashes (PowerShell.exe – legitimate but abused)
• SHA256: 3247bcfd60f6dd25f34cb74b5889ab10ef1b3ec72b4d4b3d95b5b25b534560b8
Notepad.exe Hash (legitimate)
• SHA256: 93379a2f43ac1189f29533c4de3f594bb04db76461f1d704620b9cc7cc718eeb
URLs & Domains
• wcdstaticfilesprdeus.blob.core.windows.net – WinATP-Intro-Fileless.txt
External IPs
• 202.183.149.174 (C2)
• 72.5.72.208 (host external IP)

SEVERITY ASSESSMENT
Overall Impact: High
• Multi-stage, in-memory compromise without disk artifacts
• Successful defense evasion via living-off-the-land binaries and process injection
• Unauthenticated remote C2 established
• Potential pivot and credential theft risk

LABELS / KEYWORDS
#Fileless #Discovery #Execution #DefenseEvasion #ProcessInjection #PowerShell #Notepad #CobaltStrike #AzureBlob #C2 #T1059 #T1055 #T1010 #T1036 #T1218

*Figure 29.* Incident 38 Report

Title
Operation Alpine Storm

EXECUTIVE SUMMARY
Between June 27, 2024 14:30 UTC and June 27, 2024 14:35 UTC, a human-operated intrusion chain hit AlpineSkiHouse's environment. Starting with a malicious URL click, the adversary executed a PowerShell "DomainDominance27" script to drop dual-use tools (Mimikatz, Rubeus, PsExec), disabled real-time antivirus, harvested credentials via Mimikatz and Rubeus (kerberoasting), created a backup domain account (BDAdmin), injected SID history into that account for elevated privileges, performed DCSync against the domain controller, and used overpass-the-hash and pass-the-ticket techniques to move laterally. Core systems compromised: vnevado-Win10V (user tgs2z/Ethan Brooks), vnevado-Win10B (user fk8mq/Emma Clark), and vnevado-DC.

INCIDENT TIMELINE
2024-06-27 14:31:27 UTC
• "Orchestrator.ps1" kicks off on vnevado-Win10V as user tgs2z.
2024-06-27 14:32:08 UTC
• DomainDominance27.ps1 executed by tgs2z → drops PsExec, Mimikatz, Rubeus.
2024-06-27 14:32:12 → 14:32:25 UTC
• Recon (whoami, net user/group/domain queries) by tgs2z (Discovery T1087).
2024-06-27 14:32:21 UTC
• Antivirus alert: Kekeo malware detected (informational).
2024-06-27 14:32:35 UTC
• Mimikatz "sekurlsa::logonpasswords" (Credential Access T1003).
2024-06-27 14:32:37 UTC
• Mimikatz Pass-the-Hash ("sekurlsa::pth") targeting accounts kyleg & fk8mq (T1550.003).
2024-06-27 14:32:46 → 14:32:52 UTC
• Mimikatz "sekurlsa::pth /run:Get-KRBTicket.ps1" and Rubeus ticket theft (kerberoasting T1558.003).
• Disable Windows Defender real-time monitoring (Defense Evasion T1562.001).
2024-06-27 14:33:14 → 14:33:27 UTC
• PsExec from vnevado-Win10V to vnevado-Win10B to run Rubeus and dump service tickets (Lateral Movement T1021.002).
2024-06-27 14:33:32 → 14:33:38 UTC
• Mimikatz "kerberos::ptt" and DCSync via "lsadump::dcsync" (Credential Access T1003.006).
2024-06-27 14:33:43 UTC
• New domain user BDAdmin created on vnevado-Win10V (Persistence T1136.002).
2024-06-27 14:34:10 UTC
• PowerShell adds SIDHistory for BDAdmin in NTDS and restarts service (Privilege Escalation T1134.005).
• Suspicious service launched on vnevado-DC (Execution T1569.002).
2024-06-27 14:34:38 → 14:34:44 UTC
• Additional SID history injection and suspicious service injection on DC.
2024-06-27 14:35:00 UTC
• Root cause analysis and remediation completed (alerts resolved).
2024-06-27 17:04:00 UTC
• tgs2z clicks malicious URL in quarantined email (Initial Access T1566.002).

TECHNICAL ANALYSIS
1. Initial Access (Phishing T1566.002):
– vnevado-Win10V user tgs2z clicks URL "ac27145722.livelygrass-8d4c4013..." delivered via email.
2. Execution & Tool Deployment:
– DomainDominance27.ps1 drops PsExec, Mimikatz, Rubeus. Tools hashed as:
• PsExec.exe SHA256: 57492d33b7c0755bb411b22d2dfdfdf088cbbfcd010e30dd8d425d5fe66adff4
• Mimikatz.exe SHA256: 912018ab3c6b16b39ee84f17745ff0c80a33cee241013ec35d0281e40c0658d9
• Rubeus.exe SHA256: a1fddd460edd35ed449d32cc43bc15675c48a314a6fa5fb158e3bc4fea460be1
3. Defense Evasion & Persistence:
– Disabled Defender real-time monitoring via Set-MpPreference.
– Created BDAdmin account and injected SIDHistory (S-1-5-32-544) to escalate privileges.
4. Credential Access:
– Mimikatz "sekurlsa::logonpasswords" dumps plaintext creds from LSASS.
– Kerberoasting via Rubeus "dump /service:xfbzkp /user:lucasb" to steal service tickets.
– Overpass-the-hash: forging TGT from NTLM hashes.
5. Lateral Movement (T1021.002):
– PsExec remote executions to vnevado-Win10B and vnevado-DC to run Rubeus/Mimikatz.
6. Domain Persistence & Control:
– SID history injection and NTDS restarts on vnevado-DC.
– DCSync replication from DC to exfiltrate all account hashes (T1003.006).
7. Pass-the-Ticket (T1550.003):
– Pass TGTs to authenticate as Julian Burke on alternate hosts.

AFFECTED ENTITIES
Hosts:
• vnevado-Win10V (MachineID 7cc55a46...) – initial foothold, tgs2z "DomainDominance".
• vnevado-Win10B (MachineID 5c626a5b...) – lateral target for Rubeus.
• vnevado-DC (MachineID 43a4c3f27...) – domain controller impacted by DCSync & SID injection.
Accounts:
• tgs2z / Ethan Brooks (S-5-7-21-...1422) – initial operator account.
• fk8mq / Emma Clark (S-1-5-21-...1517) – service account & ticket target.
• lucasb / Julian Burke (S-1-5-21-...1120) – IT director, ticket impersonation.
• BDAdmin – attacker-created backup domain admin.
Network & URLs:
• IP 118.254.65.186 – phishing link source.
• IP 141.216.110.153 – RDP lateral drop.
• IP 35.202.74.47 – Win10B management.
• URL https://ac27145722.livelygrass-8d4c4013.eastus2.azurecontainerapps.io/

*Figure 30.* Incident 39 Report

ATTACK METHODOLOGY (MITRE ATT&CK)
• Initial Access: T1566.002 Phishing Link
• Execution: T1059.001 cmd, T1059.003 PowerShell
• Defense Evasion: T1562 (Disable AV), T1134.005 SID History Injection
• Persistence: T1136.002 New Account
• Privilege Escalation: T1134.005, T1550.002 Pass-the-Hash
• Credential Access: T1003.* (LSASS dump, DCSync), T1558.003 Kerberoasting
• Lateral Movement: T1021.002 PsExec, T1105 Remote File Copy
• Collection: T1550.003 Pass-the-Ticket
• Discovery: T1087, T1049 SMB sessions
• Impact: T1489 Service Stop (NTDS), T1543.003 Service Registry

INDICATORS OF COMPROMISE
File hashes:
• Mimikatz.exe SHA256: 912018ab3c6b16b39ee84f17745ff0c80a33cee241013ec35d0281e40c0658d9
• Rubeus.exe SHA256: a1fddd460edd35ed449d32cc43bc15675c48a314a6fa5fb158e3bc4fea460be1
• PsExec.exe SHA256: 57492d33b7c0755bb411b22d2dfdfdf088cbbfcd010e30dd8d425d5fe66adff4
• DomainDominance27.ps1 SHA256: b284932e65dd50b731f2c6dc266ab4fe46287581498ac4dc50f13922b58d8c72
Malicious URL:
• https://ac27145722.livelygrass-8d4c4013.eastus2.azurecontainerapps.io/
Phishing IP:
• 118.254.65.186

SEVERITY ASSESSMENT
Overall Impact: Critical
• Complete domain compromise via DCSync and account migration
• Backdoor domain admin created (BDAdmin)
• Credentials for all high-value accounts dumped
• Persistent access and full lateral control across environment

IMPORTANT LABELS AND KEYWORDS
AlpineSkiHouse, human-operated, DomainDominance27, Kerberoasting, DCSync, SIDHistory, overpass-the-hash, pass-the-ticket, Mimikatz, Rubeus, PsExec, Phishing, Zero-hour Auto Purge (ZAP), Domain Controller.

*Figure 31.* Incident 39 Report (Continued.)

Title of the Multi-Stage Attack
"Phishing-Enabled ADFS Key Exfiltration and Lateral Movement Campaign"

EXECUTIVE SUMMARY
Between July 1 and July 10, 2024, an adversary executed a phishing-enabled, hands-on-keyboard campaign against AlpineSkiHouse's Windows estate. Initial access was gained via a malicious URL in a spear-phishing email to internal mailboxes. Once a user clicked the URL, the attacker ran PowerShell scripts to harvest credentials (LSASS memory read), performed reconnaissance and hardware enumerations, then leveraged Impacket to move laterally from the initial endpoint (MB-WINCLIENT) to the ADFS server (MB-ADFS). On MB-ADFS they established persistence (scheduled tasks, service creation, renamed executables), injected code into trusted processes, stole ADFS private keys via LDAP queries, and executed a DCSync attack against the domain controller (MB-DC1) to replicate directory services. They also abused a compromised service account to add credentials to an OAuth application (SimulandApp) and launched an internal phishing campaign. The impact includes credential theft, unauthorized replication of AD data, exposure of ADFS signing keys, and persistent footholds.

INCIDENT TIMELINE
2024-07-01 21:49 UTC
• "Email messages containing malicious URL removed after delivery" (O365 ATP – InitialAccess). Spear-phishing emails with URL http://kn017721628.wittytree-b6f239d6.northeurope.azurecontainerapps.io delivered to user "Nina Sullivan" (santiago@vnevado...).

2024-07-02 09:45–09:48 UTC
• User "bjenkins" on MB-WINCLIENT clicked the malicious URL ("Potentially malicious URL click detected").
• PowerShell launched in user context to download/run Midnight14 payload.
• "Suspicious system hardware discovery" and "Malicious PowerShell Cmdlet invoked" alerts triggered.
• LSASS memory accessed and dumped ("Suspicious access to LSASS service" & "Sensitive credential memory read").
• "ContosoADFSblatempcreds.ps1" executed under pwilson's context to harvest ADFS creds (Process injection alert).
• Adversary performed reconnaissance ("Suspicious sequence of exploration activities").

2024-07-02 09:47–09:48 UTC
• Attacker executed Impacket toolkit on MB-WINCLIENT ("Ongoing hands-on-keyboard attack via Impacket").
• Used WMI/SMB to reach MB-ADFS ("Suspicious remote activity").
• Created scheduled task "Run-ExportADFSTokenSigninCert..." via schtasks.exe on MB-ADFS ("Suspicious Task Scheduler activity").
• Renamed system executable (iy2orr1e.rrg.exe – renamed PowerShell) and launched it to evade detection ("System executable renamed and launched").
• Injected code into services.exe and svchost.exe ("Process was injected ... malicious code").
• Registered a malicious Windows service DDLOXJDSQSNGMUKKFUXQ ("Suspicious service registration" & Azure ATP "Suspicious service creation").
• Performed LDAP queries against ADFS objects to extract private keys ("ADFS private key extraction attempt" & Azure ATP "Suspected AD FS DKM key read").

2024-07-02 09:48 UTC
• DCSync replication request issued from MB-WINCLIENT to MB-DC1 ("Suspected DCSync attack").
• Extracted NTDS.dit data and domain credentials.

2024-07-02 12:07–12:15 UTC
• pwilson added credentials of type Password to SimulandApp in Azure AD (MCAS "Unusual addition of credentials to an OAuth app").

2024-07-02 15:01 UTC
• Compromised accounts sent internal phishing messages to other employees ("Internal phishing campaign").

2024-07-03 and July 6–9
• Additional MCAS detections of dual-purpose tool executions under unexpected filenames and cleaning up proof-of-concept artifacts on MB-ADFS.

TECHNICAL ANALYSIS
1. Initial Access (T1566.002):
– Malicious URL delivered via O365 ATP, removed after delivery but clicked by bjenkins.
2. Execution (T1059): Powershell processes launched—Midnight14 payload in Downloads, ContosoADFSblatempcreds.ps1 to extract ADFS creds.
3. Discovery (T1082, T1016, T1087): Hardware enumeration and "whoami.exe" and network reconnaissance commands executed.
4. Credential Access (T1003, T1550): LSASS memory access and dump; DCSync requests to a domain controller.
5. Lateral Movement (T1021.002, T1105): Impacket toolkit used to pass WMI and SMB commands to MB-ADFS.
6. Persistence (T1053.005, T1543.003, T1098.001): Scheduled task creation, malicious service registration, OAuth app secret addition.
7. Defense Evasion (T1036.003): System executable renamed to "iy2orr1e.rrg.exe" to hide from default path checks.
8. Privilege Escalation (T1055, T1569.002): Code injection into trusted processes, service creation for elevated execution.
9. Credential Access – ADFS (T1087.002, T1528): LDAP queries to DKM and private key objects in ADFS, exfiltrating key material.
10. AD Replication (T1003.006): DCSync replication of directory services.
11. Phishing & Internal Recon (T1534): Compromised accounts sending phishing inside network.

*Figure 32.* Incident 55 Report

AFFECTED ENTITIES
Hosts
• MB-WINCLIENT (initial compromise, credential harvesting, lateral pivot)
• MB-ADFS (persistence, code injection, ADFS key extraction)
• MB-DC1 (directory replication target)

Accounts
• bjenkins (clicked link, ran PowerShell, LSASS access)
• pwilson (executed ContosoADFSblatempcreds, Impacket lateral, DCSync source)
• gsmith (lateral pivot user on MB-ADFS, service creation, key read)
• santiago@vnevado (initial phishing target; internal sender)
• Nina Sullivan, Lucas Grey (mailboxes used/compromised)

Files & Processes
• powershell.exe (encoded commands, script execution)
• ContosoADFSblatempcreds.ps1 (ADFS credential extractor)
• iy2orr1e.rrg.exe (renamed PowerShell payload)
• schtasks.exe, services.exe, svchost.exe (persistence)

Network Indicators
• 72.5.72.208 (phishing URL host)
• 106.214.87.198 / 119.36.50.193 (sender IPs)
• phishing URLs:
– http://kn017721628.wittytree-b6f239d6.northeurope.azurecontainerapps.io/
– http://ym018491661.wittytree-b6f239d6.northeurope.azurecontainerapps.io/

OAuth & Cloud
• SimulandApp (OAuth App ID 4d9476b8-6ae6-459b-a273-5837c15b5981)

ATTACK METHODOLOGY (MITRE ATT&CK)
Initial Access T1566.002
Execution T1059.001, T1059.003
Persistence T1053.005, T1543.003, T1098.001
Privilege Escalation T1055, T1569.002
Defense Evasion T1036.003
Credential Access T1003.001, T1003.006, T1550.002, T1087.002, T1528
Discovery T1007, T1016, T1018, T1033, T1049, T1069, T1087
Lateral Movement T1021.002, T1105
Collection & Exfil implicit via DCSync
Command & Control observed in impacket remote session
Impact unauthorized data access & ADFS key theft

INDICATORS OF COMPROMISE
Files/Hashes
• ContosoADFSblatempcreds.ps1 SHA256: ad6997e67a2625a8663cb9f84d2461048b0a973b5015ae4f4cba717745cab602
• iy2orr1e.rrg.exe SHA256: 75d6634a676fb0bea5bfd8d424e2bd4f685f3885853637ea143b2671a3bb76e9
• DKM key object reads of GUID 4cac49d3-29d3-407e-8c9b-b7ebb21541b2
Processes
• svchost.exe -k netsvcs -p -s Winmgmt (PID 2880)
• powershell.exe -EncodedCommand QwA6. . .
Network
• 72.5.72.208
• 106.214.87.198, 119.36.50.193
• phishing URLs above
Scheduled task name
• Run-ExportADFSTokenSigninCert.2024-07-02_09_48_23

SEVERITY ASSESSMENT
Overall Impact: High
• Multiple accounts compromised.
• Domain controller replication via DCSync.
• Exfiltration of ADFS private keys and DKM keys enables forging SAML/ADFS tokens.
• Persistent footholds on AD FS server and cloud application (OAuth secret).
• Internal phishing indicates attacker control of user identities.

LABELS & KEYWORDS
Phishing, PowerShell, Impacket, Lateral Movement, DCSync, ADFS Key Theft, ADFS Distributed Key Manager, LSASS Dump, Scheduled Task, Service Creation, Code Injection, OAuth App Secret, Internal Phishing, Cloud App Security.

*Figure 33.* Incident 55 Report (Continued.)

Title: Multi-Stage Phishing-Driven BEC and Account Takeover Attack

1. EXECUTIVE SUMMARY
On July 17–18, 2024, the user "raphaelt@vnevado.alpineskihouse.co" (Nina Park) fell victim to a phishing campaign that delivered a malicious URL. Following the click, an attacker leveraged that access to perform unauthorized sign-ins from anonymizing and malicious IP addresses, manipulated the user's Outlook inbox rules to hide incoming mail, and executed a business email compromise (BEC) fraud attempt by sending spoofed emails to external recipients. Over the next 24 hours, the attacker escalated with a password-spray attack against Azure AD, resulting in a confirmed account takeover. This multi-stage chain encompassed Initial Access (T1566.002), Credential Access (T1110.003/T1110.001), Defense Evasion (T1564.008), Collection (T1114.002), and Reconnaissance/Suspicious Activity (T1586).

2. INCIDENT TIMELINE
2024-07-17 10:49:35 UTC
• Alert (13): "Email messages containing malicious URL removed after delivery"
– A phishing email from alyssat@vnevado.alpineskihouse.co (IP 254.241.243.229) with subject "Lee Don't miss 1969-Con Event next month" delivered to raphaelt@. . . and quarantined post-delivery.

2024-07-17 10:50:25 UTC
• Alert (0): "A potentially malicious URL click was detected"
– User clicked http://ms175052280.orangecliff-f53f26fd.eastus.azurecontainerapps.io/ (T1566.002).

2024-07-17 10:56:50 UTC
• Alert (11): "Anonymous IP address"
– Sign-in to raphaelt@. . . from Tor/VPN IP 170.54.121.63 (Amsterdam) (Initial Access).
• Alert (23): "Malicious IP address"
– Corroborates sign-in from 170.54.121.63 flagged as malicious.
• Alert (24): "Password Spray"
– Credential spray detected from same IP targeting multiple accounts.

2024-07-17 11:04:19 UTC
• Phish URL cleanup processed (alert 13 final state).

2024-07-17 11:06:04 UTC
• Alert (14): "Suspicious inbox manipulation rule"
– Attacker created hidden/move/delete rule in Nina Park's mailbox (Defense Evasion T1564.008).
• Alert (17): "BEC financial fraud"
– Attacker created hide-incoming-mail rule in Azure AD session (Collection T1114.002).

2024-07-17 11:06:54 UTC
• Alert (18): "Suspicious emails sent by BEC-related user"
– Spoofed email "Re: Lee Don't miss 777-Con Event next month" sent from raphaelt@. . . to Nathan@avoriaz.alpineskihouse.co (IP 237.7.81.122) (T1586).

2024-07-18 13:48:02 UTC
• Duplicate "Malicious IP address" alert reconfirms prior sign-in risk.

2024-07-18 14:36:59 UTC
• Alert (24) processing completes for Password Spray detection.

2024-07-18 14:54:18 UTC
• Alert (25): "Account compromised following a password-spray attack"
– Confirmed unauthorized sign-in from unusual location/browser; Nina Park's account fully compromised.

3. TECHNICAL ANALYSIS
Alert 13 & 0 (T1566.002)
– A phishing email from alyssat@. . . (IP 254.241.243.229) targeted raphaelt@. . . with a malicious link. Office 365 ATP quarantined subsequent deliveries, but the user clicked before removal, establishing initial foothold.

Alert 11, 23 & 24 (Initial Access & Credential Access)
– Shortly after the click, a cloud-logon from IP 170.54.121.63 (Amsterdam; anonymizer/Tor) succeeded, indicating the attacker either harvested credentials or session tokens. Azure AD Identity Protection flagged the IP as both anonymous and malicious and detected a password spray against multiple accounts.

Alerts 14 & 17 (Defense Evasion & Collection)
– Within 15 minutes of initial access, the attacker modified Nina Park's Outlook inbox rules to hide or delete incoming messages, preventing detection and facilitating covert BEC operations.

Alert 18 (Suspicious Activity)
– Using the compromised mailbox, the attacker dispatched fraudulent invoices or event invitations to Nathan@avoriaz. . . , likely aiming to extort or redirect payments (BEC/T1586).

Alert 25 (Account Compromise Confirmation)
– A day later, a full account takeover is confirmed post-password-spray; sign-in patterns and browser attributes were anomalous.

4. AFFECTED ENTITIES
– User Account: Nina Park (raphaelt@vnevado.alpineskihouse.co; AadUserId e036dee7-. . . )
– Mailbox: raphaelt@vnevado.alpineskihouse.co
– Phishing sender: alyssat@vnevado.alpineskihouse.co; IP 254.241.243.229
– Malicious URLs:
• http://ms175052280.orangecliff-f53f26fd.eastus.azurecontainerapps.io/
• https://ms175052280.orangecliff-f53f26fd.eastus.azurecontainerapps.io/
– Attacker IPs: 170.54.121.63 (anon/malicious), 237.7.81.122 (SMTP relay for BEC)
– External Target: Nathan@avoriaz.alpineskihouse.co
– Cloud App: Microsoft Exchange Online (AppId 20893)

*Figure 34.* Incident 134 Report

5. ATTACK METHODOLOGY (TTP)
– T1566.002 Phishing: Malicious link delivered via email.
– T1110.003/T1110.001 Credential Access: Password spray.
– T1564.008 Defense Evasion: Inbox rule manipulation.
– T1114.002 Collection: Hidden rule created to exfiltrate incoming email.
– T1586 Suspicious Activity: Outbound BEC emails.
– Persistence & Lateral Move: Retained mailbox control, risk session tokens.

6. INDICATORS OF COMPROMISE (IOCs)
– Phishing URL: ms175052280.orangecliff-f53f26fd.eastus.azurecontainerapps.io
– Sender IP: 254.241.243.229
– Attacker IPs: 170.54.121.63, 237.7.81.122
– Compromised account UPN: raphaelt@vnevado.alpineskihouse.co
– External recipient: nathan@avoriaz.alpineskihouse.co
– NetworkMessageIds: 2dbfc9f0-951f-4dd2-692b-08dca64b9909, 47e56987-44a0-45c4-d1b1-08dca6508dd1

7. SEVERITY ASSESSMENT
Overall Impact: High
– Initial phishing led to credential theft and unauthorized mailbox control.
– BEC attempts risk financial loss and reputational damage.
– Password spray expanded compromise to other accounts.
– Defense Evasion tactics concealed malicious activity.

8. KEY LABELS & KEYWORDS
Phishing, BEC, Business Email Compromise, Password Spray, Account Takeover, Inbox Rule Manipulation, Malicious URL, Anonymous IP, Malicious IP, Azure AD Identity Protection, Office 365 ATP.

*Figure 35.* Incident 134 Report (Continued.)

Title of the multi-stage attack
Business Email Compromise & Data Exfiltration via Inbox Rule Manipulation and SAP Access

1. EXECUTIVE SUMMARY
Over a 36-hour period beginning July 22, 2024, an attacker leveraged anonymous IP logons and a password-spray campaign to gain initial access to the corporate Azure AD account of "Jordan P" (laylaw@vnevado.alpineskihouse.co). Once inside, the actor deployed malicious inbox rules (T1564.008) to hide and move mail for exfiltration, conducted a business-email-compromise (BEC) campaign against an internal recipient (tony@avoriaz.alpineskihouse.co), and then authenticated to SAP (T1078) to harvest sensitive financial data. The incident spans techniques in Initial Access, Credential Access, Defense Evasion, Collection and Exfiltration, and culminates in potential financial fraud.

*Figure 36.* Incident 166 Report.

2. INCIDENT TIMELINE
2024-07-22 08:41:20 UTC
• Azure AD Identity Protection Alert (AnonymousLogin, Medium)
• Client IP 95.202.65.202 (Frankfurt, DE, Tor/anonymizer) signs in as AadUserId=89e933b9. . . ef82

2024-07-22 08:41:20 UTC (same event processed later)
• IPC Password Spray Alert (T1110.003, T1110.001, High) from IP 95.202.65.202

2024-07-22 09:07:43 UTC
• Second Anonymous IP sign-in (192.238.237.190, Hamburg, DE) for same user

2024-07-22 09:18–09:49 UTC
• MCAS "Suspicious inbox manipulation rule" (High)
– New MoveToFolder rule "ITCleanup" on laylaw@. . .
– IPs involved: 180.144.153.174, 95.202.65.202, 192.238.237.190, and attacker IP 255.246.85.58
• MTP / Defender365 "Suspicious inbox manipulation rule" (High, T1564.008)
• Azure AD "BEC financial fraud" (High)

2024-07-22 09:38:16 UTC
• MTP "Suspicious emails sent by BEC-related user" (High, T1586)
– laylaw@. . . → tony@avoriaz.alpineskihouse.co via IP 255.246.85.58

2024-07-22 09:46:21 UTC
• MTP "Suspicious SAP authentication" (Medium, T1078)
– laylaw@. . . signs in to SAP app "Lia" (AppId=100) from IP 107.253.5.27
• Repeat SAP alerts processed at 12:59, 13:19, 14:09 UTC

2024-07-23 14:23 UTC (retroactive processing)
• IPC Password Spray attack detection re-flagged against same request

2024-07-23 16:05–16:09 UTC
• MTP "Possible BEC financial fraud" (Medium, T1114.003)
– laylaw@. . . flagged as Compromised

3. TECHNICAL ANALYSIS
• Initial Access: Anonymous IP alerts (Nodes 0 & 3) and Password Spray (Node 28) indicate brute-force attempts from Tor-exit and anonymizing VPNs (95.202.65.202, 192.238.237.190).
• Defense Evasion & Persistence: MCAS and Defender365 detect creation of a stealth inbox rule "ITCleanup" (Nodes 5 & 16) to auto-move or hide legitimate emails in Jordan P's mailbox, with attacker IP 255.246.85.58 marked "Attacker."
• Collection & Exfiltration: Suspicious outbound email (Node 18) to tony@avoriaz.alpineskihouse.co carries potential fraudulent instructions or invoice requests (T1586). A mail-message entity (Nodes 20–22) tied to IP 255.246.85.58 confirms exfiltration channel.
• Lateral Movement & Data Harvesting: Multiple "Suspicious SAP authentication" alerts (Nodes 23, 26, 27) show same compromised account signing into enterprise SAP application "Lia" using stolen credentials, searching financial records (T1078).

4. AFFECTED ENTITIES
Accounts
• laylaw@vnevado.alpineskihouse.co (Jordan P, SID S-1-5-21-. . .-1602)
• tony@avoriaz.alpineskihouse.co (mailbox recipient)

IP Addresses
• 95.202.65.202 (Frankfurt, Tor exit)
• 192.238.237.190 (Hamburg)
• 180.144.153.174 (Contextual)
• 255.246.85.58 (Attacker source)
• 107.253.5.27 (SAP access)

Cloud Applications
• Microsoft Exchange Online (AppId 20893)
• Office 365 / Microsoft 365 (AppId 11161)
• SAP "Lia" (AppId 100)

Mailboxes & Messages
• Inbox rule "ITCleanup"
• mailMessage to tony@avoriaz.alpineskihouse.co

5. ATTACK METHODOLOGY (MITRE ATT&CK)
• Initial Access: T1110.003 Password Spray; AnonymousLogin (AZURE_AD_IDP)
• Defense Evasion: T1564.008 Inbox Rule Hiding
• Credential Access: Password Spray; Risky Sign-in Alerts
• Collection: T1114.002 Email Collection; T1586 Phishing
• Exfiltration: Native Mail Forwarding
• Lateral Movement: T1078 Valid Accounts (SAP sign-in)

6. INDICATORS OF COMPROMISE (IOCs)
• IPs: 95.202.65.202 / 192.238.237.190 / 255.246.85.58 / 107.253.5.27
• Compromised Account: AadUserId 89e933b9-5b2e-4bd6-bcdf-033db707ef82
• Inbox Rule Name: ITCleanup
• Malicious mail recipient: tony@avoriaz.alpineskihouse.co
• Cloud AppId: 100 (SAP Lia)

7. SEVERITY ASSESSMENT
Overall impact is High. The attacker achieved account takeover of a privileged user mailbox, established persistent mailbox rule-based exfiltration, orchestrated BEC fraud, and accessed sensitive financial systems. The combination of credential compromise, data manipulation, and potential fund diversion poses significant financial and reputational risk.

8. IMPORTANT LABELS & KEYWORDS
Business Email Compromise (BEC), Password Spray, Anonymous IP, Inbox Rule Manipulation, SAP Authentication, T1110.003, T1564.008, T1586, T1078, Initial Access, Defense Evasion, Collection, Exfiltration.

*Figure 37.* Incident 166 Report (Continued.)

Title
"Phishing-Driven Domain Credential Harvest and Cloud Evasion Attack"

EXECUTIVE SUMMARY
On August 1, 2024, an attacker deployed a multi-stage campaign against AlpineSkiHouse. A targeted phishing email containing a malicious URL was delivered to user "alyssat@vnevado.alpineskihouse.co" (Hailey Johnson). The user clicked the link, which initiated further payload retrieval and C2 communications from the host vnevado-win11h. The actor then performed suspicious Azure Resource Manager operations via a proxy/TOR-associated IP, abusing compromised credentials. Soon after, the adversary executed credential-dumping commands on the domain controller vnevado-dc, extracting NTDS.dit via ntdsutil. The attack blended traditional e-mail phishing, proxy evasion, and on-premises credential theft to achieve domain compromise.

INCIDENT TIMELINE
• 2024-08-01 11:26:07 UTC – Phishing mail with URL "dj01161621.bravesand-e1ccd718.eastus.azurecontainerapps.io" delivered to alyssat@. . .
• 12:26:22 UTC – "User accessed a link . . . quarantined by ZAP" alert (Node 23)
• 12:26:33 UTC – "Malicious URL was clicked on that device" on host vnevado-win11h (Node 33)
• 12:28:19–12:33:14 UTC – Proxy logs detect vnevado-win11h (231.60.52.209) connecting to login.micro.demo.antoinetest.ovh (Node 11)
• 12:32:45 UTC – Suspicious Azure Resource Management activities by a "risky" user (Hailey Johnson) flagged, involving proxy IP 253.1.244.215 (Node 32)
• 12:33:16–12:33:44 UTC – "Potentially malicious URL click detected" (Node 0)
• 12:34:30–12:36:30 UTC – "Emails containing malicious URL removed after delivery" (Node 2)
• 12:36:22 UTC – Azure Resource Manager operation from suspicious proxy IP (Node 25) targeting VM vnevado-dc
• 12:37:29 UTC – PowerShell invoked with encoded command dropping AD tools (Node 16 → PID 2556)
• 12:37:30 UTC – NTDS.dit dump via ntdsutil ("Suspicious credential dump" alert, Node 16)
• 12:37:30 UTC – ntdsutil collecting AD information for discovery & lateral movement (Node 22)
• 12:54:34–12:54:55 UTC – Follow-up ZAP and MDATP alerts reiterate link activity (Nodes 22, 23)

TECHNICAL ANALYSIS
Stage 1 – Initial Access (T1566.002)
• A phishing email ("Follow up – Security 101 content") delivered to alyssat@. . . contained URL dj01161621.bravesand-e1ccd718.eastus.azurecontainerapps.io.
• Hailey Johnson clicked it via msedge.exe (PID 4256) on host vnevado-win11h (172.33.118.200), triggering Office 365 ATP and ZAP quarantine.

Stage 2 – Command & Control / Proxy Evasion
• vnevado-win11h (231.60.52.209) reached out to login.micro.demo.antoinetest.ovh, a TI-matched malicious domain.
• The attacker leveraged proxy/TOR IP 253.1.244.215 for Azure Resource Manager activities against subscription 7e838342-. . . (Resource Group ctfcat) and VM vnevado-dc.

Stage 3 – Credential Access (T1003, T1003.003)
• On domain controller vnevado-dc, PowerShell (PID 2556, SHA256=de96a6e6. . . ab32c) executed an encoded script to prepare an IFM snapshot.
• ntdsutil.exe (PID 6748, SHA256=0a302650. . . 6e36b5e) ran "ac i ntds ifm create full c:
temp" to dump NTDS.dit.

Stage 4 – Discovery & Collection (T1018; T1069.002; T1087.002; T1482)
• ntdsutil usage flagged under Collection and Discovery categories—adversary gathering AD database and permissions for potential persistence or lateral movement.

Stage 5 – Cloud Evasion & Persistence (T1496)
• Risky user Hailey Johnson's Azure activity raised Entra ID Protection alerts. The actor attempted ARM operations (VM run-command, listing NICs, schedules) to probe or alter cloud assets.

AFFECTED ENTITIES
Accounts
• Hailey Johnson (alyssat@vnevado.alpineskihouse.co; AAD 5e5dd0bd-. . . )

Hosts & Devices
• vnevado-win11h.vnevado.alpineskihouse.co (AadDeviceId 76707c40-. . . ; IPs 231.60.52.209, 172.33.118.200)
• vnevado-dc.vnevado.alpineskihouse.co (Domain Controller; IP 65.233.23.156; MdatpDeviceId 43a4c3f27b4ff68-. . . )
• Azure resources in subscription 7e838342-. . . : VM vnevado-dc, NIC vnevado-dc-nic, DevTestLab schedule shutdown-computevm-vnevado-dc

Processes & Files
• msedge.exe (PID 4256)
• powershell.exe (PID 2556; SHA256=de96a6e6. . . )
• ntdsutil.exe (PID 6748; SHA256=0a302650. . . )

Network Indicators
• URLs:
– dj01161621.bravesand-e1ccd718.eastus.azurecontainerapps.io
– login.micro.demo.antoinetest.ovh
• IPs: 202.205.215.225; 228.3.31.94; 231.60.52.209; 253.1.244.215; 172.33.118.200

ATTACK METHODOLOGY
• Initial Access: Phishing via malicious URL (T1566.002)
• Execution: Browser & PowerShell encoded commands (T1059)
• Persistence: ARM operations, cloud resource probing
• Privilege Escalation: Credential dumping from NTDS.dit (T1003.003)
• Discovery: AD database enumeration & system/network inventory (T1018; T1069.002; T1087.002)
• Collection: Exfiltration of credentials & configuration data (T1482)
• Defense Evasion: Use of zero-hour auto purge (ZAP), TOR-associated proxy IP (T1496)

*Figure 38.* Incident 322 Report

INDICATORS OF COMPROMISE
• URLs:
– dj01161621.bravesand-e1ccd718.eastus.azurecontainerapps.io
– https://dj01161621.bravesand-e1ccd718.eastus.azurecontainerapps.io/
– login.micro.demo.antoinetest.ovh
• IP Addresses: 202.205.215.225; 231.60.52.209; 228.3.31.94; 253.1.244.215; 172.33.118.200
• File Hashes:
– powershell.exe (SHA256=de96a6e69944335375dc1ac238336066889d9ffc7d73628ef4fe1b1b160ab32c)
– ntdsutil.exe (SHA256=0a3026509dc46556021152242b9bb7956925d16953b05a2f548df717e5e36b5e)
• Accounts: Hailey Johnson (alyssat@. . . ) – flagged as compromised
• Processes: PID 2556 (PowerShell); PID 6748 (ntdsutil.exe)

SEVERITY ASSESSMENT
Overall Impact: High
• Complete compromise of a user account and workstation
• Unauthorized credential dump of domain controller's NTDS.dit threatens full AD domain takeover
• Use of proxy/TOR for cloud operations indicates intent to evade detection and abuse Azure resources
• Immediate risk of lateral movement, privilege escalation, and persistent foothold both on-prem and in cloud

LABELS & KEYWORDS
Phishing; ZAP; InitialAccess; T1566.002; Powershell; ntdsutil; CredentialAccess; T1003.003; Discovery; CloudEvasion; ARM; TOR; SuspiciousActivity; DomainController; AzureResourceManager; MITRE ATT&CK.

*Figure 39.* Incident 322 Report (Continued.)

