# OpenReview forum: "ExCyTIn-Bench: Evaluating LLM agents on Cyber Threat Investigation"
_ICML.cc/2026/Conference — ICML 2026 regular_

### Official Review · Reviewer_ptE4 · 2026-03-07

**Soundness:** 2
**Presentation:** 2
**Significance:** 2
**Originality:** 3
**Overall Recommendation:** 4
**Confidence:** 4

**Summary:**

This paper proposes ExCyTIn-Bench, a benchmark for evaluating LLM-based agents on cyber threat investigation. The authors collect security logs and alert data in a controlled Azure tenant environment and provide a SQL-based setting spanning 57 log tables from Microsoft Sentinel and related services. They simulate eight attack scenarios and generate 7,542 QA instances, with 589 questions used for testing.

The question generation pipeline extracts alerts and entities from tables such as SecurityIncident and SecurityAlert, constructs an alert–entity bipartite graph, and uses graph-side context and entity/alert anchors to prompt an LLM to produce questions. A shortest-path procedure is then used to generate a “reference solving chain.” On the evaluation side, SQL queries are treated as actions and returned query results as observations; the benchmark uses a decayed reward that grants partial credit for intermediate progress. The experiments compare multiple foundation models and several agent prompting / test-time scaling strategies, indicating that the task remains challenging: the best-performing model achieves around 0.606 reward on the test set.

**Compliance With Llm Reviewing Policy:**

Affirmed.

**Final Justification:**

The paper has clear merits and the rebuttal resolved my primary concerns. However, I am not placing it at **Accept** because:
- The "agent evaluation" framing remains somewhat aspirational relative to the actual experimental depth (though the rebuttal helped significantly)
- The question generation validity concern, while addressed, still leaves some residual uncertainty about whether *all* generated questions are genuinely investigative

Nonetheless, the benchmark is a valuable contribution, the evaluation is solid, and the authors have demonstrated responsiveness to concerns. I recommend **Weak Accept** and encourage the authors to incorporate the rebuttal tables and clarifications into the final manuscript.

**Key Questions For Authors:**

- Dataset validity / causal relevance: What criteria or filtering steps ensure that the chosen alert pairs correspond to meaningful investigative relations (e.g., causal, temporal, or entity-overlap relationships) rather than arbitrary co-occurrence? If a significant portion of alert pairs are weakly related, it may inflate dataset size while reducing task validity, which would affect my assessment of the benchmark’s realism and usefulness.
- Attack-grounded labeling: How do you verify that the alert vertices used in question generation are truly attack-related (as opposed to benign alerts in the environment)? Clarifying this would change my confidence in whether the benchmark measures threat investigation rather than general alert triage noise.
- Agent-method coverage and reporting: Can you provide a complete, per-attack breakdown of performance for each evaluated agent strategy (e.g., ReAct / Expel / BoN / Reflection) under consistent budgets (steps, tool calls, context length)? If agent strategies behave differently across attacks, this would strengthen the “agent evaluation” claim and could improve my overall recommendation.

**Limitations:**

Yes

**Strengths And Weaknesses:**

**Strengths**

- Practical and relevant benchmark design. The benchmark targets a realistic SOC-style investigation workflow and provides an implementable evaluation protocol for threat-investigation agents, which could be valuable for driving progress in this area.
- Broad experimental coverage on models and settings. The paper compares a range of widely used LLMs and explores several prompting and test-time scaling approaches, which helps characterize the current capability boundary on this task.

**Weaknesses**

- “Agents” evaluation seems comparatively under-explored relative to model comparison. Although the title emphasizes evaluating agents, a large portion of the experiments primarily compare base LLMs. The agent-method comparison (e.g., ReAct, Expel, BoN, Reflection) is relatively limited, and it would strengthen the paper to report these methods more systematically, ideally across all eight attacks and with clearer breakdowns.
- Concerns about the question generation strategy and validity of generated questions. In the Question–Answer–Solution generation process, questions are generated by selecting pairs of alert vertices. It is not obvious that any pair of alert nodes yields a meaningful investigative question: pairs without a causal or operational relationship may produce questions that are uninformative for threat investigation. In addition, if the process cannot ensure that selected alert nodes correspond to genuine attack-related alerts (rather than benign or noisy alerts), this could affect dataset quality and the interpretability of benchmark results.

---

> ### Author Rebuttal · Authors · 2026-03-25
>
> ## Reply to Weakness 1 & Questions 3
>
> Thank you for pointing this out. Below we report agent-method performance more systematically at the incident level. In response, we collected a complete per-incident breakdown for all 8 incidents and all evaluated agent strategies on GPT-4o, GPT-4o-mini, and o3-mini. We will add these results in the revision. We will also clarify the comparison setup: Base / Strategy / ReAct / Expel are evaluated in the single-trial setting, while BoN / Reflection variants are evaluated in the k=3 setting; within each group, methods use the same per-trial max-turn budget and environment configuration.
> The new per-incident results show:
> * No single agent dominates all incidents. The best method changes across incidents even for the same base model (e.g., for GPT-4o, incident 39 is best with Strategy+Reflect = 0.491, while incident 134 is best with **ReAct+BoN / ReAct+Reflect = 0.737).
> * The benchmark clearly separates agent strategies on the same incident. Some incidents show very large gaps between methods (e.g., GPT-4o, incident 38: ReAct = 0.091 vs. ReAct+Reflect = 0.818).
> * BoN / Reflection helps selectively rather than uniformly. Their gains are especially large on some incidents but not all (e.g., o3-mini, incident 38: Base = 0.000, Strategy = 0.273, Strategy+BoN = 0.545; but on incident 134 the gap is smaller: 0.263 / 0.298 / 0.439).
> * The strongest method is also model-dependent at the incident level. Even on the same incident, the preferred strategy can differ by model (e.g., incident 38: GPT-4o is best with ReAct+Reflect = 0.818, while o3-mini is best with Strategy+BoN = 0.545).
>
> Overall, we believe these additional results strengthen the paper’s agent-evaluation claim by showing meaningful, incident-specific differences between agent strategies, beyond model-level averages.
>
> ### Table: Per incident performance of different agents on GPT-4o
> *Due to character constraint of rebuttal text, we only include results of GPT-4o here. later we will include results of GPT-4o-mini and o3-mini when reply is open.*
>
> | Agent            | 5     | 34    | 38    | 39    | 55    | 134   | 166   | 322   | Avg   |
> |------------------|-------|-------|-------|-------|-------|-------|-------|-------|-------|
> | Base             | 0.293 | 0.268 | 0.364 | 0.242 | 0.245 | 0.421 | 0.131 | 0.268 | 0.260 |
> | Strategy         | 0.263 | 0.371 | 0.273 | 0.239 | 0.201 | 0.277 | 0.280 | 0.325 | 0.273 |
> | ReAct            | 0.365 | 0.383 | 0.091 | 0.308 | 0.324 | 0.526 | 0.264 | 0.446 | 0.354 |
> | Expel            | 0.461 | 0.395 | 0.309 | 0.363 | 0.324 | 0.456 | 0.372 | 0.400 | 0.390 |
> | Strategy+BoN     | 0.457 | 0.515 | 0.364 | 0.469 | 0.414 | 0.526 | 0.460 | 0.543 | 0.473 |
> | Strategy+Reflect | 0.480 | 0.524 | 0.545 | 0.491 | 0.474 | 0.579 | 0.483 | 0.554 | 0.505 |
> | ReAct+BoN        | 0.614 | 0.554 | 0.455 | 0.473 | 0.554 | 0.737 | 0.517 | 0.574 | 0.563 |
> | ReAct+Reflect    | 0.581 | 0.573 | 0.818 | 0.449 | 0.530 | 0.737 | 0.538 | 0.585 | 0.563 |
>
> ## Reply to Weakness 2 & Question 1 and 2
>
>
> To answer these concerns, we first explain the overall process used to construct the benchmark and generate questions:
> 1. Real-world attack scenarios are collected by experts and simulated to produce raw security logs.
> 2. **Expert-crafted detection rules** are applied to these logs to identify attack behaviors and extract evidence, which is then converted into security alerts and incidents and **manually tagged by attack scenario**.
> 3. Alert–entity graphs are constructed from the extracted logs, where each alert contains relevant entities and alerts are connected when they **share overlapping entities**.
> 4. Questions are generated only from **connected alert pairs** in this graph; the graph serves as a scaffold for grounded question generation, while answering still requires examining the underlying logs and reconstructing the attack process.
>
> *On the concern that "arbitrary alert pairs may not form meaningful investigative questions":*
>
>  The benchmark does not use arbitrary alert pairs. It only selects pairs that are connected in the alert–entity graph, meaning they share investigative evidence and belong to the same connected chain rather than merely co-occurring in the environment. This constraint is intended to ensure investigative relevance, even when the relationship is not a direct causal edge. We will make this clearer in the paper.
>
> *On the concern that "selected alerts may be benign or noisy rather than attack-related":*
>
>  The alerts used for question generation are attack-related by construction. They are **produced by expert-crafted rules** over logs from simulated attacks and are **manually tagged by scenario**. Thus, the graph and generated questions are built from an explicitly attack-grounded subset of alerts and incidents, not from arbitrary benign events. Benign or noisy activity may remain in the raw logs for realism, but it is not used as graph nodes for question generation.

---

> > ### Author Rebuttal · Reviewer_ptE4 · 2026-04-01
> >
> > Your response regarding Weakness 1 has addressed my question, but the part concerning Weakness 2 has not been fully resolved yet. Specifically, the issue about the selected alert node possibly being a benign node. Based on my understanding, your detection mechanism (which relies on expert rules) may identify normal system behavior as an alert. That’s why I suggested that the alert you selected could be a benign or noisy node. Your response only mentioned that you generated these alerts using an expert knowledge detector and manually tagged them, but it did not verify whether the alerts are actually part of the attack chain (i.e., a true positive rather than a false positive).

---

> > > ### Author Response · Authors · 2026-04-01
> > >
> > > Thank you for the clarification. We want to explain that, when we mention expert rules, we mean that these rules are examined for **each incident specifically**, and there could be new rules added for these specific incidents if some related activities are not detected. Substantial efforts are taken to ensure that the SIEM can correctly and accurately generate the alerts and incident entry for these simulated attacks. Thus, the alerts we extract for these incidents are intended to be **related to the attack**, rather than normal system behavior. Besides these extracted alerts, there are many other alerts in the environment generated by more general alert-generation rules, but those are **not** the alerts we use for graph construction and question generation.
> > >
> > > At the same time, some of these attack-related alerts can still be repetitive or not very informative. For example, one attack step may involve running a script to try many different passwords, which can produce many alerts for individual trials. These alerts are still related to the attack, but they may be redundant, and each single alert may not look very valuable on its own.
> > >
> > > These are still useful for the benchmark because our goal is to test whether an agent can start from partial evidence and use the raw logs to recover other connected information. Using the same password example, if an alert tells us that a wrong-password trial happened, can the agent further identify the *IP address* associated with that trial? This is exactly the kind of capability the benchmark is designed to evaluate: the starting node provides known evidence, and the question asks about another connected piece of information represented by the end node.
> > >
> > > We hope this clarifies the concern. We would be happy to further clarify this point if there are any remaining questions. If this explanation addresses your concern, we would sincerely appreciate your reconsideration of recommendation.
> > >
> > >
> > > -----
> > > P.S. Below are remaining results for weakness 1. We will include these results and analysis in the paper.
> > >
> > > #### Table: Per incident performance of different agents on GPT-4o-mini
> > >
> > > | Agent            | 5     | 34    | 38    | 39    | 55    | 134   | 166   | 322   | Avg   |
> > > |------------------|-------|-------|-------|-------|-------|-------|-------|-------|-------|
> > > | Base             | 0.128 | 0.183 | 0.273 | 0.154 | 0.154 | 0.211 | 0.113 | 0.257 | 0.165 |
> > > | Strategy         | 0.324 | 0.334 | 0.455 | 0.298 | 0.224 | 0.270 | 0.257 | 0.311 | 0.290 |
> > > | ReAct            | 0.278 | 0.268 | 0.545 | 0.165 | 0.244 | 0.333 | 0.287 | 0.382 | 0.274 |
> > > | Expel            | 0.371 | 0.341 | 0.182 | 0.293 | 0.260 | 0.298 | 0.269 | 0.382 | 0.311 |
> > > | Strategy+BoN     | 0.411 | 0.485 | 0.636 | 0.355 | 0.334 | 0.439 | 0.402 | 0.556 | 0.418 |
> > > | Strategy+Reflect | 0.483 | 0.480 | 0.364 | 0.436 | 0.314 | 0.456 | 0.425 | 0.561 | 0.440 |
> > > | ReAct+BoN        | 0.402 | 0.444 | 0.636 | 0.347 | 0.396 | 0.456 | 0.402 | 0.568 | 0.423 |
> > > | ReAct+Reflect    | 0.408 | 0.505 | 0.545 | 0.367 | 0.480 | 0.526 | 0.414 | 0.511 | 0.452 |
> > >
> > > #### Table: Per incident performance of different agents on o3-mini
> > >
> > > | Agent            | 5     | 34    | 38    | 39    | 55    | 134   | 166   | 322   | Avg   |
> > > |------------------|-------|-------|-------|-------|-------|-------|-------|-------|-------|
> > > | Base             | 0.248 | 0.229 | 0.000 | 0.238 | 0.194 | 0.263 | 0.115 | 0.329 | 0.219 |
> > > | Strategy         | 0.261 | 0.285 | 0.273 | 0.249 | 0.256 | 0.298 | 0.207 | 0.275 | 0.259 |
> > > | ReAct            | 0.257 | 0.298 | 0.273 | 0.240 | 0.188 | 0.368 | 0.147 | 0.329 | 0.250 |
> > > | Expel            | 0.283 | 0.256 | 0.273 | 0.226 | 0.250 | 0.333 | 0.205 | 0.364 | 0.265 |
> > > | Strategy+BoN     | 0.395 | 0.420 | 0.545 | 0.318 | 0.354 | 0.439 | 0.287 | 0.525 | 0.382 |
> > > | Strategy+Reflect | 0.434 | 0.476 | 0.455 | 0.366 | 0.350 | 0.421 | 0.269 | 0.489 | 0.394 |
> > > | ReAct+BoN        | 0.411 | 0.427 | 0.273 | 0.357 | 0.288 | 0.474 | 0.292 | 0.500 | 0.378 |
> > > | ReAct+Reflect    | 0.395 | 0.468 | 0.182 | 0.414 | 0.364 | 0.474 | 0.345 | 0.554 | 0.414 |

---

### Official Review · Reviewer_FtgP · 2026-03-10

**Soundness:** 3
**Presentation:** 3
**Significance:** 3
**Originality:** 3
**Overall Recommendation:** 4
**Confidence:** 3

**Summary:**

The summary of the manuscript titled “ExCyTIn-Bench: Evaluating LLM agents on Cyber Threat Investigation ” is as follows.
It presents a Threat Investigation using ExcyTln-Bench to evaluate the LLM agent performance, in the system security domain, using event-based security logs. There is a significant gap in the automatic system design to execute event-based threat investigation on the basis of comprehensive information. The method uses the ExCyTln-Bench dataset for evaluation from a realistic Azure tenant and Microsoft Sentinel logs stored in a SQL database containing 57 log tables of system security logs. These logs are used to construct a threat investigation graph, in which a graph node contains event-based log information and adjacent edges show their in-between relationship. This technique led to an automatic ground truth preparation of 7542 investigation questions, exploring a pair of consecutive nodes or cross node. Cross-node is an extended feature to elaborate the security environment for the enduring context information extent.  The first node represents the event-based context, and the second node has the event-based results. Each pair is ordered to represent independent security investigation information. The experimental results for this pair show the best performance rate of 0.606. It shows that the security threat investigation is still a challenging task and has a profound margin for further improvement.

**Compliance With Llm Reviewing Policy:**

Affirmed.

**Key Questions For Authors:**

1	The dataset is limited to the Azure tenant and Microsoft Sentinel logs.  Other cloud networks should be added.
2	Graph node-based context and results have a limited set of attacks. It lacks any global attack information due to static security logs.
3	Security Questions generated by LLM have limited contextual information because they are directly dependent on security logs. A real security challenge can be included.
4	This dataset has system-specific security logs, a fixed graphical node representation, and a lack of real-time security data.
5	It has a single performance measure that is associated with the relationship strength of the node; there is no context-based, reasoning-based, or (adjacent and cross) pair-based performance measure.

**Limitations:**

The section's "broader Impact" explicitly states the limitations of the work; however, the potential impact of future use of the work on society is not mentioned.

**Strengths And Weaknesses:**

It is an evaluation of realistic data through an LLM agent for security threat investigation.
It uses a graph-based data representation that is human-readable and logical for the automatic context and its result extraction from security logs.
It supports automatic security question generation based on a node that has contextual data.
It has an adjacent and cross-node ground truth generation facility. So, it has both short-term and long-term threat investigation
It allows a wide level of system scalability, in terms of security logs, because the preprocessing technique has no manual interaction.

1	The dataset is limited to the Azure tenant and Microsoft Sentinel logs.  Other cloud networks should be added.
2	Graph node-based context and results have a limited set of attacks. It lacks any global attack info due to static security logs.
3	Security Questions generated by LLM have limited contextual information because they are directly dependent on security logs. A real security challenge is not including.
4	This dataset has system-specific security logs, a fixed graphical node representation, and a lack of real-time security data.
5	It has a single performance measure that is associated with the relationship strength of the node; there is no context-based, reasoning-based, or (adjacent and cross) pair-based performance measure.

---

> ### Author Rebuttal · Authors · 2026-03-25
>
> ## Reply to 1
>
> We want to clarify that the current benchmark already suffices for our stated goal: evaluating whether an agent can perform cyber investigation over realistic security telemetry: the ability to start from alerts/entities, query heterogeneous logs, to identify information. We agree that extending to additional cloud environments is an important future direction, and we discussed in Appendix C.1, the same overall framework can be transferred to other cloud/SIEM environments. We will make this point clearer in the revision.
>
> ##  Reply to 2
> The benchmark is not limited to isolated local graph neighborhoods. It is built from real-world multi-stage attacks, and its questions are grounded in those attack traces rather than synthetic edge patterns (Sec. 3.1; Appendix C.1). At a high level, benchmark construction mirrors a real investigation:
>
> 1. We begin with real-world attacks and simulate them in a controlled tenant environment.
> 2. We generate the detections and alerts a security analyst would use to investigate the attack.
> 3. We build a graph over those alerts and entities to organize the evidence and derive grounded questions.
> 4. The graph serves only as a scaffold for question generation; answering still requires analyzing the underlying logs and reconstructing the attack process.
>
> As discussed in Sec. 3.2, each question uses only a small investigation-traced subgraph within a much larger environment-level alert-entity graph that contains many other alerts and entities beyond the incident itself. If the concern is that the benchmark should capture broader environment-level context rather than clean per-incident slices, we already partially address this in the full-history setting over the 44-day log stream (Sec. 4.3). This setting is substantially harder because it removes prior knowledge of attack boundaries and requires investigation in a noisier, more realistic environment. Broader campaign-level context or external CTI could also be valuable, but that is outside the current benchmark’s scope.
>
> ## Reply to 3
> Our aim here is to **evaluate log-grounded investigative reasoning**, not open-ended security trivia or external-knowledge recall. The benchmark starts from real-world security attacks, simulates them in a controlled tenant, and creates the corresponding detections/alerts that a security analyst would use to investigate how the attack unfolded. We then build a graph over those alerts and entities and use that graph only as a structured mechanism for generating grounded questions.
>
> Importantly, **Answering the questions requires the agent to investigate the underlying logs, correlate evidence, and reconstruct the attack process in the same spirit as a SOC analyst**. As we explain in Sec. 3.2, more naive free-form LLM QA generation led to generic questions, non-deterministic answers, or questions requiring knowledge not present in the database. Grounding questions in logs is therefore a deliberate design choice needed for reliable and reproducible evaluation, not a simplification of the task.
>
> ## Reply to 4
> Our benchmark is designed to evaluate whether an agent can perform log-grounded cyber investigation in a realistic and reproducible setting. This setup is sufficient for that purpose: although the logs come from a specific system, the agent must still analyze heterogeneous security telemetry, correlate evidence across sources, and reconstruct how an attack unfolded. That is the core capability we aim to measure.
>
> The graph representation is introduced only to support grounded question generation and evaluation, not to replace the investigation process. The agent still queries the same kind of telemetry a human security analyst would investigate, so the benchmark remains grounded in realistic analyst workflows.
>
> Similarly, the logs are operational telemetry collected from a realistic environment. They are “static” only in the benchmarking sense that we freeze a snapshot to ensure reproducible and stable comparison across agents. Without a fixed snapshot, evaluation would not be consistent.
>
> ## Reply to 5
> We would like to clarify that the benchmark is not evaluated using only a single endpoint score tied to node relationships. As described in Sec. 3.3, evaluation includes both final answer correctness and decayed reward over intermediate investigative steps. This means the benchmark already gives credit for partial investigative progress, such as recovering useful intermediate IoCs, even when the final answer is incomplete.
>
> In addition, Sec. 3.2 uses shortest-path length as a proxy for question difficulty, not as the sole evaluation metric. If the reviewer’s concern is that future versions should include even richer trajectory-level or graph-structure-aware metrics, we agree that this is a useful direction. But the current benchmark already goes beyond endpoint-only evaluation by explicitly measuring reasoning progress during investigation.

---

> > ### Author Rebuttal · Reviewer_FtgP · 2026-04-03
> >
> > The rebuttal resolves most of my concerns, and I will maintain my current recommendation.

---

### Official Review · Reviewer_c3nb · 2026-03-11

**Soundness:** 3
**Presentation:** 3
**Significance:** 3
**Originality:** 3
**Overall Recommendation:** 4
**Confidence:** 3

**Summary:**

This paper introduces ExCyTIn-Bench, a benchmark designed to evaluate LLM agents on cyber threat investigation tasks. To address this problem, the authors construct a benchmark based on security logs collected from a simulated Azure tenant containing 57 log tables and multiple multi-stage attack scenarios. From these logs, the authors build bipartite alert–entity graphs and generate question–answer pairs anchored to specific nodes and paths in the graph. The benchmark contains 7,542 generated questions, with 589 used for testing.

Extensive experiments across multiple proprietary and open-source models show that the benchmark is challenging: even the strongest model achieves only 0.606 average reward, leaving significant room for improvement.

**Compliance With Llm Reviewing Policy:**

Affirmed.

**Key Questions For Authors:**

see weakness

**Limitations:**

see weakness

**Strengths And Weaknesses:**

Strengths:
1.The paper studies an under-explored yet practically important problem: evaluating LLM agents on cyber threat investigation tasks, which require multi-hop reasoning over heterogeneous security logs.
2. This paper is well-written and easy to follow.
3. This proposed benchmark is closer to real-world investigation settings, compared to existing cybersecurity QA benchmarks.
4. The authors conduct experiments across many models and prompting strategies and provide analysis on this benchmark.

Weakness:
1. The benchmark mainly evaluates performance using a reward score based on final answers and intermediate steps. But how reliable is the step-checking mechanism for identifying whether the agent has discovered intermediate investigation steps?

2.  The paper claims that the database scale is much larger than previous work in C.1 section, but quantitative statistics of the dataset size (e.g., number of log entries, storage size) are not clearly report. It would be helpful to better position this benchmark relative to existing datasets in terms of: task difficulty; reasoning complexity; dataset scale.

---

> ### Author Rebuttal · Authors · 2026-03-25
>
> ## Re: Weakness 1
>
> Intermediate-step checking is aligned with final-answer checking rather than implemented as an independent heuristic. Concretely, final-answer evaluation asks whether the agent’s submission contains the correct final answer, whereas the intermediate checker asks whether the submitted answer string contains key intermediate IoCs. Accordingly, intermediate-step checking uses the same underlying matching primitive as final-answer evaluation. We will clarify this point in the revision and state more explicitly that step checking is conservative: it only verifies whether key intermediate investigation terms are recovered, not whether the model’s full latent reasoning exactly matches the reference path.
>
> Then, *Appendix C.5* provides supporting evidence for the LLM-based answer checking mechanism: we manually reviewed 163 randomly selected LLM-graded cases and observed no disagreements in that sampled subset. We note that this result should be interpreted as supportive evidence rather than proof of perfect reliability since we only examine a subset, but still it shows that such checking mechanism is reliable most of the time.
>
> |          | Total questions | Submitted questions | TP | FP | TN | FN |
> | -------- | --------------: | ------------------: | -: | -: | -: | -: |
> | Combined |             163 |                 132 | 56 |  0 | 76 |  0 |
>
> Table 1 (from Appendix C.5): Confusion matrix of result of manually going through 163 evaluation results graded by LLMs. Surprisingly, we observed no disagreements in this manually reviewed sample.
>
>
>
> ## Re: Weakness 2
>
> | Benchmark / Setting                                   | Avg. tables  | Avg. rows (K) | Avg. DB size (GB) |
> |-------------------------------------------------------|-----------------:|-------------------:|------------------:|
> | Spider (avg. / DB)                                    | 3.5              | 8.6                | 0.03              |
> | Ours (avg. / incident, all 8)                         | 45               | 364.7              | 0.447             |
> | Ours (avg. / incident, excluding incident_55)         | 45               | 115.4              | 0.182             |
> | Ours (full 44-day log)                                | 61               | 14,299.5           | 18.0              |
> Table 2: Per database statistics of Spider [1] and ours.
>
> We thank the reviewer for this suggestion. We agree that the current draft should report dataset-scale statistics more explicitly to better position the benchmark relative to prior work. In the revision, we will add the quantitative comparison table presented above. In this table, we compare with Spider [1], which is a classic SQL task, the information about the database are pulled from [3]. Since InterCode-SQL[2] is constructed from Spider examples rather than being an independent SQL benchmark, we report Spider in the table [2].
>
> These per-database statistics indicate the amount of data available to answer a single question, since each Spider question is associated with one database and each question in our benchmark is associated with one incident database. Compared with Spider, our benchmark is substantially larger on a per-database basis, averaging 45 tables, 364.7K rows, and 0.447 GB per incident across all eight incidents. Because one incident (incident_55) is unusually large, we will also report a robust version excluding that outlier; even then, our benchmark still averages 45 tables, 115.4K rows, and 0.182 GB per incident, which remains markedly larger than Spider. In addition, beyond the per-incident setting, the full 44-day log available for investigation contains 61 tables, 14.3M rows, and 18 GB of data. We will incorporate these statistics into the paper and distinguish more clearly between dataset scale, reasoning complexity, and empirical difficulty.
>
> [1] Yu et al. Spider: A Large-Scale Human-Labeled Dataset for Complex and Cross-Domain Semantic Parsing and Text-to-SQL Task. EMNLP 2018.
>
> [2] Yang et al. InterCode: Standardizing and Benchmarking Interactive Coding with Execution Feedback. NeurIPS 2023 Datasets and Benchmarks Track.
>
> [3] Stockinger et al. ScienceBenchmark: A Complex Real-World Benchmark for Evaluating Natural Language to SQL Systems. PVLDB 2024.

---

> > ### Author Rebuttal · Reviewer_c3nb · 2026-03-31
> >
> > My concerns have been adequately addressed, so I maintain my original socre

---

### Decision · Program_Chairs · 2026-04-30

**Decision:**

Accept (regular)

**Comment:**

All reviewers had their concerns addressed, and would be happy to accept the paper after revisions based on the discussion. In summary, the paper provides a significantly larger and more complex dataset prior work on investigating cyber threats. Their rebuttal addressed concerns about the validity of evaluation and otherwise provided clarity that answered the reviewers questions. I recommend the paper for acceptance.